# On the Emergence of Implicit Curriculum in RLVR Learning Dynamics

**Yu Huang** [* 1] **Zixin Wen** [* 2] **Yuejie Chi** [3] **Yuting Wei** [1] **Aarti Singh** [2] **Yingbin Liang** [4] **Yuxin Chen** [1]

## Abstract

Reinforcement learning with verifiable rewards (RLVR) has been a main driver of recent breakthroughs in large reasoning models. Yet it remains a mystery how rewards based solely on final outcomes can help overcome the long-horizon barrier to extended reasoning. To understand this, we develop a theory of the training dynamics of RLVR for transformers on compositional reasoning tasks. Our theory shows that mixed-difficulty training naturally induces an **implicit curriculum**: without any explicit schedule, easier problems become learnable first and shape the frontier for harder ones, creating a learning progression from easy to hard during optimization. The effectiveness of this curriculum is governed by the smoothness of the difficulty spectrum. When the spectrum is smooth, training dynamics enter a well-behaved *relay* regime, in which persistent gradient signals on easier problems make slightly harder ones tractable and keep training at the edge of competence. When the spectrum contains abrupt discontinuities, training undergoes grokking-type phase transitions with prolonged plateaus before progress recurs. As a technical contribution, our analysis develops and adapts techniques from Fourier analysis on finite groups to our setting. We validate the predicted mechanisms empirically via synthetic experiments.

## 1. Introduction

Large language models (LLMs) such as OpenAI-o1 (OpenAI, 2024) and DeepSeek R1 (DeepSeek-AI, 2025) have shown striking performance in complex reasoning tasks. A key enabler of this recent progress is reinforcement learning with verifiable rewards (RLVR) (Shao et al., 2024; Lambert et al., 2024; Gao et al., 2024), which fine-tunes pre-trained base models via reinforcement learning (RL) using automatically verifiable, outcome-based feedback, such as a binary signal indicating whether the final answer is correct or not.

This raises an immediate question: if RLVR relies on outcome-based feedback only at the end of a reasoning trajectory, how can such a sparse reward mechanism drive effective learning on long-horizon problems? As the horizon grows, RL algorithms encounter an inherent search barrier, because useful signals are buried within an exponentially expanding space of trajectories. While recent studies have sought to understand the mechanism of RLVR (Yeo et al., 2025; Wu et al., 2025; Yue et al., 2026; Sun et al., 2025; Yuan et al., 2025; Wen et al., 2025), existing findings often provide mixed, inconclusive results across different tasks and setups. It remains unclear under what conditions outcome-only rewards are sufficient to enable effective RL.

A recent controlled study (Zhang et al., 2025) offered an important insight: RLVR is most effective near the model's *edge of competence*, where problems are neither already solved nor impossible. They conducted experiments showing that RLVR improves model's reasoning performance the most when the data sits right above the model's capability frontier, while other difficulty levels induce early plateaus. The work (Zhang et al., 2025) suggests picking problems close to the model's capability frontier, but did not answer why RLVR in practice can work on the general mixture of problems. In practice, RLVR is not done on a single fixed difficulty level but on a spectrum of problems, in which some are already learnable and others still beyond reach (Shao et al., 2024; DeepSeek-AI, 2025). This motivates the following question:

*How does the difficulty spectrum of data shape RLVR learning dynamics?*

To address this question, we study how policy-gradient dynamics evolve across controlled difficulty spectra. We adopt a minimal transformer model (Vaswani et al., 2017), which is the backbone architecture of most LLMs. It consists of a softmax-based attention layer followed by a multilayer perceptron (MLP) layer. We freeze the MLP layer in a way

---

[1]University of Pennsylvania [2]Carnegie Mellon University [3]Yale University [4]The Ohio State University. [*]Equal contribution, we recommend the full version for readability: arXiv:2602.14872. Correspondence to: Yuxin Chen <yuxinc@wharton.upenn.edu>.

*Proceedings of the $43^{rd}$ International Conference on Machine Learning*, Seoul, South Korea. PMLR 306, 2026. Copyright 2026 by the author(s).

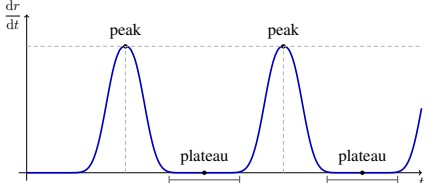
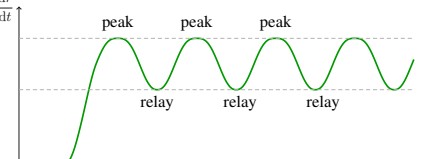
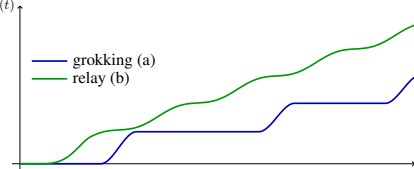

*(a) If difficulty ratio $R = L_{k+1}/L_k \gg 1$, learning exhibits grokking-type transitions in-between difficulty levels.*

*(b) If difficulty ratio $R = L_{k+1}/L_k \lesssim 1$, learning process proceeds smoothly across different difficulty levels from $L_1$ to $L_{\max}$.*

*(c) Reward trajectories $r(t)$ induced by different ratio $R$ in (a) grokking dynamics and (b) smooth relay dynamics.*

*Figure 1.* **Reward-growth dynamics in mixed-difficulty RL.** A schematic (not empirical) illustration of the reward growth rate $dr/dt$ and $r(t)$ for mixed-difficulty RL, demonstrating how the difficulty ratio $R = L_{k+1}/L_k$ changes the learning dynamics at the edge of model's competence, which yields either grokking-type phase transitions or smooth relays.

that perfectly implements the atomic operation, modeling the regime where the model already possesses the requisite atomic skills and RLVR only needs to learn how to compose these skills (Yuan et al., 2025). We study RL training on this task under outcome-based rewards via the standard policy gradient algorithm REINFORCE (Williams, 1992). Within this setting, we track the learning dynamics of the transformer model and identify the factors that govern progress across increasing horizons. Our main answer is that mixed-difficulty training naturally gives rise to an *implicit curriculum*: if the difficulty spectrum is smooth enough, the gradient signal is relayed from easier problems to slightly harder ones, keeping optimization near the edge of competence; if the spectrum contains large jumps, the same curriculum develops delayed handoffs and training exhibits grokking-like plateaus before abrupt progress. An overview of our main contributions is provided below.

1. **A comparative study between short-horizon learning and long-horizon barriers.** We show that with outcome-based rewards, REINFORCE-style policy gradient algorithms provably learn short-horizon compositions. Meanwhile, beyond a critical horizon, the gradient field at initialization becomes exponentially flat, indicating an optimization barrier for near-random policies. In contrast, supervised fine-tuning (SFT) can provably learn beyond this critical horizon by providing intermediate feedback for sequential compositional reasoning.

2. **A theoretical characterization of the learning dynamics in RLVR.** On an easy-to-hard mixture over horizons, we establish polynomial-time convergence guarantees for outcome-based RL training. We show that an *implicit curriculum* emerges in RLVR: in the smooth regime, progress transfers across adjacent horizons through a *relay effect*, which keeps training near the edge of competence and yields steady progress toward harder problems; when the spectrum contains large discontinuities, the same curriculum instead exhibits delayed progress, long plateaus, and abrupt, grokking-like phase transitions (Sun et al., 2025).

3. **Novel techniques from Fourier analysis on groups.** We introduce a Fourier analysis (Terras, 1999) frame-

work that transforms the problem of trajectory-level success conditioning into tractable calculations based on Fourier analysis for convolutions of measures. Our new framework allows us to compute the magnitudes of policy gradients in long-horizon group composition problems by resorting to the spectral properties of the group representations, which greatly simplifies the analysis of combinatorial event probabilities.

**The implicit curriculum: induced learning dynamics over difficulty spectra.** Collectively, our results provide an optimization-based explanation for how RLVR improves model's capability frontier. The high-level phenomenon is an *implicit curriculum*: when the training distribution displays a spectrum of difficulties, optimization naturally progresses from already-learnable problems to slightly harder ones without any explicit schedule. When the difficulty spectrum is smooth, the underlying mechanism is a *relay effect*: easier horizons continue to provide non-vanishing gradient signal while the next harder horizon is becoming learnable, so gradient dominance is handed off before training stalls. When the gap between adjacent difficulty levels is too large, i.e., when the spectrum is non-smooth, the same curriculum becomes delayed and unpredictable: the handoff arrives only after a prolonged plateau, followed by a grokking-style jump (illustrated in Figure 1). Thus, we conclude that RLVR's effectiveness is governed not only by exploration efficiency, but also by whether the difficulty spectrum supports a smooth learning progression.

## 2. Problem Setup

In this section, we present the formal setup for our theoretical investigation. We first define the compositional reasoning task, and then describe a minimal transformer architecture as well as the policy gradient objective used to study RL training dynamics.

### 2.1. Compositional Reasoning

To study the mechanistic challenges of multi-step reasoning, we consider the state-tracking task that has received much

recent attention (Liu et al., 2023; Merrill et al., 2024; Huang et al., 2025). This setting serves as a simple example of compositional reasoning: while each individual step is computationally simple, the task requires the precise sequential composition of $L$ transitions in order to compute the final result.

**Definition 2.1** (*L*-step compositional reasoning). Let $\mathcal{Y}$ be a finite set (the *state space*) and $\mathcal{G}$ a finite group, acting on $\mathcal{Y}$ via $(g, y) \mapsto g(y)$. For any initial state $y_0 \in \mathcal{Y}$ and sequence of transitions $g_1, \ldots, g_L \in \mathcal{G}$, we can obtain a trajectory $y_\ell = g_\ell(y_{\ell-1})$ where $\ell = 1, \ldots, L$. The goal is to predict the final state $y_L$ given the sequence $(y_0, g_1, \ldots, g_L)$. We define $L$ as the *length* or *horizon* of the problem.

Our analysis rests on some structural assumptions on the group actions of $\mathcal{G}$ on $\mathcal{Y}$, described below.

**Assumption 2.1** (Group structure and action). We assume $\mathcal{G}$ is a finite non-abelian simple group that acts simply transitively on the set $\mathcal{Y}$, which implies that there is a bijective correspondence between group elements and states, such that $|\mathcal{G}| = |\mathcal{Y}| = d$. We focus on the asymptotic regime where the dimension of the state space scales tends to infinity (i.e., $d \to \infty$).

*Remark* 2.1. Due to the inherent mixing properties of non-abelian groups (Larsen & Shalev, 2007), different sequences of operations rarely lead to the same state unless the sequences are identical. However, in abelian groups, composing the same operations in different orders yields the same result. Requiring the group to be non-abelian reduces this type of shortcut solutions and leads to cleaner analysis.

Next, we specify the format of reasoning data and the distribution over such instances.

**Definition 2.2** (Reasoning problems). Fix a set of positional identifiers $\mathcal{X} \subset \mathbb{R}^d$ consisting of mutually orthogonal vectors. We encode each token as a position–symbol pair $(x, s)$ where $x \in \mathcal{X}$ and $s \in \mathcal{G} \cup \mathcal{Y}$. A length-$L$ reasoning instance is $Z^L = (Z_p^L, Z_a^L)$ consisting of:

- *problem description (prompt):* a sequence of $L$ transition tokens:

$$Z_p^L = \big((x_{p,1}, g_1), (x_{p,2}, g_2), \ldots, (x_{p,L}, g_L)\big),$$

where $\{x_{p,\ell}\}_{1 \le \ell \le L} \subset \mathcal{X}$ are distinct and $\{g_\ell\}_{1 \le \ell \le L} \subset \mathcal{G}$.

- *compositional solution:* a sequence of $(L + 1)$ state tokens:

$$Z_a^L = \big((x_{a,0}, y_0), (x_{a,1}, y_1), \ldots, (x_{a,L}, y_L)\big),$$

where $y_\ell = g_\ell(y_{\ell-1})$ for all $\ell \in [L]$.

- *position alignment:* the prompt and solution positions are related by a fixed *unknown* permutation $\mathfrak{s} : \mathcal{X} \to \mathcal{X}$ such that $x_{a,\ell-1} = \mathfrak{s}(x_{p,\ell}), \forall \ell \in [L]$.

Since the network we will use is a transformer (Section 2.2), token positions carry no intrinsic meaning in the absence of absolute positional information. Hence, the prompt/solution format encodes only a relative correspondence and does not provide any positional shortcut. Now we are ready to define the data distribution for the reasoning problems.

**Definition 2.3** (Data distribution $\mathcal{D}^L$). Given a problem length $L$, we sample a reasoning instance $Z^L = (Z_p^L, Z_a^L)$ as in Definition 2.2 by the following process:

1. $g_1, \ldots, g_L$ is sampled from $\mathcal{G}$ uniformly without replacement;
2. the initial state $y_0$ is sampled uniformly at random from the set $\mathcal{Y}$;
3. we sample distinct prompt identifiers $\{x_{p,1}, \ldots, x_{p,L}\}$ uniformly from $\mathcal{X}$;
4. Set $x_{a,\ell-1} = \mathfrak{s}(x_{p,\ell})$ for all $\ell \in \{1, \ldots, L\}$. Additionally, $x_{a,L}$ is sampled from $\mathcal{X} \setminus \{x_{a,k}\}_{k=0}^{L-1}$.
5. the intermediate states are computed via the group action: $y_\ell = g_\ell(y_{\ell-1})$.

An instance $Z^L \triangleq (Z_p^L, Z_a^L)$ generated by the above procedure is said to be sampled from $\mathcal{D}^L$. Since both $\{g_\ell\}_{\ell=1}^L$ and $\{x_{p,\ell}\}_{\ell=1}^L$ are sampled without replacement, the length is bounded by $L \le \min\{|\mathcal{X}| - 1, d\}$.

**Assumption 2.2.** We assume $|\mathcal{X}| = \Theta(d^{c_x})$ for some constant $c_x \in (0.1, 1)$. We denote by $L_{\max} = |\mathcal{X}| - 1$ the maximum problem length.

**Embeddings and tokenizer.** We embed each group symbol $g \in \mathcal{G}$ and each state symbol $y \in \mathcal{Y}$ into $\mathbb{R}^{2d}$ via a shared orthonormal map: $\text{emb} : \mathcal{G} \cup \mathcal{Y} \to \mathbb{R}^{2d}$. With a slight abuse of notation, whenever a symbol $g \in \mathcal{G}$ or $y \in \mathcal{Y}$ is used as a model input or appears in the network computation, we write the symbol itself to denote its embedding $\text{emb}(g)$ or $\text{emb}(y)$, respectively, whenever this causes no ambiguity. In the sequel, we use $s \in \mathcal{G} \cup \mathcal{Y}$ to refer generically to a symbol from either set. We fix a bijective tokenizer $\tau : \mathcal{Y} \to [d]$ to index states for the next-state prediction, so that predicting $y \in \mathcal{Y}$ is equivalent to predicting the class label $\tau(y) \in [d]$.

### 2.2. Transformer Architecture

Building on Definitions 2.2 and 2.3, we now define a simple transformer, aimed at predicting the next state in the solution trace given the prompt and the current state token.

**Definition 2.4** (One-layer transformer (Vaswani et al., 2017)). We consider a simplified transformer consisting of a single attention module followed by a one-hidden-layer MLP. For a prompt $Z_p^L = ((x_{p,\ell}, g_\ell))_{\ell=1}^L$ and the current solution token $Z_{a,k} = (x_{a,k}, y_k)$, the transformer with parameter $\theta = (W, Q)$ outputs an (unnormalized) score vector over the next-state index in $[d]$:

$$\mathsf{TF}_\theta(Z_{a,k}, Z_p^L) = \mathsf{MLP}_W\big(\mathsf{Attn}_Q(Z_{a,k}, Z_p^L)\big) \in \mathbb{R}^d, \quad (1)$$

where the operators $\mathsf{Attn}_Q$ and $\mathsf{MLP}_W$ are defined shortly in Equation (2) and Equation (3), respectively.

**Attention Layer.** The attention module uses the current solution position $x_{a,k}$ to form weights over the prompt positions $\{x_{p,\ell}\}_{\ell=1}^L$, and returns a vector that aggregates the transition embeddings. Specifically, given a query weight $Q \in \mathbb{R}^{d \times d}$, we define

$$\mathsf{Attn}_Q(Z_{a,k}, Z_p^L) \tag{2a}$$
$$\triangleq \frac{1}{2}\Big(y_k + \sum_{\ell=1}^L \mathbf{Attn}_{a,k \to p,\ell}(Z_{a,k}, Z_p^L) \cdot g_\ell\Big) \in \mathbb{R}^{2d},$$

where the attention weight $\mathbf{Attn}_{a,k \to p,\ell}(Z_{a,k}, Z_p^L)$ is obtained by softmax-normalizing the inner products $\langle Q x_{a,k}, x_{p,\ell} \rangle$:

$$\mathsf{softmax}\Big(\big(\langle Q x_{a,k}, x_{p,1}\rangle, \ldots, \langle Q x_{a,k}, x_{p,L}\rangle\big)\Big)_\ell. \tag{2b}$$

Here, for any $u \in \mathbb{R}^n$, we denote the $i$-th entry of the softmax output as $\mathsf{softmax}(u)_i \triangleq \frac{\exp(u_i)}{\sum_{j=1}^n \exp(u_j)}$. In standard transformer architectures, the score typically takes the form $\langle W_Q x_{a,k}, W_K x_{p,\ell}\rangle$ rather than using a single matrix $Q$. We adopt the equivalent reparameterization commonly used in theoretical studies to simplify analysis without changing expressivity (Huang et al., 2024; Yang et al., 2024; Zhang et al., 2024). The factor $1/2$ normalizes the combined contribution of the residual term $y_k$ and the attention output.

**MLP layer.** Given the attention output in $\mathbb{R}^{2d}$, the MLP operator with parameter $W$ maps it to logits in $\mathbb{R}^d$ for next-state prediction (indexed by $\tau : \mathcal{Y} \to [d]$). With $m$ hidden units and ReLU activation $\sigma(z) = \max\{0, z\}$, for each $j \in [d]$,

$$\big[\mathsf{MLP}_W(Z)\big]_j = \sum_{r=1}^m \sigma\big(\langle W_{j,r}, Z\rangle\big), \quad W_{j,r} \in \mathbb{R}^{2d}.$$

In the present work, we keep $W$ fixed and assume that the MLP has already acquired pre-trained *atomic skills* for one-step transitions. Specifically, given a state $y$ and a transition symbol $g$ appearing in the form $\frac{1}{2}(g + y)$, the MLP implements the map $y \mapsto g(y)$. The structural details of the MLP and the existence of such an implementation are deferred to Section 2.3. Accordingly, our focus is on how the attention module enables long-horizon composition once the model already possesses this one-step atomic skill. This perspective aligns with recent RLVR studies that investigate compositional reasoning beyond single-step transitions (Yuan et al., 2025; Park et al., 2025).

**Induced next-state distribution.** Given the transformer's output $\mathsf{TF}_\theta(Z_{a,k}, Z_p^L)$, we define the induced next-state distribution (policy) by softmax normalization:

$$\pi_\theta\big(j \mid Z_{a,k}, Z_p^L\big) \triangleq \mathsf{softmax}\Big(\mathsf{TF}_\theta(Z_{a,k}, Z_p^L)\Big)_j \tag{3}$$

for all $j \in [d]$. Equivalently, we write $\pi_\theta(y \mid Z_{a,k}, Z_p^L) \triangleq \pi_\theta(\tau(y) \mid Z_{a,k}, Z_p^L)$ for $y \in \mathcal{Y}$. In our reasoning format, the answer positions $(x_{a,1}, \ldots, x_{a,L})$ are part of the instance and are not predicted; the model only predicts the next state symbol at each step. Starting from the prefix $Z^{L,0} = [Z_{a,0}, Z_p^L]$, the induced distribution over the generated state sequence $\widehat{y}^L = (\widehat{y}_1, \ldots, \widehat{y}_L)$ factorizes autoregressively as

$$\pi_\theta(\widehat{y}^L \mid Z^{L,0}) = \prod_{k=0}^{L-1} \pi_\theta(\widehat{y}_{k+1} \mid \widehat{Z}_{a,k}, Z_p^L), \tag{4}$$
$$\text{where } \widehat{Z}_{a,k} = (x_{a,k}, \widehat{y}_k), \ \widehat{y}_0 = y_0.$$

When it is clear from the context, we abbreviate the conditioning as $(y_0, G^L)$, where $G^L = (g_1, \ldots, g_L)$. Formally, $\pi_\theta^L(\cdot \mid y_0, G^L)$ still conditions on the full instance $Z^L$ (including $(x_{p,1:L}, x_{a,0:L})$ and the fixed permutation $\pi$); we simply suppress these positional variables in the notation.

### 2.3. Pretrained Atomic Skills

As alluded to previously, we assume that the MLP module provides a pre-trained atomic skill for single-step transitions, and we keep its parameter $W$ fixed throughout RL training. This allows us to focus on long-horizon composition in the attention dynamics.

For each output index $j \in [d]$ and hidden neuron $r \in [m]$, let us define the feature magnitude

$$V_{j,r}(s) \triangleq \langle W_{j,r}, s\rangle, \qquad s \in \mathcal{G} \cup \mathcal{Y}.$$

For each pair $(g, y)$, let $j = \tau(g(y))$ be the correct next-state index. Within the neuron group $\{W_{j,r}\}_{r \in [m]}$, we designate a unique neuron $r_{g \cdot y} \in [m]$ associated with this pair. We define $B = C_B \log d$ for some sufficiently large integer $C_B = O(1)$ and $\sigma_0 = d^{-1/2}$, and assume the features satisfy

$$V_{j,r_{g \cdot y}}(g) = B, \qquad V_{j,r_{g \cdot y}}(y) = B + 2\sigma_0; \tag{5a}$$
$$V_{j,r_{g \cdot y}}(s) = -B, \quad \forall s \in (\mathcal{G} \cup \mathcal{Y}) \setminus \{g, y\}; \tag{5b}$$
$$V_{j,r}(s) = 0, \quad \forall r \notin \{r_{g \cdot y}\}_{\tau(g(y))=j}, \ \forall s \in \mathcal{G} \cup \mathcal{Y}. \tag{5c}$$

**Proposition 2.1.** *Under Assumptions 2.1-2.2, if the MLP weight $W$ satisfies Equation (5a)–Equation (5c), then given any $Q$, for any $y_0 \in \mathcal{Y}$ and $G^1 = (g_1)$ with $g_1 \in \mathcal{G}$, we have $\pi_\theta\big(g_1(y_0) \mid y_0, G^1\big) = 1 - \frac{1}{\mathsf{poly} d}$.*

Note that for $L = 1$, the model necessarily attends to the only prompt $Z_{p,1}$. Combined with the residual connection, the MLP receives an aggregate input $\frac{1}{2}(g_1 + y_0)$. Proposition 2.1 thus guarantees that an MLP equipped with the above structural properties can perfectly implement the atomic group action.

*Remark* 2.2. At a high level, when the MLP input contains the correct pair $(g, y)$, a unique activated neuron creates a

large positive margin for the correct logit, while mismatched symbols produce canceling (negative) contributions. Consequently, the MLP predicts $g(y)$ with near-perfect accuracy. Prior analysis (Huang et al., 2025) shows that an MLP can learn such a feature-separated structure under supervised training with suitable initialization.

Therefore, we formalize this one-step capability into the following structural assumption on $W$.

**Assumption 2.3** (Pretrained MLP). *The MLP weight $W$ is fixed and satisfies Equation (5a)–Equation (5c).*

**How does the transformer reason sequentially?** Solving the $L$-step state-tracking task reduces to carrying out the *single-step* operation $L$ times: retrieve the required transition $g_\ell$ from the prompt and apply it to the current predicted state $\widehat{y}_{\ell-1}$. Under the above assumption, the pre-trained MLP contains all the atomic skills for one-step transitions, and the remaining challenge lies in *association*: the attention layer must find the correct $g_\ell$ for the current reasoning step, which shall be learned in RL or SFT.

### 2.4. Outcome-based RL Objective

We train the induced policy $\pi_\theta$ using an outcome-based RL objective with a terminal reward. Given an instance $Z^L \sim \mathcal{D}^L$ (equivalently, $(y_0, G^L)$), we generate a state sequence $\widehat{y}^L = (\widehat{y}_1, \ldots, \widehat{y}_L)$ with the policy $\pi_\theta$ via Equation (4). We assign reward 1 if the final prediction $\widehat{y}_L$ matches the true final state $y_L$, and 0 otherwise:

$$r(\widehat{y}^L \mid y_0, G^L) \triangleq \mathbb{1}\{\widehat{y}_L = y_L\},$$

where $y_L = g_L(\cdots g_1(y_0))$. The RL objective is defined as the expected terminal reward:

$$\mathcal{J}_L(\theta) = \mathbb{E}_{Z^L}\Big[\mathbb{E}_{\widehat{y}^L \sim \pi_\theta^L(\cdot | y_0, G^L)}\big[r(\widehat{y}^L \mid y_0, G^L)\big]\Big]. \quad (6)$$

For comparison, we also consider an SFT-type objective (Chu et al., 2025). Unlike the outcome-based RL objective, which provides a terminal reward only after the full rollout, this supervised objective employs teacher forcing (Huang et al., 2025; Kim & Suzuki, 2025; Wen et al., 2024; Yang et al., 2025): at each step $k$ we condition on the ground-truth current state $y_{k-1}$ and apply immediate supervision to the next state $y_k$. Formally, the SFT objective is written as

$$\mathsf{Loss}_L(\theta) \triangleq \mathbb{E}_{Z^L}\Big[\frac{1}{L}\sum_{k=1}^{L} -\log \pi_\theta(y_k \mid y_{k-1}, G^L)\Big]. \quad (7)$$

**Learning algorithm.** We consider the REINFORCE algorithm of policy gradient (Williams, 1992):

$$\nabla \mathcal{J}_L(\theta) = \mathbb{E}_{Z^L, \widehat{y}^L}\Big[r(\widehat{y}^L \mid y_0, G^L)\nabla \log \pi_\theta^L(\widehat{y}^L \mid y_0, G^L)\Big].$$

Given that $W$ is kept fixed, we study gradient ascent on $Q$ with *length-normalized* (He et al., 2025; Gao et al., 2024) policy gradient:

$$Q^{(t+1)} = Q^{(t)} + \eta \nabla_Q \widetilde{\mathcal{J}}_L(\theta^{(t)}), \quad (8)$$

where $\widetilde{\mathcal{J}}_L(\theta^{(t)}) = \frac{1}{L}\mathcal{J}_L(\theta^{(t)})$ and $\eta > 0$ represents the step size. In addition, we consider gradient descent for optimizing the SFT loss:

$$Q^{(t+1)} = Q^{(t)} - \eta \nabla_Q \mathsf{Loss}_L(\theta^{(t)}). \quad (9)$$

**Assumption 2.4** (Initialization). *At $t = 0$, $Q^{(0)}$ is initialized to be the zero matrix $\mathbf{0}_{d \times d}$.*

For simplicity, in the ensuing discussions, we let $A^{(t)}$ represent the value of $A$ at iteration $t$, dropping the explicit dependence on the $\theta^{(t)}$ or $Q^{(t)}$ when the context allows.

## 3. Learning Short-horizon Compositional Reasoning

In this section, we examine the RL dynamics of transformers on short-horizon compositional tasks. We prove that RL successfully learns compositional reasoning up to a critical horizon, beyond which a flat-gradient barrier emerges due to the nature of sparse, outcome-based rewards. In contrast, we demonstrate that SFT can overcome this limitation by leveraging immediate supervision.

### 3.1. RL for Short-horizon Compositions

Following the setup in Section 2.3, the transformer $\mathsf{TF}_{\theta^{(0)}}$ at initialization can execute the atomic one-step skills. We assume that the attention is approximately uniform at $t = 0$, so the induced policy does not reliably implement multi-step compositions. Our first result shows that, for any short horizon $L \le C_B = O(1)$ where $C_B$ is determined by the pretrained MLP parameters, policy-gradient RL successfully learns the $L$-step composition and yields the desired attention concentration pattern.

**Theorem 3.1** (RL for short-horizon problems). *Suppose Assumptions 2.1-2.3 hold, and assume that $\mathsf{TF}_{\theta^{(0)}}$ is initialized according to Assumption 2.4. Consider any $L \in [2, C_B]$ (where $C_B > 0$ is some sufficiently large constant as mentioned in Section 2.3), $\eta = \frac{1}{\mathrm{poly}(d)}$ and $\epsilon \in \left(\frac{1}{\log^{\Omega(1)}(d)}, \frac{1}{4}\right)$. Then the transformer $\mathsf{TF}_{\theta^{(t)}}$ trained via Equation (8) on the objective $\mathcal{J}_L$ after $T_{L,\epsilon} = O\left(\frac{L_{\max}\log(L/\epsilon)}{\eta \log d} \cdot d^{(1-\epsilon)C_B - 1}\right)$ iterations attains:*

*(a) **Reward optimality:** At $t = T_{L,\epsilon}$, the reward is optimal in the sense that*

$$\mathcal{J}_L^{(t)} \ge 1 - O\left(\frac{1}{d^{C_B(1-\epsilon)-1}}\right).$$

*(b) **Optimal short-horizon attention:** At $t = T_{L,\epsilon}$, for any $\ell \leq L$, we have*

$$\mathbf{Attn}^{(t)}_{a,\ell-1 \to p,\ell} \geq 1 - \epsilon.$$

Theorem 3.1 provides the first provable guarantee that a transformer can learn multi-step *compositional* reasoning via outcome-only policy gradients, even though the initialization implements only the atomic one-step skill. Our theory unveils an explicit short-horizon regime ($L \leq C_B$) in which learning is provably efficient. Beyond reward optimality, this theorem also uncovers an attention concentration pattern, providing a mechanistic characterization of how the learned transformer implements the $L$-step composition.

### 3.2. Critical Horizon and Exponentially Flat Region

When $L$ exceeds the critical threshold $C_B$, the near-uniform attention at initialization yields little informative signal to the MLP layer. As a consequence, the model behaves nearly randomly on long compositional instances. Under outcome-only rewards, the resulting policy-gradient signal becomes exponentially small even when a nonzero reward is received, as formalized in the result below.

**Proposition 3.1** (Exponentially flat region). *Suppose Assumptions 2.1-2.3 hold, and assume that the $\mathsf{TF}_{\theta^{(0)}}$ is initialized according to Assumption 2.4. Then for any horizon $L > 2C_B$, whenever the feature magnitude $\max_{x,x' \in \mathcal{X}} \langle Q^{(t)} x, x' \rangle \leq 0.01$, we have $\mathcal{J}_L^{(t)} = \frac{1}{d}(1 \pm o(1))$, and*

$$\max_{x,x' \in \mathcal{X}} \left| \left\langle [\nabla_Q \widetilde{\mathcal{J}}_L^{(t)}] x, x' \right\rangle \right| \leq \widetilde{O}\left( \frac{1}{L_{\max}} \right) \cdot d^{-\Omega(L)}.$$

**Why is the landscape flat for RL initially?** Conceptually, the initial training horizon controls the concentration of signals the model could learn from each sample, which dilutes due to the $O(d^L)$ possible trajectories if the model uniformly traverses the actions specified by the problem instance. In this case, outcome-based rewards, due to *lack of process-level feedback*, make it extremely difficult to extract out sufficient signals from policy gradients.

### 3.3. SFT Succeeds Beyond the Critical Horizon

For comparison purposes, we also consider an SFT objective as in Equation (7). Since SFT provides intermediate supervision rather than only an outcome-based reward, it remains effective even for long-horizon problems.

**Theorem 3.2** (SFT provably escapes initial flat region). *Suppose Assumptions 2.1-2.3 hold, and assume that $\mathsf{TF}_{\theta^{(0)}}$ is initialized according to Assumption 2.4. Then for any length $2 \leq L \leq \mathrm{polylog}(d)$, $\eta = \frac{1}{\mathrm{poly}(d)}$ and $\epsilon \in \left( \frac{1}{\log^{\Omega(1)}(d)}, \frac{1}{4} \right)$, the transformer $\mathsf{TF}_{\theta^{(t)}}$ trained via Equation (9) on the ob-*

*jective $\mathsf{Loss}_L$ (cf. Equation (7)) for*

$$T_{L,\epsilon} = O\left( \frac{L_{\max} \log(L/\epsilon) d^{(1-\epsilon)C_B - 1}}{\eta \epsilon \log d} + \frac{L_{\max} L}{\eta \log d} \right)$$

*iterations, satisfies:*

*(a) **Loss convergence:** At $t = T_{L,\epsilon}$, the loss converges in the sense that*

$$\mathsf{Loss}_L^{(t)} \leq O\left( \frac{1}{d^{C_B(1-\epsilon)-1}} \right).$$

*(b) **Optimal attention:** For any $\ell \leq L$, we have*

$$\mathbf{Attn}^{(t)}_{a,\ell-1 \to p,\ell} \geq 1 - \epsilon.$$

In words, Theorem 3.2 demonstrates that SFT can successfully train the transformer to solve the composition reasoning beyond the critical horizon.

## 4. Learning Dynamics of Mixed-difficulty RL

In practice, RL datasets typically contain instances of mixed complexity, due to either heterogeneous data collection or explicit curriculum strategies (Zeng et al., 2025; Parashar et al., 2025; Chen et al., 2025c), which can fundamentally affect optimization dynamics. Motivated by this observation, we study policy-gradient training under *mixed-difficulty* distributions, which is close to how large scale LLM RL is performed in production (DeepSeek-AI, 2025). Our central message is that large gaps in difficulty level lead to delayed, grokking-like handoffs, whereas smoother spectra enable a relay effect that transfers learning progress from easier to harder horizons.

### 4.1. Easy-to-Hard Mixture

To model mixed difficulty, we consider a mixture over multiple reasoning horizons. This geometric horizon schedule serves as our formalization of a *difficulty spectrum*: adjacent horizons play the role of neighboring difficulty levels through which the implicit curriculum unfolds. Let us choose *difficulty ratio* $R > 1$ and fix a starting horizon $L_1 \geq 2$. Define a set of horizons $\mathcal{L}_R = \{L_1, L_2, \ldots, L_K\}$ by the recursion:

$$L_k = \begin{cases} \lceil RL_{k-1} \rceil, & \text{if } L_{\max} > \lceil RL_{k-1} \rceil \\ L_{\max}, & \text{otherwise} \end{cases}$$

where $K = \min\{k : L_k = L_{\max}\}$ so that $L_K = L_{\max}$. We then define the mixed-difficulty objective as the uniform mixture

$$\mathcal{J}_{\mathrm{mix},R}(\theta) = \mathbb{E}_{L \sim \mathrm{Unif}(\mathcal{L}_R)} \big[ \mathcal{J}_L(\theta) \big]. \tag{10}$$

To optimize $\mathcal{J}_{\mathrm{mix},R}$, we consider a length-normalized update:

$$Q^{(t+1)} = Q^{(t)} + \eta \nabla_Q \widetilde{\mathcal{J}}_{\mathrm{mix},R}(\theta^{(t)}), \qquad (11)$$

where $\nabla_Q \widetilde{\mathcal{J}}_{\mathrm{mix},R}(\theta) = \mathbb{E}_{L \sim \mathrm{Unif}(\mathcal{L}_R)}[\nabla_Q \widetilde{\mathcal{J}}_L(\theta)]$, and recall $\widetilde{\mathcal{J}}_L(\theta) = \frac{1}{L}\mathcal{J}_L(\theta)$.

## 4.2. Grokking Dynamics Under Large Difficulty Ratios

Intuitively, under mixed-difficulty training, the shorter-horizon tasks are simpler and are therefore solved first, after which learning attempts to extend to longer horizons. When the gap between adjacent difficulty levels is large, i.e., when the mixture ratio $R$ is large, this inter-horizon handoff becomes severely delayed: the implicit curriculum turns jagged because the next horizon remains too far beyond the current competence frontier. As a result, the policy can spend an extended period with near-zero reward on the next horizon before making noticeable improvements. This plateau-and-jump pattern resembles an empirical phase-transition phenomenon dubbed *grokking* (Power et al., 2022; Sun et al., 2025): after a prolonged phase of receiving near-zero reward, the policy abruptly rises to near-perfect accuracy. Our next theorem formalizes this behavior in the mixed-horizon setting by quantitatively characterizing the length of the plateau and the ensuing transition.

To state our result, we introduce two observable states for each horizon that capture (i) when progress first becomes noticeable and (ii) when the horizon is essentially solved. **Stopping times for mastery and visible states.** For any horizon $L_k \in \mathcal{L}_R$, we say that the task at horizon $L_k$ has *visible return* at iteration $t$ if $\mathcal{J}_{L_k}^{(t)} \geq 0.01$:

$$T_{\mathrm{vis},k} \triangleq \min\left\{t : \mathcal{J}_{L_k}^{(t)} \geq 0.01\right\}. \qquad (12)$$

We say that the horizon $L_k$ is *mastered* at iteration $t$ if $\mathcal{J}_{L_k}^{(t)} \geq 0.99$:

$$T_{\mathrm{mas},k} \triangleq \min\left\{t : \mathcal{J}_{L_k}^{(t)} \geq 0.99\right\}. \qquad (13)$$

**Theorem 4.1** (Grokking dynamics). *Let $\mathcal{J}_{\mathrm{mix},R}$ be the mixed-difficulty objective with ratio $\omega(1) \leq R \leq \frac{L_{\max}}{2C_B}$ and starting horizon $L_1 = C_B$. Under Assumptions 2.1–2.4, consider the RL training process under the length-normalized update Equation (11) with step size $\eta = 1/\mathsf{poly}(d)$. Then for each $1 \leq k \leq K-2$, we have:*

*(a)* ***Long inter-difficulty plateaus.*** *Before the next horizon $L_{k+1}$ makes noticeable progress (i.e., before it enters the visible-return state), the inter-horizon plateau length satisfies*

$$T_{\mathrm{vis},k+1} - T_{\mathrm{mas},k} = \widetilde{\Theta}\left(\frac{L_{\max}}{\eta}\right) \cdot d^{C_B - 1}. \qquad (14)$$

*(b)* ***Grokking-like phase transitions.*** *Once $L_{k+1}$ enters the visible reward state, it reaches mastery quickly:* $T_{\mathrm{mas},k+1} - T_{\mathrm{vis},k+1} \leq \widetilde{O}\left(\frac{L_{\max}}{\eta}\right) \cdot L_{k+1}.$

Theorem 4.1 shows that each transition $L_k \to L_{k+1}$ consists of a long near-zero-return plateau followed by a rapid rise to mastery once return becomes visible. Aggregating these transitions yields a time-to-mastery bound for the longest horizon, in which the total runtime is dominated by the plateaus.

**Corollary 4.1.** *Under the assumptions of Theorem 4.1, suppose $c_x < \frac{C_B - 2}{C_B + 2}$, where $c_x$ is defined in Assumption 2.2. Then the first time the longest horizon $L_{\max}$ reaches mastery satisfies*

$$T_{\mathrm{mas},K} = \widetilde{\Theta}\left(\frac{L_{\max}}{\eta}\right) \cdot d^{C_B - 1} \triangleq \mathcal{T}_{\mathsf{plat}}. \qquad (15)$$

**Why does grokking happen in RL?** For long-horizon tasks, reward can either come from fully correct traces or from rare lucky guesses that reach the correct final answer despite intermediate mistakes. Before the policy can reliably generate correct traces at long horizons, the reward stays near-zero, and the gradient signal mainly consists of those from the lucky guesses, which are random and uninformative. Meanwhile, gradient updates from shorter horizons keep sharpening the internal features long after their rewards have saturated. In other words, shorter horizons continue to supply hidden structural progress, but this progress is handed off to the next horizon only after a long delay because the difficulty gap is too large. Once the longer horizon finally becomes learnable, rewards improve rapidly within a few iterations.

## 4.3. Relay Dynamics under Moderate Difficulty Ratios

The grokking dynamics above highlight that when the mixture contains well-separated horizons, training can stall at each new horizon: even after $L_k$ is mastered, the next horizon $L_{k+1}$ may remain in the near-zero-reward regime for a long plateau before its reward becomes visible. We now show the smooth counterpart: when the difficulty spectrum is sufficiently smooth (i.e., when $R$ is a moderate constant), the implicit curriculum enters a well-behaved relay regime. The underlying optimization mechanism is a *relay* effect, in which progress on easier horizons continuously supports the next harder horizon, preventing prolonged plateaus.

Our next theorem formalizes the relay regime by providing an upper bound on $T_{\mathrm{vis},k+1} - T_{\mathrm{mas},k}$, which will be significantly smaller than the long plateaus in the regime with large difficulty gaps.

**Theorem 4.2** (Relay dynamics). *Let $\mathcal{J}_{\mathrm{mix},R}$ be the mixed-difficulty objective with ratio $2 \leq R \leq O(1)$ and starting horizon $L_1 = C_B$. Under Assumptions 2.1–2.4, consider*

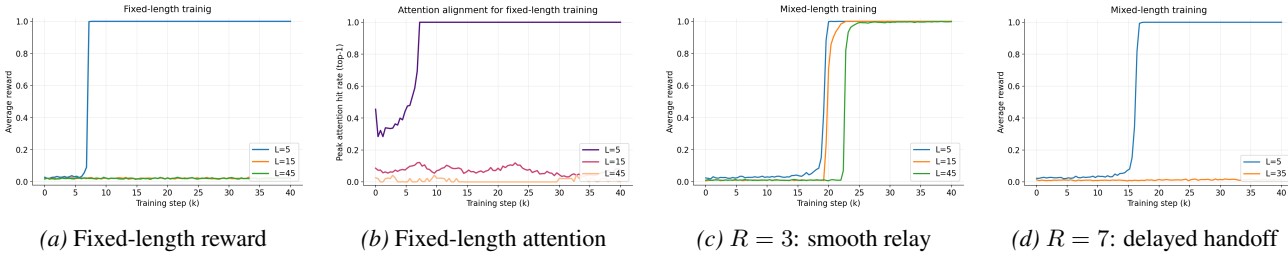

*(a)* Fixed-length reward     *(b)* Fixed-length attention     *(c)* $R = 3$: smooth relay     *(d)* $R = 7$: delayed handoff

*Figure 2.* Synthetic experiments for fixed-length and mixed-length RL training. The reward, attention, and implicit curriculum behavior are consistent with the optimization regimes predicted by the theory. Additional details are deferred to Section A.

*the RL training process under the length-normalized update Equation* (11) *with step size* $\eta = 1/\mathsf{poly}(d)$. *Then for each* $k \leq K - 2$, *before* $L_{k+1}$ *enters the visible-return state, the inter-horizon plateau length satisfies*

$$T_{\mathsf{vis},k+1} - T_{\mathsf{mas},k} \leq \widetilde{O}\Big(\frac{L_{\max}}{\eta}\Big) \cdot d^{(1 - \frac{C_B}{C_B + R})C_B - 1}.$$

*Moreover, once* $L_{k+1}$ *gains the visible return, it reaches mastery rapidly:* $T_{\mathsf{mas},k+1} - T_{\mathsf{vis},k+1} \leq \widetilde{O}\Big(\frac{L_{\max}}{\eta}\Big) L_{k+1}$.

Compared with the grokking regime, Theorem 4.2 shortens each inter-horizon plateau by a factor $d^{\Theta(1)}$. Although a smoother spectrum induces more horizons, we have $K = O(\log_R(L_{\max}/C_B)) \leq O(\log d)$. Therefore, the total time to reach mastery at the longest horizon is still governed by the (much shorter) relay plateaus, leading to a strictly faster overall convergence bound than in the large-ratio regime.

**Corollary 4.2.** *Under the assumptions of Theorem 4.2, suppose* $c_x < \frac{C_B - 2}{C_B + 2}$, *where* $c_x$ *is defined in Assumption 2.2. Then the first time that the longest horizon* $L_{\max}$ *reaches mastery satisfies* $T_{\mathsf{mas},K} \leq \mathcal{T}_{\mathsf{relay}}$, *where*

$$\mathcal{T}_{\mathsf{relay}} \leq \widetilde{O}\Big(\frac{L_{\max}}{\eta}\Big) d^{(1 - \frac{C_B}{C_B + R})C_B - 1} \leq \widetilde{O}\Big(d^{-\frac{C_B^2}{C_B + R}}\Big) \mathcal{T}_{\mathsf{plat}}.$$

**Implicit curriculum at the edge of competence.** Theorem 4.1 and Theorem 4.2 show that the mixed-horizon dynamics are governed by how quickly training can move from mastering $L_k$ to making noticeable progress on $L_{k+1}$. These are two regimes of the same implicit curriculum. When $R = O(1)$, the handoff is overlapping and smooth: progress on $L_k$ starts benefiting $L_{k+1}$ before $L_k$ fully saturates, so the next horizon gains a visible return much sooner and training relays steadily across horizons. This is the relay regime. The policy remains "just competent enough" on $L_k$ that success on $L_{k+1}$ is no longer purely random, while $L_k$ still provides a strong learning signal; the two horizons therefore improve in tandem, keeping optimization near the edge of competence (Zhang et al., 2025). In the large-$R$ regime, by contrast, the same curriculum has a delayed handoff: the policy must become overwhelmingly competent on $L_k$ before $L_{k+1}$ ceases to be random, so training exhibits a long plateau followed by a grokking-like phase transition. See Section 5.2 for a gradient-level mechanism explanation.

## 5. Proof Overview

This section explains the main proof ideas behind our learning-dynamic analysis. The central technical ingredient is a characterization of the policy-gradient signal as a function of a *step-wise probability*, which reveals an explicit long-horizon thresholding mechanism.

### 5.1. Technical Preliminaries

We first map our reasoning mechanism to some step-invariant quantities. At step $\ell$, attention weights over prompt tokens are given by a softmax of scores $\langle Q x_{a,\ell-1}, x_{p,\ell'} \rangle$. Since $x_{a,\ell-1} = \mathfrak{s}(x_{p,\ell})$, correct retrieval means that the *aligned* prompt token $x_{p,\ell}$ receives a strictly larger score than all misaligned tokens $x_{p,\ell'}$ with $\ell' \neq \ell$. With initialization $Q^{(0)} = \mathbf{0}$ and the symmetry of $\mathcal{D}^L$, the expected policy gradient update preserves a two-level score structure: for all $x \in \mathcal{X}$ and $x' \neq x$,

$$\langle Q^{(t)}\mathfrak{s}(x), x \rangle = q^{(t)}, \quad \langle Q^{(t)}\mathfrak{s}(x), x' \rangle = r^{(t)}.$$

Consequently, the attention weight is step-invariant: $\mathbf{Attn}_{a,\ell-1 \to p,\ell}^{(t)} \equiv \mathbf{Attn}_L^{(t)}$. The following lemma translates $\mathbf{Attn}_L^{(t)}$ into the corresponding three classes of step-invariant next-state probabilities.

**Lemma 5.1** (Step-invariant probability). *Given* $\mathbf{Attn}_L^{(t)}$, *with a fixed MLP in Assumption 2.3, the next-state distribution assigns step-invariant probability masses to:*

- *Target:* $p_{L,1}^{(t)} \triangleq \pi_{\theta^{(t)}}\big(g_\ell(\widehat{y}_{\ell-1}) \mid \widehat{y}_{\ell-1}, G^L\big)$;
- *Context distractor:* $p_{L,2}^{(t)} \triangleq \pi_{\theta^{(t)}}\big(g_{\ell'}(\widehat{y}_{\ell-1}) \mid \widehat{y}_{\ell-1}, G^L\big)$ *for any* $\ell' \neq \ell$;
- *Other distractor:* $p_{L,3}^{(t)} \triangleq \pi_{\theta^{(t)}}\big(y \mid \widehat{y}_{\ell-1}, G^L\big)$ *for any* $y \notin \{g_{\ell'}(\widehat{y}_{\ell-1}) \mid \ell' \in [L]\}$.

*Moreover, it holds that* $p_{L,1}^{(t)} \propto d^{C_B \mathbf{Attn}_L^{(t)}}$. *For later use, define the effective probability margin:* $\Delta_L^{(t)} := p_{L,1}^{(t)} - p_{L,3}^{(t)}$ *and* $\delta_L^{(t)} := p_{L,2}^{(t)} - p_{L,3}^{(t)}$.

### 5.2. Key Lemma for Gradient Estimation

The preceding discussion reduces the model's one-step behavior to two scalars $(q^{(t)}, r^{(t)})$. We now state the main

technical result: for length-$L$ tasks, the policy-gradient signal in the $(q^{(t)}, r^{(t)})$ coordinates admits an explicit characterization in terms of the step-level margin $\Delta_L^{(t)}$. This turns the learning dynamics into an effective one-dimensional evolution of attention concentration.

**Lemma 5.2** (Gradient characterization)**.** *Throughout the mixed-training process, given $L \in \mathcal{L}_R$, if the step-wise probability satisfies $\frac{\delta_L^{(t)}}{\Delta_L^{(t)}} \ll \frac{1}{L^2}(1 - \Delta_L^{(t)})$ and $\frac{p_{L,2}^{(t)}}{p_{L,1}^{(t)}} \ll 1 - p_{L,1}^{(t)}$, then we have*

$$\nabla_q \widetilde{\mathcal{J}}_L^{(t)} \propto (\Delta_L^{(t)})^L (1 - \Delta_L^{(t)}) . \mathcal{J}_L^{(t)} \qquad (16)$$

*Moreover, it holds that $|\nabla_r \widetilde{\mathcal{J}}_L^{(t)}| \leq O(\frac{1}{L_{\max}}) \nabla_q \widetilde{\mathcal{J}}_L^{(t)}$.*

*Remark* 5.1. Lemma 5.2 shows that the policy gradient is essentially driven by the $q$-direction, with $|\nabla_r \widetilde{\mathcal{J}}_L|$ being a lower-order term. Its magnitude exhibits two regimes: when $\Delta_L$ is small, the long-horizon factor $(\Delta_L)^L$ exponentially suppresses the gradient; when $\Delta_L \approx 1$, the update is in a convergence regime and is controlled by the shrinking term $1 - \Delta_L$. Since larger $q$ increases the target attention weight $\mathbf{Attn}_L$ and hence the margin $\Delta_L$ via Lemma 5.1, the takeaway is that length-$L$ learning is negligible until $q$ is large enough to make $\Delta_L$ moderate, after which progress slows down again as $\Delta_L \to 1$.

**Smooth vs. delayed implicit curriculum.** The key difference is whether $\nabla_q \widetilde{\mathcal{J}}_{L_{k+1}}$ becomes non-negligible before $\nabla_q \widetilde{\mathcal{J}}_{L_k}$ has decayed to a saturated signal. With Lemmas 5.1 and 5.2 in mind, activating $L_{k+1}$ in the first place requires its target attention $\mathbf{Attn}_{L_{k+1}}$ to reach a constant-level regime so that the gate $(\Delta_{L_{k+1}})^{L_{k+1}}$ is not exponentially suppressed; the handoff is then controlled by when $(\Delta_{L_{k+1}})^{L_{k+1}} \approx 1 - \Delta_{L_k}$. For large $R = \omega(1)$, reaching this regime for $L_{k+1}$ forces $q$ so large that $L_k$ is already driven to $\mathbf{Attn}_{L_k} = 1 - o(1)$, making $1 - \Delta_{L_k}$ small. $\nabla_q \widetilde{\mathcal{J}}_{L_{k+1}}$ stays negligible over a long plateau, leading to delayed curriculum handoff. For $R = O(1)$, the same catch-up happens while $L_k$ is still away from full saturation, so $\nabla_q \widetilde{\mathcal{J}}_{L_k}$ and $\nabla_q \widetilde{\mathcal{J}}_{L_{k+1}}$ overlap and jointly drive progress, which makes the handover smooth near the edge of competence.

### 5.3. Proof of Lemma 5.2: Fourier Analysis on Groups

We begin by discussing the central technical challenge in analyzing long-horizon policy gradients, and then introduce our Fourier-based techniques for tackling the challenges. **Key technical challenges.** The starting point is to express the policy gradient in terms of the one-step *action distribution* on the group. By simple transitivity, each transition $\widehat{y}_{\ell-1} \to \widehat{y}_\ell$ corresponds to a unique group element $u_\ell \in \mathcal{G}$ such that $\widehat{y}_\ell = u_\ell(\widehat{y}_{\ell-1})$. Let $\mu_\ell$ denote the one-step action law of $u_\ell$ on $\mathcal{G}$ (under the current policy), and write $\mu_\ell(g) = \mathbb{P}(u_\ell = g)$. With this notation, the step-$\ell$ gradient

reduces to a posterior-vs-prior gap for the target action:

$$\nabla_q \mathcal{J}_L \propto \sum_{\ell \in [L]} \left( \mathbb{P}(u_\ell = g_\ell \mid \widehat{y}_L = y_L) - \mu_\ell(g_\ell) \right).$$

Let $G_* = g_L \circ \cdots \circ g_1$. Simple transitivity also implies that terminal success is exactly the group-product constraint

$$\widehat{y}_L = y_L \iff u_L \circ \cdots \circ u_1 = G_*.$$

Hence, the success probability and the numerator in the posterior can be written as point evaluations of convolution products (see Section E.2 for formal definitions of convolution $*$ and Fourier transforms): $\mathbb{P}(\widehat{y}_L = y_L) = (\mu_L * \cdots * \mu_1)(G_*)$. The challenge is that these are high-order convolutions evaluated at a specific group element. When $L$ is large, a direct expansion in the group domain involves exponentially many mixed terms and offers no clean control: conditioning on $u_L \circ \cdots \circ u_1 = G_*$ couples all steps, so the posterior $\mathbb{P}(u_\ell = \cdot \mid \widehat{y}_L = y_L)$ is inherently trajectory-level and does not factorize into per-step statistics.

**Fourier analysis to estimate the dominant signal.** To make these convolution powers tractable, we pass to the Fourier domain on $\mathcal{G}$ (Kondor, 2008; Terras, 1999), where convolution becomes multiplication, turning the $L$-fold convolution into a structured product of Fourier operators: $\prod_{\ell \in [L]} \widehat{\mu}_\ell(\lambda)$. Here $\widehat{\mu}_\ell(\lambda)$ is the Fourier transform of $\mu_\ell$ at an irreducible unitary representation $\lambda$. Note that the step-invariant three-way partition of next-token outcomes in Lemma 5.1 is equivalently a three-way partition of $u_\ell$, thus

$$\mu_\ell(g) = p_{L,1} \mathbb{1}_{\{g = g_\ell\}} + p_{L,2} \mathbb{1}_{\{g \neq g_\ell \in G^L\}} + p_{L,3} \mathbb{1}_{\{g \notin G^L\}}.$$

Exploiting this structure, we obtain

$$\widehat{\mu}_\ell(\lambda) = \Delta_L \lambda(g_\ell) + \delta_L \sum_{g \in G^L \setminus \{g_\ell\}} \lambda(g),$$

which splits into a target-aligned contribution and a residual contribution from context distractors. Taking products across $L$ steps, the leading contribution arises from selecting the aligned component at each step (under mild separation conditions on $\Delta_L$ and $\delta_L$), which yields the characteristic $(\Delta_L)^L (1 - \Delta_L)$ structure in the posterior deviation.

## 6. Conclusions

In this work, we have analyzed the training dynamics of RLVR on a multi-step compositional reasoning task. To the best of our knowledge, we have provided the first end-to-end learning dynamics analysis for outcome-based RL with transformer-based policies, accompanied by explicit convergence guarantees. Our main conclusion is that RLVR scales through the emergence of an *implicit curriculum*: optimization naturally progresses across difficulty levels, and the difficulty spectrum determines whether that progression is smooth or not. Technically, we have introduced a novel Fourier analysis on groups that makes long-horizon conditioning and compositional structure tractable.

## Acknowledgement

The work of Z. Wen is supported in part by NSF DMS-2134080, DMS-2134133, CCF-2106778, and Simons Foundation grant 888970. Y. Wei is supported in part by the NSF CAREER award DMS-2143215, the NSF grants CCF-2418156, CCF-2106778, and the Wharton Dean's Research Fund. Y. Chen is supported in part by the Alfred P. Sloan Research Fellowship, the NSF grants IIS-2218773 and CIF-2221009, the ONR grants N00014-22-1-2354 and N00014-25-1-2344, the Wharton AI & Analytics Initiative's AI Research Fund, and the Amazon Research Award. The work of Y. Liang is supported in part by NSF grants DMS-2134145 and ECCS-2515482.

## Impact Statement

This paper presents work whose goal is to advance the field of Theoretical Machine Learning. There are many potential societal consequences of our work, none of which we feel must be specifically highlighted here.

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

# Supplementary Material

This appendix contains supporting materials for the main paper. Section A reports additional experimental details, Section B provides a detailed discussion of related work, and Section C-F presents the complete proofs of all theoretical claims.

## A. Experimental Details

In this section, we provide details for the synthetic experiments reported in Figure 2.

### A.1. Main Observations

**Fixed-length training.** In Figure 2, the short-horizon setting ($L = 5$) achieves nearly optimal reward together with strong attention concentration, whereas longer horizons exhibit near-flat plateaus in both reward and attention. This behavior is consistent with the predictions of Theorem 3.1 and Proposition 3.1.

**Mixed-length training.** In Figure 2, both mixed-length settings exhibit an implicit curriculum, but in different forms. The moderate-ratio setting ($R = 3$) shows a smooth relay regime, where progress transfers efficiently from easier to harder tasks. In contrast, the large-ratio setting ($R = 7$) shows a delayed-handoff regime, where the longer horizon remains near zero reward for a prolonged period. These observations validate the mixed-difficulty predictions in Theorems 4.1 and 4.2.

### A.2. General Setup

The experimental setting is designed to mirror our theoretical framework and assumptions in Section 2. We consider a cyclic group action over $\mathbb{Z}_{96}$, and study two training paradigms:

- fixed-length training with reasoning lengths $L \in \{5, 15, 45\}$;
- mixed-length training: $\mathcal{L}_3 = \{5, 15, 45\}$ and $\mathcal{L}_7 = \{5, 35\}$. The reasoning depth $L$ is stochastically assigned within each training batch ($N = 512$), where the problem length of each individual sample is uniformly drawn from $\mathcal{L}_R$ with $R$ the difficulty ratio.

We use an abelian cyclic action for experimental convenience; for the lengths considered here ($L \geq 5$), the commutativity-induced shortcut effects discussed in Remark 2.1 are mild.

**Model and training settings.** We employ a one-layer detached attention layer paired with a fixed MLP transition head. First, the transition MLP is pretrained via supervised learning to master the one-step group operation $(y, g) \mapsto y \cdot g$, after which its parameters are frozen. The attention parameters $Q$ are subsequently trained using the REINFORCE algorithm to maximize the terminal reward. We utilize an exponential moving-average baseline (momentum 0.95) and an entropy penalty ($10^{-3}$) to facilitate stable policy gradients.

**Evaluation.** We periodically compute the per-length average success rate by running greedy rollouts and calculating the fraction of episodes where the model correctly predicts the entire trajectory $\{\widehat{y}_L = y_L\}$ over 30 batches of size 512. We additionally report an attention-alignment metric, peak attention-hit rate, defined as the fraction of steps where the argmax attention weight selects the unique prompt position corresponding to the current action token, i.e., the prompt index $\mathfrak{s}^{-1}(x_{a,k})$ within the sampled set $\{x_{p,\ell}\}_{\ell=1}^{L}$. A higher hit rate indicates that the attention layer recovers the underlying permutation $\mathfrak{s}$ by locating the correct prompt position at each step, consistent with attention concentration ($\mathbf{Attn}_L \rightarrow 1$).

## B. Related Work and Discussions

### B.1. Related Work

**Empirical understanding of RLVR mechanisms.** With RLVR's recent success and apparent scalability, there has been growing interest in understanding its mechanics, sparking an active debate: *what does RLVR actually teach base LLMs beyond pre-training?* The existing literature provides mixed evidence. Some works characterize RL primarily as a capability refiner or reranker (Yeo et al., 2025; Wu et al., 2025; Yue et al., 2026; Zhao et al., 2025), whereas others argue that RL can induce substantial reasoning gains beyond the base model (Sun et al., 2025; Yuan et al., 2025; Wen et al., 2025; Liu et al., 2025). Related findings suggest that RLVR can yield notable improvements even from spurious rewards (Shao et al., 2025) or extremely limited RL data (e.g., one-shot) (Wang et al., 2025b). Complementary evidence further highlights an

entropy-based mechanism as a potential driver of such gains (Cui et al., 2025; Wang et al., 2025a). Recently, (Zhang et al., 2025) helps reconcile these views by showing that genuine capability gains arise mainly when there is sufficient headroom beyond pre-training and the RL data are calibrated to the model's edge of competence.

**Theory of RL training for LLM.** The empirical success of RLVR has spurred theoretical studies from various perspectives (Chen et al., 2025a;b; Zhu et al., 2025; Ran-milo et al., 2026; Rad et al., 2026; Lyu et al., 2025; Bu et al., 2025; Tsilivis et al., 2025; Davis & Recht, 2025). Of particular relevance are gradient-based analyses with transformer policies (Lyu et al., 2025; Bu et al., 2025; Ran-milo et al., 2026). (Bu et al., 2025) formalize the benefits of curriculum-style RL post-training, while (Lyu et al., 2025) studies learnability in settings with intermediate supervision. In contrast, we focus on outcome-based RL where no dense feedback is available. A closely related work (Ran-milo et al., 2026) shows that outcome-based RL can induce step-by-step reasoning under gradient flow, with asymptotic guarantees under static data conditions. We go beyond this by analyzing realistic gradient dynamics along the full learning trajectory and characterizing phase transitions in both reward and gradient.

**Curriculum and difficulty-aware RL.** Curriculum learning in RL studies how sequencing tasks or samples can make hard problems learnable (Narvekar et al., 2020). Recent LLM reasoning work uses easy-to-hard schedules (Parashar et al., 2025), learned curriculum policies (Chen et al., 2025c), and online filtering or frontier-of-learnability sampling to focus training on intermediate-difficulty problems with useful success signals (Bae et al., 2025; Foster et al., 2025). Since generalization across difficulty levels is limited (Kordi et al., 2026), broad and smooth difficulty spectra can be preferable to relying only on easy or hard subsets. Related hard-problem exploration methods reduce effective difficulty through privileged or off-policy prefixes, as in POPE (Qu et al., 2026) and PrefixRL (Setlur et al., 2026). These works support the real-world relevance of our mechanism: sparse terminal rewards become useful when data or prompting keeps training near the current competence frontier.

**Grokking in supervised learning and RL.** Grokking characterizes delayed generalization in supervised learning, where performance stays flat for long periods before improving abruptly (Power et al., 2022). Similar plateau-to-jump dynamics also appear in RLVR training (Sun et al., 2025), and have been described as "aha moments" in RL systems such as DeepSeek-R1-Zero (DeepSeek-AI, 2025). Prior work investigates mechanisms behind such phase changes (Nanda et al., 2023; Kumar et al., 2024; Liu et al., 2022; Tian, 2025), while our theory on the learning dynamics provides a mechanistic explanation for both the plateau and the subsequent transition in RLVR.

**Prior use of group representations in machine learning.** Group representation theory has been widely explored in machine learning to model and exploit symmetry (Esteves, 2020; Marchetti et al., 2024) and to analyze structured distributions (Chen et al., 2020; Kondor, 2008). Our use is different in both setting and goal: we bring Fourier analysis into the study of *long-horizon, compositional* RL objectives, where the terminal success event couples all steps and makes policy-gradient estimation inherently trajectory-dependent. The group structure turns this global conditioning into an $L$-fold convolution object that can be controlled spectrally.

### B.2. Discussions

**Limitations and future directions.** One major limitation of our study is that we focus on a single dimension of curriculum structure: we control for all other factors and only scale the length of compositions in the data distribution. This simplification allows us to study the optimization dynamics of RL in a clean, abstracted setting. However, this design choice also limits the generality of our conclusions. In real world reasoning problems, hard reasoning problems may be different from easy problems in many aspects. For example, they may require more long-tailed distributed atomic skills than easy problems, and they may sit in very different semantic contexts than easy problems, both of which we cannot study theoretically yet. There are also other reasoning patterns, such as planning and search, which we cannot analyze in the current setting. We hope our work can inspire future research into these topics.

## C. Preliminaries

In this section, we introduce some useful notation and derive several preliminary policy gradient lemmas, which will be repeatedly used in the subsequent training-dynamics analysis.

**Notation.** For functions $h, g$, write $h(x) = \Omega(g(x))$ (resp. $O(g(x))$) if there exist universal constants $C > 0$ and $a$ such that $|h(x)| \geq C|g(x)|$ (resp. $\leq C|g(x)|$) for all $x \geq a$; write $h(x) = \Theta(g(x))$ if both bounds hold. We write $h(x) = o(g(x))$ if $\lim_{x \to \infty} \frac{h(x)}{g(x)} = 0$, and $h(x) = \omega(g(x))$ if $\lim_{x \to \infty} \frac{h(x)}{g(x)} = \infty$. Let $\mathbb{1}\{\cdot\}$ be the indicator and $[L] = 1, \ldots, L$. We use

$\widetilde{O}$, $\widetilde{\Theta}$ and $\widetilde{\Omega}$ to suppress logarithmic factors, and use $\mathrm{poly}(d)$ and $\mathrm{polylog}(d)$ for polynomials in $d$ and $\log d$, respectively. Throughout the proof, we use $d_{\mathsf{p}}$ to denote $|\mathcal{X}|$.

## C.1. Gradient Computations

**Notation for gradient expressions.** Consider a problem instance of length $L$, we denote a full answer trajectory with initial prefix $Z_{a,0}$ by

$$\widehat{Z}_a^L \triangleq \left(Z_{a,0}, \widehat{Z}_{a,1}, \ldots, \widehat{Z}_{a,L}\right).$$

For each $1 \le \ell \le L$, we denote the partial trajectory up to step $\ell$ (including the initial prefix) by

$$\widehat{Z}_a^{L,\ell} \triangleq \left(Z_{a,0}, \widehat{Z}_{a,1}, \ldots, \widehat{Z}_{a,\ell}\right).$$

We introduce the following shorthand notation (for $j \in \tau(\mathcal{Y})$, $r \in [m]$, and $\ell, k \in [L]$):

$$\mathcal{E}_j(\widehat{Z}_a^{L,\ell}, Z_p^L) \triangleq \mathbb{1}_{\tau(\widehat{y}_\ell)=j} - \pi_\theta(j \mid \widehat{y}_{\ell-1}, G^L), \tag{17a}$$

$$\Lambda_{j,r}(\widehat{Z}_a^{L,\ell-1}, Z_p^L) \triangleq \frac{1}{2}\left(\langle W_{j,r}, \widehat{Z}_{a,k-1}\rangle + \sum_{k\in[L]} \mathbf{Attn}_{a,\ell-1\to p,k} \cdot \langle W_{j,r}, Z_{p,k}\rangle\right). \tag{17b}$$

$$\Xi_{\ell,k}(\widehat{Z}_a^{L,\ell}, Z_p^L) \triangleq \frac{1}{2}\sum_{j\in\tau(\mathcal{Y})} \mathcal{E}_j(\widehat{Z}_a^{L,\ell}, Z_p^L) \sum_{r\in[m]} \sigma'\left(\Lambda_{j,r}(\widehat{Z}_a^{L,\ell-1}, Z_p^L)\right)\langle W_{j,r}, Z_{p,k}\rangle. \tag{17c}$$

Here, $\pi_\theta(j \mid \widehat{y}_{\ell-1}, G^L) = \mathsf{softmax}\left(\mathsf{TF}_\theta\left(\widehat{Z}_{a,\ell-1}, Z_p^L\right)\right)_j$.

*Fact* C.1 (Gradients of Q). Given a problem length $L$, we have the following expression for the policy gradient w.r.t. the attention matrix $Q$:

$$\nabla_Q \widetilde{\mathcal{J}}_L = \frac{1}{L}\mathbb{E}_{Z^L, \widehat{y}^L}\left[\mathbb{1}_{\widehat{y}_L=y_L} \sum_{\ell=1}^L \nabla_Q \log \pi_\theta(\widehat{y}_\ell \mid \widehat{y}_{\ell-1}, G^L)\right],$$

where

$$\nabla_Q \log \pi_\theta(\widehat{y}_\ell \mid \widehat{y}_{\ell-1}, G^L)$$
$$= \sum_{k\in[L]} \mathbf{Attn}_{a,\ell-1\to p,k} \cdot \left(\Xi_{\ell,k}(\widehat{Z}_a^{L,\ell}, Z_p^L) - \sum_{k'\in[L]} \mathbf{Attn}_{a,\ell-1\to p,k}\Xi_{\ell,k'}(\widehat{Z}_a^{L,\ell}, Z_p^L)\right) x_{a,\ell-1}x_{p,k}^\top.$$

Moreover, the gradient of $Q$ for the supervised loss $\mathsf{Loss}_L$ can be written as:

$$-\nabla_Q \mathsf{Loss}_L$$
$$= \frac{1}{L}\mathbb{E}_{Z^L}\left[\sum_{\ell=1}^L \sum_{k=1}^L \mathbf{Attn}_{a,\ell-1\to p,k} \cdot \left(\Xi_{\ell,k}(Z_a^{L,\ell}, Z_p^L) - \sum_{k'\in[L]} \mathbf{Attn}_{a,\ell-1\to p,k}\Xi_{\ell,k'}(Z_a^{L,\ell}, Z_p^L)\right) x_{a,\ell-1}x_{p,k}^\top\right].$$

Then we consider the gradient of $x^\top \nabla_Q \log \pi_\theta(\widehat{y}_\ell \mid \widehat{y}_{\ell-1}, G^L)x'$ for $x, x' \in \mathcal{X}$.

**Lemma C.1.** *Given $x, x' \in \mathcal{X}$, if $x = \mathfrak{s}(x')$, when $x_{a,\ell-1} = x$ for $\ell \in [L]$, then we have*

$$x^\top \nabla_Q \log \pi_\theta(\widehat{y}_\ell \mid \widehat{y}_{\ell-1}, G^L)x'$$
$$= \mathbf{Attn}_{a,\ell-1\to p,\ell} \cdot \left(\Xi_{\ell,\ell}(\widehat{Z}_a^{L,\ell}, Z_p^L) - \sum_{k'\in[L]} \mathbf{Attn}_{a,\ell-1\to p,k'}\Xi_{\ell,k'}(\widehat{Z}_a^{L,\ell}, Z_p^L)\right)$$
$$= \mathbf{Attn}_{a,\ell-1\to p,\ell} \cdot \left(\sum_{j\in\tau(\mathcal{Y})} \mathcal{E}_j(\widehat{Z}_a^{L,\ell}, Z_p^L) \sum_{r\in[m]} \sigma'\left(\Lambda_{j,r}(\widehat{Z}_a^{L,\ell-1}, Z_p^L)\right) \cdot \left(\langle W_{j,r}, Z_{p,\ell}\rangle - \Lambda_{j,r}(\widehat{Z}_a^{L,\ell-1}, Z_p^L)\right)\right).$$

**Lemma C.2.** *Given $x, x' \in \mathcal{X}$, if $x \neq \mathfrak{s}(x')$, when $x_{p,\ell'} = x'$ and $x_{a,\ell-1} = x$ for $\ell, \ell' \in [L]$, noticing that $\ell' \neq \ell$, then we have*

$$
x^\top \nabla_Q \log \pi_\theta(\widehat{y}_\ell \mid \widehat{y}_{\ell-1}, G^L)x'
$$

$$
= \mathbf{Attn}_{a,\ell-1 \to p,\ell'} \cdot \left( \Xi_{\ell,\ell'}(\widehat{Z}_a^{L,\ell}, Z_p^L) - \sum_{k' \in [L]} \mathbf{Attn}_{a,\ell-1 \to p,k'} \Xi_{\ell,k'}(\widehat{Z}_a^{L,\ell}, Z_p^L) \right)
$$

$$
= \mathbf{Attn}_{a,\ell-1 \to p,\ell'} \cdot \left( \sum_{j \in \tau(\mathcal{Y})} \mathcal{E}_j(\widehat{Z}_a^{L,\ell}, Z_p^L) \sum_{r \in [m]} \sigma'(\Lambda_{j,r}(\widehat{Z}_a^{L,\ell-1}, Z_p^L)) \cdot \left( \langle W_{j,r}, Z_{p,\ell'} \rangle - \Lambda_{j,r}(\widehat{Z}_a^{L,\ell-1}, Z_p^L) \right) \right).
$$

Observe that there are $d^{L-1}$ intermediate-state sequences $(\widehat{y}_1, \ldots, \widehat{y}_{L-1}) \in \mathcal{Y}^{L-1}$ that lead to $\widehat{y}_L = y_L$. Only these trajectories yield a nonzero terminal reward. Hence, it suffices to restrict our attention to their contributions. Combining this with the preceding lemmas, a direct calculation gives

**Lemma C.3.** *Given $x, x' \in \mathcal{X}$, we have*

- *if $x = \mathfrak{s}(x')$, then*

$$
x^\top \nabla_Q \widetilde{\mathcal{J}}_L x' = \frac{1}{2Ld_{\mathsf{p}}} \mathbb{E}_{Z^L} \left[ \sum_{v_{1:L} \in \mathcal{Y}^{L-1} \times \{y_L\}} \left( \prod_{\ell'=1}^{L} \pi_\theta(v_{\ell'} \mid v_{\ell'-1}, G^L) \right) \cdot \left( \sum_{\ell=1}^{L} \mathbf{Attn}_{a,\ell-1 \to p,\ell} \right. \right.
$$

$$
\left. \left. \left( \sum_{j \in \tau(\mathcal{Y})} \mathcal{E}_j(\widehat{Z}_a^{L,\ell}, Z_p^L) \sum_{r \in [m]} \sigma'(\Lambda_{j,r}(\widehat{Z}_a^{L,\ell-1}, Z_p^L)) \cdot \left( \langle W_{j,r}, Z_{p,\ell} \rangle - \Lambda_{j,r}(\widehat{Z}_a^{L,\ell-1}, Z_p^L) \right) \right) \Big|_{\widehat{y}_{1:\ell} = v_{1:\ell}} \right) \right].
$$

- *else,*

$$
x^\top \nabla_Q \widetilde{\mathcal{J}}_L x' = \frac{1}{2Ld_{\mathsf{p}}(d_{\mathsf{p}} - 1)} \mathbb{E}_{Z^L} \left[ \sum_{v_{1:L} \in \mathcal{Y}^{L-1} \times \{y_L\}} \left( \prod_{\ell'=1}^{L} \pi_\theta(v_{\ell'} \mid v_{\ell'-1}, G^L) \right) \cdot \left( \sum_{\ell=1}^{L} \sum_{\ell' \neq \ell} \mathbf{Attn}_{a,\ell-1 \to p,\ell'} \right. \right.
$$

$$
\left. \left. \left( \sum_{j \in \tau(\mathcal{Y})} \mathcal{E}_j(\widehat{Z}_a^{L,\ell}, Z_p^L) \sum_{r \in [m]} \sigma'(\Lambda_{j,r}(\widehat{Z}_a^{L,\ell-1}, Z_p^L)) \cdot \left( \langle W_{j,r}, Z_{p,\ell'} \rangle - \Lambda_{j,r}(\widehat{Z}_a^{L,\ell-1}, Z_p^L) \right) \right) \Big|_{\widehat{y}_{1:\ell} = v_{1:\ell}} \right) \right].
$$

*Proof.* The two results are obtained by invoking Lemma C.1 and Lemma C.2, respectively. Specifically, for a given $\ell$, the event $\{x_{a,\ell-1} = x\}$ occurs with probability $1/d_{\mathsf{p}}$, while for $k \neq \ell$, the event $\{x_{p,\ell-1} = \mathfrak{s}^{-1}(x), x_{p,k} = x', x' \neq \mathfrak{s}^{-1}(x)\}$ occurs with probability $\frac{1}{d_{\mathsf{p}}(d_{\mathsf{p}}-1)}$. The remaining calculation details are omitted here for brevity. $\square$

Similarly, for the gradient of SFT loss $\mathsf{Loss}_L$, we have

**Lemma C.4.** *Given $x, x' \in \mathcal{X}$, we have*

- *if $x = \mathfrak{s}(x')$, then*

$$
-x^\top \nabla_Q \mathsf{Loss}_L x' = \frac{1}{2Ld_{\mathsf{p}}} \mathbb{E}_{Z^L} \left[ \sum_{\ell=1}^{L} \mathbf{Attn}_{a,\ell-1 \to p,\ell} \right.
$$

$$
\left. \left( \sum_{j \in \tau(\mathcal{Y})} \mathcal{E}_j(Z_a^{L,\ell}, Z_p^L) \sum_{r \in [m]} \sigma'(\Lambda_{j,r}(Z_a^{L,\ell-1}, Z_p^L)) \cdot \left( \langle W_{j,r}, Z_{p,\ell} \rangle - \Lambda_{j,r}(Z_a^{L,\ell-1}, Z_p^L) \right) \right) \right].
$$

- *else,*

$$
-x^\top \nabla_Q \mathsf{Loss}_L x' = \frac{1}{2Ld_{\mathsf{p}}(d_{\mathsf{p}} - 1)} \mathbb{E}_{Z^L} \left[ \sum_{\ell=1}^{L} \sum_{\ell' \neq \ell} \mathbf{Attn}_{a,\ell-1 \to p,\ell'} \right.
$$

$$
\left. \left( \sum_{j \in \tau(\mathcal{Y})} \mathcal{E}_j(Z_a^{L,\ell}, Z_p^L) \sum_{r \in [m]} \sigma'(\Lambda_{j,r}(Z_a^{L,\ell-1}, Z_p^L)) \cdot \left( \langle W_{j,r}, Z_{p,\ell'} \rangle - \Lambda_{j,r}(Z_a^{L,\ell-1}, Z_p^L) \right) \right) \right].
$$

We further introduce additional notation to simplify the presentation. Given $G^L = \{g_1, \cdots, g_L\}$ and the initial value $y_0$ with induced $\{y_1, \cdots, y_L\}$, define

$$\mathfrak{J}(\theta; y_0, G^L) \triangleq \sum_{\boldsymbol{v} \in \mathcal{Y}^{L-1} \times \{y_L\}} \left( \prod_{\ell'=1}^{L} \pi_\theta\left(v_{\ell'} \mid v_{\ell'-1}, G^L\right) \right) \left( \sum_{\ell=1}^{L} \mathbf{Attn}_{a,\ell-1 \to p,\ell} \mathfrak{G}_\ell(\theta; \boldsymbol{v}) \right) \tag{18}$$

where $\mathfrak{G}_\ell(\theta; \boldsymbol{v}) \triangleq \sum_{j \in \tau(\mathcal{Y})} \mathcal{E}_j(\widehat{Z}_a^{L,\ell}, Z_p^L) \sum_{r \in [m]} \sigma'\left(\Lambda_{j,r}(\widehat{Z}^{L,\ell-1})\right) \cdot \left(\langle W_{j,r}, Z_{p,\ell}\rangle - \Lambda_{j,r}(\widehat{Z}^{L,\ell-1})\right)\Big|_{\widehat{y}_{1:\ell}=v_{1:\ell}}. \tag{19}$

In what follows, we suppress the dependence on $\theta$ and write $\mathfrak{J}(y_0, G^L)$ and $\mathfrak{G}_\ell(\boldsymbol{v})$ for brevity.

**Notations for scalarized attention dynamics.** Based on the gradient update, the quantity $\langle x, Qx'\rangle$ takes only two possible values, depending on whether $x = \mathfrak{s}(x')$ (the matched position) or $x \neq \mathfrak{s}(x')$ (a mismatched position). Accordingly, we define the (unnormalized) *target* and *non-target* attention scores as

$$q \triangleq \langle Q\,\mathfrak{s}(x), x\rangle, \qquad x \in \mathcal{X}, \tag{20a}$$

$$r \triangleq \langle Q\,\mathfrak{s}(x), x'\rangle, \qquad x' \in \mathcal{X} \setminus \{x\}. \tag{20b}$$

With this notation, Lemma C.3 can be viewed as a policy-gradient update on $(q, r)$. Hence, it suffices to track the dynamics of these two scalars in the sequel. Thus, following Equation (18), we have

$$\nabla_q \widetilde{\mathcal{J}}_L = \frac{1}{2Ld_{\mathsf{p}}} \mathbb{E}_{Z^L} \left[ \mathfrak{J}(y_0, G^L) \right]. \tag{21}$$

Furthermore, under this reduction, for a fixed problem length $L$, the attention weights $\mathbf{Attn}_{a,\ell-1 \to p,\ell'}$ (for $\ell, \ell' \in [L]$) take only two distinct values depending on whether the prompt position matches:

$$\mathbf{Attn}_{a,\ell-1 \to p,\ell} = \frac{e^q}{e^q + (L-1)e^r}, \tag{22a}$$

$$\mathbf{Attn}_{a,\ell-1 \to p,\ell'} = \frac{e^r}{e^q + (L-1)e^r}, \qquad \ell' \neq \ell. \tag{22b}$$

When the context is clear, we denote the target attention weight $\mathbf{Attn}_{a,\ell-1 \to p,\ell}$ by $\mathbf{Attn}_L$ for brevity.

## C.2. Some Useful Bounds

**Notation for activated neurons.** Fix an output index $j \in \tau(\mathcal{Y})$, define the fiber $\mathfrak{F}_j \triangleq \{(g, y) \in \mathcal{G} \times \mathcal{Y} : \tau(g(y)) = j\}$, i.e., the set of transition–state pairs whose next state is tokenized as $j$. For each $(g, y) \in \mathfrak{F}_j$, let $r_{g \cdot y}$ denote the (unique) neuron in the pre-trained MLP that is activated for predicting $j = \tau(g(y))$ as defined in Equation (5). We further define the set of all activated neurons

$$\mathfrak{A} \triangleq \cup_{j \in \tau(\mathcal{Y})} \mathfrak{A}_j, \quad \text{where } \mathfrak{A}_j \triangleq \{r \mid \exists (g, y) \in \mathfrak{F}_j, r = r_{g \cdot y}\}.$$

Equivalently, $\mathfrak{A}$ collects the activated neurons across all fibers $\{\mathfrak{F}_j\}_{j \in [n_Y]}$.

Substituting the conditions from Equation (5) yields the following characterizations of $\Lambda_{j,r}$.

**Lemma C.5** (Characterizations of $\Lambda$). *Given $\ell \in [L]$ and $(\widehat{Z}_a^{L,\ell-1}, Z_p^L)$. Let $\{\mathbf{Attn}_{a,\ell-1 \to p,k}\}_{k=1}^L$ denote the attention weights from the answer token at step $\ell - 1$ to the $L$ prompt tokens. Then we have:*

*(a) For any $j \in \tau(\mathcal{Y})$ and any activated neuron $r \in \mathfrak{A}_j$,*

$$\Lambda_{j,r}(\widehat{Z}_a^{L,\ell-1}, Z_p^L) = \frac{1}{2}\left(V_{j,r}(\widehat{y}_{\ell-1}) + \sum_{k=1}^L \mathbf{Attn}_{a,\ell-1 \to p,k} V_{j,r}(g_k)\right).$$

*(b) For any $j \in \tau(\mathcal{Y})$ and any non-activated neuron $r \notin \mathfrak{A}_j$,*

$$\Lambda_{j,r}(\widehat{Z}_a^{L,\ell-1}, Z_p^L) = 0.$$

**Lemma C.6** (Values of $\Lambda$ at step $\ell$). *Fix $\ell \in [L]$ and an input $(\widehat{Z}_a^{L,\ell-1}, Z_p^L)$. Let $\{\mathbf{Attn}_{a,\ell-1\to p,k}\}_{k=1}^L$ denote the attention weights. Then the following properties hold.*

*(a)* ***target transition*** *$g_\ell$. Let $j := \tau\big(g_\ell \cdot \widehat{y}_{\ell-1}\big)$. Then*

$$\Lambda_{j,\,r_{g_\ell \cdot \widehat{y}_{\ell-1}}}(\widehat{Z}_a^{L,\ell-1}, Z_p^L) = \mathbf{Attn}_{a,\ell-1\to p,\ell}\, B + \sigma_0,$$

*and for any $r \in \mathfrak{A}_j \setminus \{r_{g_\ell \cdot \widehat{y}_{\ell-1}}\}$,*

$$\Lambda_{j,r}(\widehat{Z}_a^{L,\ell-1}, Z_p^L) < 0.$$

*(b)* ***in-context distractor*** *$g_k$, $k \neq \ell$. Fix $k \in [L] \setminus \{\ell\}$ and let $j := \tau\big(g_k \cdot \widehat{y}_{\ell-1}\big)$. Then*

$$\Lambda_{j,\,r_{g_k \cdot \widehat{y}_{\ell-1}}}(\widehat{Z}_a^{L,\ell-1}, Z_p^L) = \mathbf{Attn}_{a,\ell-1\to p,k}\, B + \sigma_0,$$

*and for any $r \in \mathfrak{A}_j \setminus \{r_{g_k \cdot \widehat{y}_{\ell-1}}\}$,*

$$\Lambda_{j,r}(\widehat{Z}_a^{L,\ell-1}, Z_p^L) < 0.$$

*(c)* ***vocabulary distractor*** *$g$. For any $g \in \mathcal{G} \setminus G^L$ and $j := \tau\big(g \cdot \widehat{y}_{\ell-1}\big)$,*

$$\Lambda_{j,\,r_{g \cdot \widehat{y}_{\ell-1}}}(\widehat{Z}_a^{L,\ell-1}, Z_p^L) = \sigma_0,$$

*and for any $r \in \mathfrak{A}_j \setminus \{r_{g \cdot \widehat{y}_{\ell-1}}\}$,*

$$\Lambda_{j,r}(\widehat{Z}_a^{L,\ell-1}, Z_p^L) < 0.$$

*Proof.* By Lemma C.5, for any $j \in \tau(\mathcal{Y})$ and any neuron $r \in \mathfrak{A}_j$, we have

$$\Lambda_{j,r}(\widehat{Z}_a^{L,\ell-1}, Z_p^L) = \frac{1}{2}\Big(V_{j,r}(\widehat{y}_{\ell-1}) + \sum_{k=1}^L \mathbf{Attn}_{a,\ell-1\to p,k} V_{j,r}(g_k)\Big).$$

By the simple transitivity assumption, there exists a unique $g^\star \in \mathcal{G}$ such that $\tau\big(g^\star(\widehat{y}_{\ell-1})\big) = j$. Invoking Equation (5), we have

$$V_{j,r_{g^\star \cdot \widehat{y}_{\ell-1}}}(\widehat{y}_{\ell-1}) = B + 2\sigma_0.$$

If $g^\star = g_k \in G^L$ for some $k \in [L]$, then $V_{j,r_{g^\star \cdot \widehat{y}_{\ell-1}}}(g_k) = B$ and $V_{j,r_{g^\star \cdot \widehat{y}_{\ell-1}}}(g_{k'}) = -B$ for all $k' \neq k$. Therefore,

$$2\Lambda_{j,r_{g^\star \cdot \widehat{y}_{\ell-1}}}(\widehat{Z}_a^{L,\ell-1}, Z_p^L) = \mathbf{Attn}_{a,\ell-1\to p,k} B + \sum_{k' \neq k} \mathbf{Attn}_{a,\ell-1\to p,k'}(-B) + (B + 2\sigma_0)$$

$$= \mathbf{Attn}_{a,\ell-1\to p,k} B - (1 - \mathbf{Attn}_{a,\ell-1\to p,k})B + B + 2\sigma_0$$

$$= 2\mathbf{Attn}_{a,\ell-1\to p,k} B + 2\sigma_0.$$

Otherwise, if $g^\star \notin G^L$, then $V_{j,r_{g^\star \cdot \widehat{y}_{\ell-1}}}(g_k) = -B$ for all $k \in [L]$, and hence

$$2\Lambda_{j,r_{g^\star \cdot \widehat{y}_{\ell-1}}}(\widehat{Z}_a^{L,\ell-1}, Z_p^L) = \sum_{k=1}^L \mathbf{Attn}_{a,\ell-1\to p,k}(-B) + (B + 2\sigma_0) = 2\sigma_0.$$

Finally, consider any other pair $(g, y)$ such that $\tau(g(y)) = j$ but $y \neq \widehat{y}_{\ell-1}$. For the corresponding neuron $r_{g \cdot y} \in \mathfrak{A}_j$, we have $V_{j,r_{g \cdot y}}(\widehat{y}_{\ell-1}) = -B$. Moreover, among $\{g_k\}_{k=1}^L$, at most one index can contribute $+B$ and the remaining contribute $-B$, so

$$2\Lambda_{j,r_{g \cdot y}}(\widehat{Z}_a^{L,\ell-1}, Z_p^L) \leq \Big(2 \max_{k \in [L]} \mathbf{Attn}_{a,\ell-1\to p,k} - 1\Big)B - B$$

$$< 0.$$

This concludes the proof. $\qquad\square$

Throughout the following analysis, we suppress the dependence on the underlying instance. When the context is clear, we abbreviate $\Lambda_{j,r}(\widehat{Z}_a^{L,\ell-1}, Z_p^L)$ as $\Lambda_{j,r}$ and $\mathcal{E}_j(\widehat{Z}_a^{L,\ell}, Z_p^L)$ as $\mathcal{E}_j$.

Hence, combining the reduced attention pattern in Equation (22), namely, one target receiving weight $\mathbf{Attn}_L^{(t)}$ and $L-1$ symmetric non-targets, the above characterization of the activations $\Lambda_{j,r}$ implies a step-invariant, context-level structure for the next-state distribution $\pi_\theta(\cdot \mid \widehat{y}_{\ell-1}, G^L)$. In particular, the candidates decompose into three groups: (i) the target $g_\ell \cdot \widehat{y}_{\ell-1}$; (ii) the $L-1$ symmetric non-targets $\{g_{\ell'} \cdot \widehat{y}_{\ell-1} : \ell' \neq \ell\}$; and (iii) the remaining $d-L$ states outside the context induced set (i.e., vocabulary distractors). We formalize this decomposition in the following lemma.

**Lemma C.7.** *At step $\ell$, conditioning on $\widehat{y}_{\ell-1}$ and $G^L$, the policy $\pi_\theta^{(t)}(\cdot \mid \widehat{y}_{\ell-1}, G^L)$ satisfies:*

*(i) For $j = \tau(g_\ell(\widehat{y}_{\ell-1}))$,*

$$\pi_\theta^{(t)}(j \mid \widehat{y}_{\ell-1}, G^L) = \frac{d^{\mathbf{Attn}_L^{(t)} C_B}}{d^{\mathbf{Attn}_L^{(t)} C_B} + (L-1)d^{\frac{1-\mathbf{Attn}_L^{(t)}}{L-1} C_B} + (d-L)} \triangleq p_{L,1}^{(t)}. \tag{23}$$

*(ii) For $j \in \tau\Big(\{g \cdot \widehat{y}_{\ell-1} : g \in G^L, g \neq g_\ell\}\Big)$,*

$$\pi_\theta^{(t)}(j \mid \widehat{y}_{\ell-1}, G^L) = \frac{d^{\frac{1-\mathbf{Attn}_L^{(t)}}{L-1} C_B}}{d^{\mathbf{Attn}_L^{(t)} C_B} + (L-1)d^{\frac{1-\mathbf{Attn}_L^{(t)}}{L-1} C_B} + (d-L)} \triangleq p_{L,2}^{(t)}. \tag{24}$$

*(iii) For any other $j \in \tau(\mathcal{Y})$,*

$$\pi_\theta^{(t)}(j \mid \widehat{y}_{\ell-1}, G^L) = \frac{1}{d^{\mathbf{Attn}_L^{(t)} C_B} + (L-1)d^{\frac{1-\mathbf{Attn}_L^{(t)}}{L-1} C_B} + (d-L)} \triangleq p_{L,3}^{(t)}. \tag{25}$$

*Moreover, $\pi_\theta^{(t)}(j \mid \widehat{y}_{\ell-1}, G^L)$ does not depend on $\ell$. Hence, we suppress the index $\ell$ and write $p_{L,1}^{(t)}$, $p_{L,2}^{(t)}$, and $p_{L,3}^{(t)}$ for brevity.*

**Probabilistic Event.** We conclude this subsection by introducing a probabilistic event that characterizes the potential for *path collisions*, where an incorrect sequence of operations inadvertently leads to the same outcome as the intended compositional path. Such an event serves as a key tool for bounding the interference from distracting trajectories in our subsequent analysis:

$$\mathfrak{E}_L \triangleq \Big\{\exists(\widehat{g}_1, \ldots, \widehat{g}_L), \, \widehat{g}_\ell \in \{g_1, \ldots, g_L\} \text{ for all } \ell \in [L], \text{ s.t. } \widehat{g}_L \circ \cdots \widehat{g}_1(y_0) = y_L\Big\}.$$

**Lemma C.8** (Probability of Trajectory Collision). *Under Assumption 2.1, let $\mathcal{G}$ be a finite non-abelian simple group of order $d$. For a target state $y_L \in \mathcal{Y}$ reached by a specific sequence of $L$ actions, the probability that any alternative sequence of $L$ actions (formed by the same set of available operators) hits $y_L$ is bounded by:*

$$\mathbb{P}(\mathfrak{E}_L) = O\Big(\frac{L^L}{d}\Big). \tag{26}$$

*Proof.* To establish the bound, we consider the total number of possible compositional paths and the collision probability associated with each. In our setting, there are $L$ choices for each of the $L$ steps in a trajectory, resulting in at most $L^L$ possible sequences within the set $\{g_1, \ldots, g_L\}^L$. According to the mixing properties of non-abelian simple groups (Larsen & Shalev, 2007), any sequence that is not algebraically identical to the correct path $(\widehat{g}_L \circ \cdots \circ \widehat{g}_1)$ induces a near-uniform distribution over $\mathcal{Y}$. Consequently, for any single incorrect trajectory, the probability of it hitting the specific target state $y_L$ is $1/d + o(1/d)$. This asymptotic uniformity holds generally across the group, including the case where the target $y_L$ is the initial state $y_0$. By applying the union bound over the collection of all $L^L$ possible paths, we find that the total probability of an accidental collision is at most $L^L \cdot (1/d + o(1/d))$. As the state space size $d \to \infty$, the probability of an accidental activation via an incorrect compositional path becomes negligible, yielding $\mathbb{P}(\mathfrak{E}_L) = O(L^L/d)$. $\qquad\square$

# D. Learning Dynamics of Short-horizon RL

In this section, we focus on the regime $L \leq C_B$. Our analysis tracks the training dynamics of the two scalar quantities $q$ and $r$ defined in Equation (20a) and Equation (20b). We proceed in three steps. First, we state an induction hypothesis that is maintained throughout training. Second, under this hypothesis, we derive one-step update bounds for $q$ and $r$. Finally, we close the induction by showing that the hypothesis holds for all iterations.

We will focus on the RL training dynamics; the same proof structure and bookkeeping apply to SFT training. Accordingly, at the end of this section, we briefly list the key lemmas and the corresponding induction for SFT, and omit the details.

**Induction D.1.** *Given $\Omega(\frac{1}{\text{polylog} d}) < \epsilon < \frac{1}{4}$, and let $T_1$ be the first iteration such that $\text{Attn}_L^{(t)} \geq 1 - \epsilon$. Then for every iteration $t < T_1$, the following statements hold:*

*(a) $O\left(\log \frac{L}{\epsilon}\right) \geq q^{(t)} \geq 0$, and $q^{(t)}$ is monotonically nondecreasing in $t$ (starting from 0);*
*(b) $|r^{(t)}| \leq O(1/d_{\mathsf{p}})q^{(t)}$.*

## D.1. Attention and Logit Preliminaries

We first introduce several properties of the attention scores and logits if Induction D.1 holds.

**Lemma D.1.** *If Induction D.1 holds for all iterations $< t$, then we have*

*(a) $\text{Attn}_L^{(t)} = \dfrac{e^{q^{(t)} - r^{(t)}}}{e^{q^{(t)} - r^{(t)}} + (L-1)} \geq \frac{1}{L}$;*
*(b) $\text{Attn}_{a, \ell-1 \to p, k}^{(t)} = \dfrac{1}{(L-1) + e^{q^{(t)} - r^{(t)}}} = \frac{1}{L-1}\left(1 - \text{Attn}_L^{(t)}\right)$ for $k \neq \ell$.*

Therefore, direct calculations by combining Lemma D.1 and Lemma C.7 yield the following lemma.

**Lemma D.2.** *Assume that Induction D.1 holds for all iterations $< t$. We have*

$$p_{L,1}^{(t)} \geq \Omega(1), \quad 1 - p_{L,1}^{(t)} \geq \Omega\left(\frac{1}{d^{(1-\epsilon)C_B - 1}}\right)$$

*and the following bounds on the transition probabilities $p_{L,2}^{(t)}$ and $p_{L,3}^{(t)}$.*

*(1) **Regime I:** if $\text{Attn}_L^{(t)} < 1 - \frac{L-1}{C_B}$, then*
   *(a) in-context distractor transition*

$$p_{L,2}^{(t)} = \Theta\left(d^{-\left(\text{Attn}_L^{(t)} - \frac{1 - \text{Attn}_L^{(t)}}{L-1}\right)C_B}\right) = O\left(\frac{1}{L}\right)\left(1 - p_{L,1}^{(t)}\right).$$

   *(b) vocabulary distractor transition*

$$p_{L,3}^{(t)} = O\left(d^{-\text{Attn}_L^{(t)}C_B}\right) = O\left(\frac{1}{d^{\frac{C_B}{L-1}(1 - \text{Attn}_L^{(t)})}}\right)\left(1 - p_{L,1}^{(t)}\right).$$

*(2) **Regime II:** if $\text{Attn}_L^{(t)} \geq 1 - \frac{L-1}{C_B}$, then*
   *(a) in-context distractor transition*

$$p_{L,2}^{(t)} = O\left(d^{-\text{Attn}_L^{(t)}C_B}\right) = O\left(\frac{1}{d}\right)\left(1 - p_{L,1}^{(t)}\right).$$

   *(b) vocabulary distractor transition*

$$p_{L,3}^{(t)} = O\left(d^{-\text{Attn}_L^{(t)}C_B}\right) = O\left(\frac{1}{d}\right)\left(1 - p_{L,1}^{(t)}\right).$$

## D.2. Gradient Lemma

Since the initialization is uniform, the initial step-wise probabilities satisfy $p_{L,1}^{(0)} = p_{L,2}^{(0)}$, so the gradient characterization in Proposition E.2 is not directly applicable. We therefore need finer control of the gradients at the very beginning of training.

**Lemma D.3.** *Assume that Induction D.1 holds for all iterations $< t$, when $1 - \mathbf{Attn}_L^{(t)} \geq \Omega(1)$, we have*

$$\nabla_q \widetilde{\mathcal{J}}_L^{(t)} \geq \Omega\Big(\frac{\log d}{d_\mathsf{p} d^{\mathbf{Attn}_L^{(t)} C_B - 1}}\Big).$$

*Proof.* By Equation (21), we have

$$\nabla_q \widetilde{\mathcal{J}}_L^{(t)} = \frac{1}{2Ld_\mathsf{p}} \mathbb{E}_{Z^L} \Big[ \mathfrak{J}^{(t)}(y_0, G^L) \Big]$$

$$= \frac{1}{2Ld_\mathsf{p}} \mathbb{E}_{Z^L} \Bigg[ \sum_{\boldsymbol{v} \in \mathcal{Y}^{L-1} \times \{y_L\}} \Big( \prod_{\ell'=1}^{L} \pi_\theta^{(t)}\big(v_{\ell'} \mid v_{\ell'-1}, G^L\big) \Big) \Big( \sum_{\ell=1}^{L} \cdot \mathbf{Attn}_{a,\ell-1\to p,\ell} \mathfrak{G}_\ell(\boldsymbol{v}) \Big) \Bigg]. \tag{27}$$

Therefore, in the following, we will divide the trajectory $\boldsymbol{v} \in \mathcal{V}^{L-1} \times \{y_L\}$ into three different categories:

- $\boldsymbol{v} = \boldsymbol{v}^* \triangleq (y_1, \cdots, y_L)$, by Lemma D.2, we can obtain

$$\prod_{\ell'=1}^{L} \pi_\theta^{(t)}\big(v_{\ell'} \mid v_{\ell'-1}, G^L\big) = p_{L,1}^{(t)} \geq \Omega(1).$$

Along the trajectory $\boldsymbol{v}^*$, by Lemma C.6, we have

$$\mathfrak{G}_\ell(\boldsymbol{v}) = \sum_{j \in \tau(\mathcal{Y})} \mathcal{E}_j^{(t)} \sum_{r \in [m]} \sigma'\big(\Lambda_{j,r}^{(t)}\big) \cdot \Big( \langle W_{j,r}, Z_{p,\ell} \rangle - \Lambda_{j,r}^{(t)} \Big)$$

$$= \big(1 - p_{L,1}^{(t)}\big) \Big( B - \mathbf{Attn}_L^{(t)}(B + \sigma_0) \Big) - (L-1)p_{L,2}^{(t)} \Big( -B - \mathbf{Attn}_L^{(t)}(B + \sigma_0) \Big)$$

$$\quad - (d - L)p_{L,3}^{(t)} \Big( -B - \sigma_0 \Big)$$

$$\geq \Omega(B) \cdot \big(1 - \mathbf{Attn}_L^{(t)}\big) \cdot \Big( 1 - p_{L,1}^{(t)} \Big). \tag{28}$$

Therefore, we have

$$\mathbb{E}_{Z^L} \Bigg[ \Big( \prod_{\ell'=1}^{L} \pi_\theta^{(t)}\big(y_{\ell'} \mid y_{\ell'-1}, G^L\big) \Big) \Big( \sum_{\ell=1}^{L} \mathbf{Attn}_{a,\ell-1\to p,\ell} \mathfrak{G}_\ell(\boldsymbol{v}^\star) \Big) \Bigg] \geq \Omega\Big( \frac{\log d}{d_\mathsf{p} d^{\mathbf{Attn}_L^{(t)} C_B - 1}} \Big). \tag{29}$$

- On the event $\mathfrak{E}_L$, there exists other trajectories (may more than one) $\boldsymbol{v}' \neq \boldsymbol{v}^*$, s.t., $v'_\ell = g'_\ell(v'_{\ell-1})$ with some $g'_\ell \in G^L$ for all $\ell \in [L]$ (letting $v_0 = y_0$). We denote the collection of such $\boldsymbol{v}'$ as $\mathcal{V}'$. In the following, we denote $g_{i_\ell} = g'_\ell$. For $\boldsymbol{v}' \in \mathcal{V}'$, there will be at least one $\widetilde{\ell}$, s.t., $i_{\widetilde{\ell}} \neq \widetilde{\ell}$. Thus, by Lemma D.2, we have

$$\pi_\theta^{(t)}\big(v'_{\widetilde{\ell}} \mid v'_{\widetilde{\ell}-1}, G^L\big) \leq O(1/L) \cdot \Big( 1 - p_{L,1}^{(t)} \Big).$$

Hence,

$$\prod_{\ell'=1}^{L} \pi_\theta^{(t)}\big(v'_{\ell'} \mid v'_{\ell'-1}, G^L\big) \leq O(1) \cdot \Big( 1 - p_{L,1}^{(t)} \Big) \cdot \prod_{\ell'=1}^{L} \pi_\theta^{(t)}\big(y_{\ell'} \mid y_{\ell'-1}, G^L\big). \tag{30}$$

Moreover, along the trajectory $\boldsymbol{v}'$, the analysis is similar as $\boldsymbol{v}^*$, we have

$$\Bigg| \sum_{j \in \tau(\mathcal{Y})} \mathcal{E}_j^{(t)} \sum_{r \in [m]} \sigma'\big(\Lambda_{j,r}^{(t)}\big) \cdot \Big( \langle W_{j,r}, Z_{p,\ell} \rangle - \Lambda_{j,r}^{(t)} \Big) \Bigg|$$

$$\leq \big(1 - \pi_\theta^{(t)}\big(g_{i_\ell}(\widehat{y}_{\ell-1}) \mid v'_{\ell-1}, G^L\big)\big) \Big( B + \mathbf{Attn}_L^{(t)} B + \sigma_0 \Big)$$

$$\quad + \sum_{\ell' \neq i_\ell} \pi_\theta^{(t)}\big(g_{\ell'}(\widehat{y}_{\ell-1}) \mid v'_{\ell-1}, G^L\big) \Big( B + \mathbf{Attn}_L^{(t)} B + \sigma_0 \Big)$$

$$\quad + \sum_{g \notin G^L} \pi_\theta^{(t)}\big(g(\widehat{y}_{\ell-1}) \mid v'_{\ell-1}, G^L\big) \Big( B + \sigma_0 \Big) \tag{31}$$

$$\leq O(B). \tag{32}$$

Hence, combining Equation (28) Equation (32) and the probability of $\mathfrak{E}_L$ from Lemma C.8 together, it holds that

$$\left| \mathbb{E}_{Z^L} \left[ \sum_{\boldsymbol{v} \in \mathcal{V}'} \left( \prod_{\ell'=1}^{L} \pi_\theta^{(t)}\big(v_{\ell'} \mid v_{\ell'-1}, G^L\big) \right) \left( \sum_{\ell=1}^{L} \cdot \mathbf{Attn}_{a,\ell-1\to p,\ell} \mathfrak{G}_\ell(\boldsymbol{v}) \right) \right] \right|$$

$$\leq O\Big(\frac{L^L}{d}\Big) \mathbb{E}_{Z^L} \left[ |\mathcal{V}'| \left( \prod_{\ell'=1}^{L} \pi_\theta^{(t)}\big(y_{\ell'} \mid y_{\ell'-1}, G^L\big) \right) \left( \sum_{\ell=1}^{L} \cdot \mathbf{Attn}_{a,\ell-1\to p,\ell} \mathfrak{G}_\ell(\boldsymbol{v}^\star) \right) \right]$$

$$\leq O\Big(\frac{1}{d}\Big) \mathbb{E}_{Z^L} \left[ \left( \prod_{\ell'=1}^{L} \pi_\theta^{(t)}\big(y_{\ell'} \mid y_{\ell'-1}, G^L\big) \right) \left( \sum_{\ell=1}^{L} \cdot \mathbf{Attn}_{a,\ell-1\to p,\ell} \mathfrak{G}_\ell(\boldsymbol{v}^\star) \right) \right], \tag{33}$$

where the last inequality follows the fact that $|\mathcal{V}'| \leq L^L = O(1)$.

- for other $\boldsymbol{v} \in \left( \mathcal{Y}^{L-1} \times \{y_L\} \right) \setminus \left( \mathcal{V}' \cup \{\boldsymbol{v}^*\} \right)$, there will be at least one $\widehat{\ell} \in [L]$, s.t., $v_{\widehat{\ell}} = g(v_{\widehat{\ell}-1})$ with $g \neq \widehat{\mathcal{G}}$.

  – if there exists only one such $\widehat{\ell}$, then since the group is simply transitive, there exists another $\widehat{\ell'} \neq \widehat{\ell}$ s.t., $v_{\widehat{\ell'}} = g(v_{\widehat{\ell'}-1})$ with $g \neq g_{\widehat{\ell'}}$. By Lemma D.2, for such $\widehat{\ell}$ and $\widehat{\ell'}$, we have

$$\pi_\theta^{(t)}\big(v_{\widehat{\ell}} \mid v_{\widehat{\ell}-1}, G^L\big) \cdot \pi_\theta^{(t)}\big(v_{\widehat{\ell'}} \mid v_{\widehat{\ell'}-1}, G^L\big)$$

$$\leq O\left( d^{-\mathbf{Attn}_L^{(t)} C_B} \right) \cdot \left( 1 - p_{L,1}^{(t)} \right) = O\left( \frac{1}{d^{1+\Omega(1)}} \right) \left( 1 - p_{L,1}^{(t)} \right). \tag{34}$$

  – if there exists exact $k > 1$ such $\widehat{\ell}$, denoted as $\widehat{\ell}_1, \cdots, \widehat{\ell}_k$. By Lemma D.2, we have

$$\prod_{i=1}^{k} \pi_\theta^{(t)}\big(v_{\widehat{\ell}_i} \mid v_{\widehat{\ell}_i-1}, G^L\big)$$

$$\leq O\left( d^{-(k-1)\mathbf{Attn}_L^{(t)} C_B} \right) \cdot O\left( \frac{1}{d^{\frac{C_B}{L-1}(1-\mathbf{Attn}_L^{(t)})}} \right) \left( 1 - p_{L,1}^{(t)} \right)$$

$$\leq O\left( \frac{1}{d^{k+\Omega(1)}} \right) \left( 1 - p_1^{(t)} \right). \tag{35}$$

Here, the last inequality holds since $\big((k-1)x + \frac{1-x}{L-1}\big)C_B$ is monotonically increase for $x \geq \frac{1}{L}$ and the minimum value is $\frac{kC_B}{L} = k + \Omega(1)$.

Moreover, we have

$$\left| \sum_{j \in \tau(\mathcal{Y})} \mathcal{E}_j^{(t)} \sum_{r \in [m]} \sigma'\big(\Lambda_{j,r}^{(t)}\big) \cdot \left( \langle W_{j,r}, Z_{p,\ell} \rangle - \Lambda_{j,r}^{(t)} \right) \right| \leq O(B), \tag{36}$$

which in turn leads to

$$\left| \mathbb{E}_{Z^L} \left[ \sum_{\boldsymbol{v} \in \left( \mathcal{Y}^{L-1} \times \{y_L\} \right) \setminus \left( \mathcal{V}' \cup \{\boldsymbol{v}^*\} \right)} \left( \prod_{\ell'=1}^{L} \pi_\theta^{(t)}\big(v_{\ell'} \mid v_{\ell'-1}, G^L\big) \right) \left( \sum_{\ell=1}^{L} \cdot \mathbf{Attn}_{a,\ell-1\to p,\ell} \mathfrak{G}_\ell(\boldsymbol{v}) \right) \right] \right|$$

$$\leq \sum_{k=1}^{L} \binom{L}{k} O(N^k) O\Big(\frac{1}{d^{k+\Omega(1)}}\Big) \mathbb{E}_{Z^L} \left[ \left( \prod_{\ell'=1}^{L} \pi_\theta^{(t)}\big(y_{\ell'} \mid y_{\ell'-1}, G^L\big) \right) \left( \sum_{\ell=1}^{L} \cdot \mathbf{Attn}_{a,\ell-1\to p,\ell} \mathfrak{G}_\ell(\boldsymbol{v}^\star) \right) \right]$$

$$\leq O\Big(\frac{1}{d^{\Omega(1)}}\Big) \mathbb{E}_{Z^L} \left[ \left( \prod_{\ell'=1}^{L} \pi_\theta^{(t)}\big(y_{\ell'} \mid y_{\ell'-1}, G^L\big) \right) \left( \sum_{\ell=1}^{L} \cdot \mathbf{Attn}_{a,\ell-1\to p,\ell} \mathfrak{G}_\ell(\boldsymbol{v}^\star) \right) \right]. \tag{37}$$

Therefore, we put Equation (29), Equation (33), Equation (37) together, and thus conclude that

$$\nabla_q \widetilde{\mathcal{J}}_L^{(t)} = \frac{1}{Ld_{\mathsf{p}}} \mathbb{E}_{Z^L} \left[ \mathfrak{J}^{(t)}(y_0, G^L) \right] \geq \Omega\Big( \frac{\log d}{d_{\mathsf{p}} d^{\mathbf{Attn}_L^{(t)} C_B - 1}} \Big).$$

$\square$

**Lemma D.4.** *Assume that Induction D.1 holds for all iterations $< t$, when $\mathbf{Attn}_L^{(t)} \geq 1 - \frac{L-1}{C_B}$, we have*

$$\nabla_q \widetilde{\mathcal{J}}_L^{(t)} = \Theta\left(\frac{\log d}{d_\mathsf{p} d^{\mathbf{Attn}_L^{(t)} C_B - 1}}\right).$$

*Proof.* By Lemma D.2, when $\mathbf{Attn}_L^{(t)} \geq 1 - \frac{L-1}{C_B}$, we have $\Delta_L^{(t)}, p_{L,1}^{(t)} = \Omega(1)$, and $\delta_L^{(t)} \leq p_{L,2}^{(t)} \leq O(1/d)(1 - \Delta_L^{(t)})$. Hence, the condition in Proposition E.2 holds, and we can directly apply it to complete the proof. $\square$

**Lemma D.5.** *Assume that Induction D.1 holds for all iterations $< t$. Then,*

$$\left|\nabla_r \widetilde{\mathcal{J}}_L^{(t)}\right| \leq O\left(\frac{1}{d_\mathsf{p}}\right) \left|\nabla_q \widetilde{\mathcal{J}}_L^{(t)}\right|.$$

*Proof.* The claim follows from Lemma C.3, we thus omit the details. $\square$

### D.3. Proof of Theorem 3.1

By combining the gradient bounds in Lemmas D.3 to D.5, we show that the induction hypothesis Induction D.1 is maintained throughout this stage until $q^{(t)}$ reaches the target scale $\Omega\left(\log(L/\epsilon)\right)$. At that point, we obtain $\mathbf{Attn}_L^{(t)} \geq 1 - \epsilon$, which leads to the following lemma.

**Lemma D.6** (End of Constant-Length Training). *For any $\Omega(1/\mathrm{polylog} d) < \epsilon < \frac{L-1}{C_B}$, the induction hypothesis Induction D.1 holds for all iterations*

$$t < T_1 = O\left(d_\mathsf{p} d^{(1-\epsilon)C_B - 1} \cdot \frac{\log(L/\epsilon)}{\eta \log d}\right).$$

*Moreover, at $t = T_1$ we have: $q^{(T_1)} \geq \Omega(\log(L/\epsilon))$; $\left|r^{(T_1)}\right| \leq O\left(\frac{1}{d}\right) q^{(T_1)}$.*

*Proof.* Assume Induction D.1 holds up to iteration $t$. Then Lemmas D.3 and D.4 imply that the policy gradient for $q$ is strictly positive and satisfies a lower bound of the form

$$\nabla_q \widetilde{\mathcal{J}}_L^{(t)} \geq \Omega\left(\frac{\log d}{d_\mathsf{p}}\right) \cdot d^{-(1-\epsilon)C_B + 1},$$

where we used that along this stage $\mathbf{Attn}_L^{(t)} \leq 1 - \epsilon$. Under policy gradient update with step size $\eta$, we therefore have the per-iteration increase

$$q^{(t+1)} - q^{(t)} = \eta \nabla_q \widetilde{\mathcal{J}}_L^{(t)} \geq \eta \cdot \Omega\left(\frac{\log d}{d_\mathsf{p}}\right) \cdot d^{-(1-\epsilon)C_B + 1}.$$

Summing over iterations until $q^{(t)}$ reaches $\Omega(\log(L/\epsilon))$ yields

$$T_1 = O\left(\frac{d_\mathsf{p}}{\eta \log d} \cdot \log(L/\epsilon) \cdot d^{(1-\epsilon)C_B - 1}\right),$$

as claimed.

Finally, Lemma D.5 gives $\left|\nabla_r \widetilde{\mathcal{J}}_L^{(t)}\right| \leq O(\frac{1}{d})\nabla_q \widetilde{\mathcal{J}}_L^{(t)}$ throughout this stage, and hence $r^{(t)}$ remains slaved to $q^{(t)}$, i.e., $\left|r^{(t)}\right| \leq O(\frac{1}{d})q^{(t)}$ for all $t \leq T_1$. $\square$

Theorem 3.1 follows immediately from Lemma D.6.

## D.4. Proof of Theorem 3.2

The proof follows the same template as the RL case: we (i) set up an induction hypothesis controlling the key parameters, (ii) derive lower/upper bounds on the relevant gradients under this hypothesis, and (iii) combine these bounds to upper bound the time needed for $q^{(t)}$ to reach the target scale, at which point the attention satisfies $\mathbf{Attn}_L^{(t)} \geq 1 - \epsilon$. In other words, Theorem 3.2 is obtained by assembling the lemmas below.

By comparing Lemma C.3 and Lemma C.4, we observe that while the gradient forms are structurally similar, the supervised analysis is more direct. This simplification arises because we only need to track the ground-truth trajectory defined by $\boldsymbol{v} = (y_1, \ldots, y_L)$. Consequently, we can establish an induction hypothesis analogous to Induction D.1.

**Induction D.2.** *For any length* $2 \leq L \leq \mathrm{polylog}d$, *fix any* $\Omega(\frac{1}{\mathrm{polylog}d}) < \epsilon < \min\{\frac{1}{2}(1 - \frac{1}{C_B}), \frac{L-1}{2L}\}$, *and let* $T_1$ *be the first iteration such that* $\mathbf{Attn}_L^{(t)} \geq 1 - \epsilon$. *Then for every iteration* $t < T_1$, *the following statements hold:*

*(a)* $O(\log \frac{L}{\epsilon}) \geq q^{(t)} \geq 0$, *and* $q^{(t)}$ *is monotonically nondecreasing in* $t$ *(starting from* 0*);*
*(b)* $|r^{(t)}| \leq O(1/d_{\mathsf{p}})q^{(t)}$.

Building on this induction, we characterize the gradient dynamics through the following lemmas.

**Lemma D.7.** *Assume that Induction D.2 holds for all iterations* $< t$, *then we have*

- *if* $\mathbf{Attn}_L^{(t)} \leq \frac{1}{C_B}$, *then*

$$-\nabla_q \mathsf{Loss}_L^{(t)} \geq \Omega\Big(\frac{\log d}{d_{\mathsf{p}} L}\Big).$$

- *else, we have*

$$-\nabla_q \mathsf{Loss}_L^{(t)} \geq \Omega\Big(\frac{\epsilon \log d}{d_{\mathsf{p}} d^{(1-\epsilon)C_B - 1}}\Big).$$

*Proof.* The claim follows from a similar analysis as Equation (28) in the proof of Lemma D.3, we thus omit the details. □

**Lemma D.8.** *Assume that Induction D.2 holds for all iterations* $< t$. *Then,*

$$\left|\nabla_r \mathsf{Loss}_L^{(t)}\right| \leq O\left(\frac{1}{d_{\mathsf{p}}}\right)\left|\nabla_q \mathsf{Loss}_L^{(t)}\right|.$$

By combining the results above, we obtain the total training time required to reach the target attention level:

**Lemma D.9** (End of Training). *For any constant length* $2 \leq L \leq \mathrm{polylog}d$, *fix any* $\Omega(\frac{1}{\mathrm{polylog}d}) < \epsilon < \min\{\frac{1}{2}(1 - \frac{1}{C_B}), \frac{L-1}{2L}\}$, *the induction hypothesis Induction D.2 holds for all iterations*

$$t < T_1 = O\left(d_{\mathsf{p}} d^{(1-\epsilon)C_B - 1} \cdot \frac{\log(L/\epsilon)}{\eta \epsilon \log d} + \frac{L d_{\mathsf{p}}}{\eta \log d}\right).$$

*Moreover, at* $t = T_1$ *we have:* $q^{(T_1)} \geq \Omega(\log(L/\epsilon))$; $\left|r^{(T_1)}\right| \leq O\left(\frac{1}{d}\right)q^{(T_1)}$.

Theorem 3.2 follows immediately from Lemma D.9 by noting that $q^{(T_1)} = \Omega(\log(L/\epsilon))$ implies $\mathbf{Attn}_L^{(T_1)} \geq 1 - \epsilon$ (by the definition of $T_1$) and the stated bound on $T_1$ matches the claimed iteration complexity.

## E. Gradient Characterization for General Length

In this section, we use spectral analysis to derive explicit gradient formulas for $q^{(t)}$ and $r^{(t)}$ under the step-invariant constraints in Lemma C.7. For each fixed problem length $L$, these characterizations provide the analytical foundation for our subsequent study of training dynamics across different training settings in later sections.

**Action distribution induced by $\pi_\theta$.** We introduce an action variable $u_\ell \in \mathcal{G}$ as the unique group element applied at step $\ell$. By the simply-transitive action of $\mathcal{G}$ on $\mathcal{Y}$, each transition $\widehat{y}_{\ell-1} \to \widehat{y}_\ell$ corresponds to a unique $u_\ell$ such that $\widehat{y}_\ell = u_\ell(\widehat{y}_{\ell-1})$. With this notation, Lemma C.7 can be equivalently stated as a step-invariant action distribution.

**Lemma E.1.** *Fix a step $\ell$ and condition on $(\widehat{y}_{\ell-1}, G^L)$. Then the policy $\pi_\theta(\cdot \mid \widehat{y}_{\ell-1}, G^L)$ can be partitioned into the following three classes:*

- *target action ($p_{L,1}$): applying the correct in-context rule $g_\ell$,*

$$p_{L,1} := \pi_\theta\left(j \mid \widehat{y}_{\ell-1}, G^L\right), \qquad j = \tau\left(g_\ell(\widehat{y}_{\ell-1})\right). \tag{38a}$$

- *in-context distractor actions ($p_{L,2}$): applying an incorrect rule from the context,*

$$p_{L,2} := \pi_\theta\left(j \mid \widehat{y}_{\ell-1}, G^L\right), \qquad j \in \tau\left(\left\{g(\widehat{y}_{\ell-1}) : g \in G^L, g \neq g_\ell\right\}\right). \tag{38b}$$

- *vocabulary distractor actions ($p_{L,3}$): any other token not corresponding to an in-context transition,*

$$p_{L,3} := \pi_\theta\left(j \mid \widehat{y}_{\ell-1}, G^L\right), \qquad j \in \tau\left(\left\{g(\widehat{y}_{\ell-1}) : g \notin G^L\right\}\right). \tag{38c}$$

### E.1. Trajectory measure induced by $\mathsf{TF}_\theta$

In this part, we formally define the trajectory measure induced by the model $\mathsf{TF}_\theta$ and derive an explicit gradient representation under this probabilistic framework. This formulation allows us to express the optimization objective in terms of conditional (posterior) probabilities over trajectories.

Given a problem instance $(y_0, G^L)$, for any trajectory $\boldsymbol{v} = (v_1, \ldots, v_L)$, we define the trajectory measure $\widetilde{\mathbb{P}}_{\theta,(y_0, G^L)}$ induced by $\mathsf{TF}_\theta$ as

$$\widetilde{\mathbb{P}}_{\theta,(y_0, G^L)}(\boldsymbol{v}) := \prod_{\ell'=1}^{L} \pi_\theta\left(v_{\ell'} \mid v_{\ell'-1}, G^L\right). \tag{39}$$

Under the induced measure Equation (39), we can rewrite the gradient expression in Equation (21) as

$$\mathfrak{J}(y_0, G^L) = \mathbf{Attn}_L \cdot \widetilde{\mathbb{P}}_{\theta,(y_0, G^L)}(v_L = y_L) \cdot \widetilde{\mathbb{E}}_{\theta,(y_0, G^L)}\left[\sum_{\ell=1}^{L} \mathfrak{G}_\ell(\boldsymbol{v}) \,\middle|\, v_L = y_L\right], \tag{40}$$

where

$$\widetilde{\mathbb{P}}_{\theta,(y_0, G^L)}(v_L = y_L) = \sum_{\boldsymbol{v} \in \mathcal{Y}^{L-1} \times \{y_L\}} \widetilde{\mathbb{P}}_{\theta,(y_0, G^L)}(\boldsymbol{v}), \tag{41}$$

$$\widetilde{\mathbb{E}}_{\theta,(y_0, G^L)}\left[\sum_{\ell=1}^{L} \mathfrak{G}_\ell(\boldsymbol{v}) \,\middle|\, v_L = y_L\right] = \frac{\sum_{\boldsymbol{v} \in \mathcal{Y}^{L-1} \times \{y_L\}} \widetilde{\mathbb{P}}_{\theta,(y_0, G^L)}(\boldsymbol{v}) \sum_{\ell=1}^{L} \mathfrak{G}_\ell(\boldsymbol{v})}{\widetilde{\mathbb{P}}_{\theta,(y_0, G^L)}(v_L = y_L)}. \tag{42}$$

We now turn to the term $\mathfrak{G}_\ell$. By Lemma C.6, we can rewrite $\mathfrak{G}_\ell(\boldsymbol{v})$ as

$$
\begin{aligned}
\mathfrak{G}_\ell(\boldsymbol{v}) &= \sum_{j \in \tau(\mathcal{Y})} \mathcal{E}_j \sum_{r \in [m]} \sigma'(\Lambda_{j,r})\left(V_{j,r}(g_\ell) - \Lambda_{j,r}\right) \\
&= \sum_{g \in \mathcal{G}} \mathcal{E}_{\tau(g(\widehat{y}_{\ell-1}))}\left(V_{\tau(g(\widehat{y}_{\ell-1})), r_{g(\widehat{y}_{\ell-1})}}(g_\ell) - \Lambda_{\tau(g(\widehat{y}_{\ell-1})), r_{g(\widehat{y}_{\ell-1})}}\right) \\
&= \mathbb{1}\{g(v_{\ell-1}) = v_\ell\}\left(V_{\tau(g(\widehat{y}_{\ell-1})), r_{g(\widehat{y}_{\ell-1})}}(g_\ell) - \Lambda_{\tau(g(\widehat{y}_{\ell-1})), r_{g(\widehat{y}_{\ell-1})}}\right) \\
&\quad - \sum_{g \in \mathcal{G}} \pi_\theta\left(\tau(g(v_{\ell-1})) \mid v_{\ell-1}, G^L\right)\left(V_{\tau(g(\widehat{y}_{\ell-1})), r_{g(\widehat{y}_{\ell-1})}}(g_\ell) - \Lambda_{\tau(g(\widehat{y}_{\ell-1})), r_{g(\widehat{y}_{\ell-1})}}\right).
\end{aligned}
$$

Using the step-invariant three-way partition in Lemma C.7, the last term further simplifies to

$$\mathfrak{G}_\ell(\boldsymbol{v}) = \mathbb{1}\{g(v_{\ell-1}) = v_\ell\}\Big(V_{\tau(g(\widehat{y}_{\ell-1})), r_{g(\widehat{y}_{\ell-1})}}(g_\ell) - \Lambda_{\tau(g(\widehat{y}_{\ell-1})), r_{g(\widehat{y}_{\ell-1})}}\Big)$$
$$- p_{L,1}\Big((1 - \mathbf{Attn}_L)B - \sigma_0\Big)$$
$$+ (L-1)p_{L,2}\Big((1 + \tfrac{1-\mathbf{Attn}_L}{L-1})B + \sigma_0\Big)$$
$$+ (d-L)p_{L,3}\Big(B + \sigma_0\Big),$$

where we used that the in-context distractor set has size $L-1$ and the remaining vocabulary set has size $d-L$.

Next, define the posterior probabilities (under the trajectory measure conditioned on success)

$$\rho_{\ell,1} \triangleq \widetilde{\mathbb{P}}_{\theta,(y_0, G^L)}\left(u_\ell = g_\ell | v_L = y_L\right), \tag{43}$$
$$\rho_{\ell,2} \triangleq \widetilde{\mathbb{P}}_{\theta,(y_0, G^L)}\left(u_\ell \in G^L \setminus \{g_\ell\} | v_L = y_L\right). \tag{44}$$

Taking the conditional expectation of $\mathfrak{G}_\ell(\boldsymbol{v})$ given $v_L = y_L$ yields

$$\widetilde{\mathbb{E}}_{\theta,(y_0, G^L)}[\mathfrak{G}_\ell(\boldsymbol{v})|v_L = y_L] = \rho_{\ell,1}\Big((1 - \mathbf{Attn}_L)B - \sigma_0\Big) - \rho_{\ell,2}\Big((1 + \tfrac{1-\mathbf{Attn}_L}{L-1})B + \sigma_0\Big)$$
$$- (1 - \rho_{\ell,1} - \rho_{\ell,2})\Big(B + \sigma_0\Big)$$
$$- p_{L,1}\Big((1 - \mathbf{Attn}_L)B - \sigma_0\Big) + (L-1)p_{L,2}\Big((1 + \tfrac{1-\mathbf{Attn}_L}{L-1})B + \sigma_0\Big)$$
$$+ \Big(1 - p_{L,1} - (L-1)p_{L,2}\Big)\Big(B + \sigma_0\Big)$$
$$= \left[(\rho_{\ell,1} - p_{L,1})(2 - \mathbf{Attn}_L) + \Big(p_{L,2} - \frac{\rho_{\ell,2}}{L-1}\Big)(1 - \mathbf{Attn}_L)\right]B.$$

Putting it back to Equation (40), we have

$$\nabla_q \widetilde{\mathcal{J}}_L = \frac{1}{2Ld_{\mathsf{p}}}\mathbb{E}_{Z^L}\left[\mathfrak{J}(y_0, G^L)\right]$$
$$= \frac{1}{2Ld_{\mathsf{p}}}\mathbb{E}_{Z^L}\left[\mathbf{Attn}_L \cdot \widetilde{\mathbb{P}}_{\theta,(y_0, G^L)}(v_L = y_L)B \cdot \sum_\ell \Big((\rho_{\ell,1} - p_{L,1})(2 - \mathbf{Attn}_L) + \Big(p_{L,2} - \frac{\rho_{\ell,2}}{L-1}\Big)(1 - \mathbf{Attn}_L)\Big)\right].$$
$$\tag{45}$$

Intuitively, the gradient is driven by the gap between the posterior action probabilities conditioned on success and their unconditional counterparts. Therefore, controlling the gradient reduces to estimating the posterior probabilities $\rho_{\ell,1}$ and $\rho_{\ell,2}$ as functions of the probability tuple $(p_{L,1}, p_{L,2}, p_{L,3})$. In what follows, we suppress the dependence on $(\theta, (y_0, G^L))$ whenever the context is clear.

### E.2. Preliminaries: Harmonic Analysis on $\mathcal{G}$

The trajectory measure introduced in the previous part involves cumulative products of group actions, which correspond to repeated convolutions of measures on the underlying group. A direct combinatorial analysis of these convolutions is often intractable. To address this, we work in the Fourier domain via the irreducible representations of the group.

In this section, we briefly review the basic facts of harmonic analysis on finite groups and collect the spectral tools we will use to decouple these convolution operations. For background and more detailed treatments, see Serre (1977); Kondor (2008).

**Definition E.1** (Irreducible Representations). Let $\mathcal{G}$ be a finite group of order $|\mathcal{G}| = N$. Let $\Lambda$ denote the set of equivalence classes of irreducible unitary representations of $\mathcal{G}$. For each $\lambda \in \Lambda$, $\lambda : \mathcal{G} \to U(d_\lambda)$ is homomorphism, where $U(d_\lambda)$ is the group of $d_\lambda \times d_\lambda$ unitary matrices.

- The trivial representation is denoted by $\mathbf{1}$, with $d_{\mathbf{1}} = 1$ and $\mathbf{1}(g) = 1, \forall g$.
- Orthogonality Relations: For $\lambda, \eta \in \Lambda$:

$$\langle \lambda_{ij}, \eta_{kl} \rangle = \sum_{g \in \mathcal{G}} \lambda_{ij}(g) \overline{\eta_{kl}(g)} = \frac{N}{d_\lambda} \delta_{\lambda\eta} \delta_{ik} \delta_{jl}$$

Specifically, for $\lambda \neq \mathbf{1}$, $\sum_{g \in \mathcal{G}} \lambda(g) = \mathbf{0}_{d_\lambda \times d_\lambda}$.

**Definition E.2** (Fourier Transform). Let $\Lambda$ be the set of irreducible unitary representations of $\mathcal{G}$. For any $\lambda \in \Lambda$, let $d_\lambda$ be its dimension. For a function $f : \mathcal{G} \to \mathbb{C}$, the Fourier transform is:

$$\widehat{f}(\lambda) \triangleq \sum_{h \in \mathcal{G}} f(h) \lambda(h).$$

**Definition E.3** (Convolution). The convolution of two functions $f, \nu : \mathcal{G} \to \mathbb{C}$ is defined as:

$$(f * \nu)(g) := \sum_{h \in \mathcal{G}} f(gh^{-1}) \nu(h).$$

The Fourier transform maps convolution to matrix multiplication:

$$\widehat{f * \nu}(g) = \widehat{f}(g) \widehat{\nu}(g).$$

**Lemma E.2** (Fourier Inversion Formula & Plancherel Identity). *Function $f : \mathcal{G} \to \mathbb{C}$ can be reconstructed from its Fourier coefficients:*

$$f(g) = \frac{1}{N} \sum_{\lambda \in \Lambda} d_\lambda \mathit{Tr}\left( \widehat{f}(\lambda) \lambda(g)^{-1} \right).$$

*Using the inversion formula at $g = e$ (identity), we have the identity:*

$$\sum_{\lambda \in \Lambda} d_\lambda^2 = N.$$

*We distinguish the trivial representation $\mathbf{1}$ (where $\lambda(h) = 1$) from non-trivial representations $\lambda \neq \mathbf{1}$. Note that $\sum_{\lambda \neq \mathbf{1}} d_\lambda^2 = N - 1$.*

**Definition E.4** (Character Value & Spectral Decay Factor). Let $\mathcal{G}$ be a finite group and let $\Lambda$ denote the set of its irreducible unitary representations. The **character** of a representation $\lambda \in \Lambda$, denoted by $\chi_\lambda : \mathcal{G} \to \mathbb{C}$, is defined as the trace of the linear operator $\lambda(g)$ for each $g \in \mathcal{G}$:

$$\chi_\lambda(g) := \mathrm{Tr}(\lambda(g)).$$

The scalar $\chi_\lambda(g)$ is referred to as the character value of the element $g$ corresponding to $\lambda$. Furthermore, we define the **spectral decay factor**, denoted by $\gamma(\mathcal{G})$, as the maximum normalized character value over all non-trivial representations and non-identity elements:

$$\gamma(\mathcal{G}) := \max_{\substack{\lambda \in \Lambda, \lambda \neq \mathbf{1} \\ g \in \mathcal{G}, g \neq e}} \frac{|\chi_\lambda(g)|}{d_\lambda},$$

where $d_\lambda$ denotes the dimension of the representation $\lambda$.

*Remark* E.1 (Magnitude of $\gamma(\mathcal{G})$). For any finite group $\mathcal{G}$, it holds that $0 \leq \gamma(\mathcal{G}) \leq 1$. Specifically, if $\mathcal{G}$ is abelian or has a non-trivial center $Z(\mathcal{G}) \neq \{e\}$, then $\gamma(\mathcal{G}) = 1$. If $\mathcal{G}$ is a non-abelian simple group, then $\gamma(\mathcal{G}) < 1$. Furthermore, for many sequences of simple groups (e.g., $PSL_2(q)$), $\gamma(\mathcal{G}) \to 0$ as $|\mathcal{G}| \to \infty$, indicating rapid spectral decay.

### E.3. Spectral Decomposition of the Step Measure

With the harmonic analysis framework in place, we translate conditional expectations under the trajectory measure induced by $\mathsf{TF}_\theta$ into convolution equations on the group.

**Reduction to Group Actions.** Since $\mathcal{G}$ acts simply transitively on $\mathcal{Y}$, for any trajectory $v$ with fixed $v_0 = y_0$ there exists a unique sequence of group actions $(u_1, \ldots, u_L) \in \mathcal{G}^L$ such that $v_\ell = u_\ell(v_{\ell-1})$ for $\ell \in [L]$. Consequently,

$$v_L = (u_L \cdots u_1)(y_0).$$

Let the target path be given by $(g_1, \ldots, g_L)$ so that $y_L = (g_L \cdots g_1)(y_0)$, and define the target composition $G_* \triangleq g_L \cdots g_1$. Then the endpoint constraint is equivalent to the group equation

$$v_L = y_L \quad \Longleftrightarrow \quad u_L \cdots u_1 = G_*.$$

Under this representation, the posteriors of interest can be written as

$$\rho_{\ell,1} = \widetilde{\mathbb{P}}\left(u_\ell = g_\ell \middle| u_L \cdots u_1 = G_*\right), \qquad \rho_{\ell,2} = \widetilde{\mathbb{P}}\left(u_\ell \in G^L \setminus \{g_\ell\} \middle| u_L \cdots u_1 = G_*\right).$$

**Definition E.5** (One-step measure). For each step $\ell \in [L]$, define a probability measure $\mu_\ell$ on $\mathcal{G}$ by

$$\mu_\ell(h) \triangleq \widetilde{\mathbb{P}}(u_\ell = h) = \begin{cases} p_{L,1}, & h = g_\ell, \\ p_{L,2}, & h \in G^L \setminus \{g_\ell\}, \\ p_{L,3}, & h \in \mathcal{G} \setminus G^L. \end{cases}$$

Equivalently,

$$\mu_\ell = p_{L,1}\delta_{g_\ell} + p_{L,2}\delta_{G^L \setminus \{g_\ell\}} + p_{L,3}\delta_{\mathcal{G} \setminus G^L},$$

where $\delta_S$ denotes the (unnormalized) uniform measure on a set $S \subseteq \mathcal{G}$. Moreover, the marginal probability of the endpoint is given by

$$\widetilde{\mathbb{P}}(v_L = y_L) = (\mu_L * \cdots * \mu_1)(G_*).$$

We now compute the Fourier transform $\widehat{\mu}_\ell(\lambda)$ for a nontrivial irreducible representation $\lambda \neq \mathbf{1}$.

**Definition E.6** (Spectral objects and effective parameters). Given $\ell \in [L]$, for an irreducible representation $\lambda$ of $\mathcal{G}$, define the sample operator

$$W_\ell(\lambda) \triangleq \sum_{h \in G^L \setminus \{g_\ell\}} \lambda(h).$$

We also define the effective parameters

$$\Delta_L \triangleq p_{L,1} - p_{L,3}, \qquad \delta_L \triangleq p_{L,2} - p_{L,3}, \qquad \sigma_{G^L} \triangleq \max_{\lambda \neq \mathbf{1}} \max_{\ell \in [L]} \|W_\ell(\lambda)\|_{\mathrm{op}},$$

where $\| \cdot \|_{\mathrm{op}}$ denotes the operator norm induced by $\| \cdot \|_2$, i.e., $\|A\|_{\mathrm{op}} := \sup_{\|x\|_2 = 1} \|Ax\|_2$.

**Lemma E.3** (Fourier transform of the one-step measure). *Let $\mu_\ell$ be the one-step measure in Definition E.5. For any nontrivial irreducible representation $\lambda \neq \mathbf{1}$,*

$$\widehat{\mu}_\ell(\lambda) = \Delta_L \lambda(g_\ell) + \delta_L W_\ell(\lambda).$$

*Proof.* By definition of $\mu_\ell$,

$$\widehat{\mu}_\ell(\lambda) = p_{L,1}\lambda(g_\ell) + p_{L,2} \sum_{h \in G^L \setminus \{g_\ell\}} \lambda(h) + p_{L,3} \sum_{h \in \mathcal{G} \setminus G^L} \lambda(h).$$

For $\lambda \neq \mathbf{1}$, by Lemma E.2, we have $\sum_{h \in \mathcal{G}} \lambda(h) = 0$, hence $\sum_{h \in \mathcal{G} \setminus G^L} \lambda(h) = -(\lambda(g_\ell) + W_\ell(\lambda))$. Substituting and collecting terms yields the claim. □

### E.4. Posterior Estimation

Building on the operator decomposition, we compute the posterior probabilities by evaluating traces of the resulting spectral products. For notational convenience, we define the events

$$E \triangleq \{v_L = y_L\}, \qquad A_\ell \triangleq \{u_\ell = g_\ell\}, \qquad B_\ell \triangleq \{u_\ell \in G^L \setminus \{g_\ell\}\}.$$

Thus, the main task is to control $\widetilde{\mathbb{P}}(E)$ as well as the joint probabilities $\widetilde{\mathbb{P}}(A_\ell \cap E)$ and $\widetilde{\mathbb{P}}(B_\ell \cap E)$.

**Lemma E.4.** *With $E, A_\ell, B_\ell$ defined above, we have the expansions*

$$\widetilde{\mathbb{P}}(E) = \frac{1}{d} + \left(1 - \frac{1}{d}\right)\Delta_L^L + \mathcal{R}_E, \tag{46a}$$

$$\widetilde{\mathbb{P}}(A_\ell \cap E) = \frac{p_{L,1}}{d} + \left(1 - \frac{1}{d}\right)p_{L,1}\Delta_L^{L-1} + \mathcal{R}_A, \tag{46b}$$

$$\widetilde{\mathbb{P}}(B_\ell \cap E) = \frac{(L-1)p_{L,2}}{d} + \mathcal{R}_B. \tag{46c}$$

*Moreover, the remainders satisfy*

$$|\mathcal{R}_E| \le \left(1 - \frac{1}{d}\right)\left[(\Delta_L + \sigma_{G^L}\delta_L)^L - \Delta_L^L - L\sigma_{G^L}\delta_L\Delta_L^{L-1} + (L-1)L\gamma(\mathcal{G})\delta_L\Delta_L^{L-1}\right], \tag{47a}$$

$$|\mathcal{R}_A| \le p_{L,1}\left(1 - \frac{1}{d}\right)\left[(\Delta_L + \sigma_{G^L}\delta_L)^{L-1} - \Delta_L^{L-1} - (L-1)\sigma_{G^L}\delta_L\Delta_L^{L-2} + (L-1)^2\gamma(\mathcal{G})\delta_L\Delta_L^{L-2}\right], \tag{47b}$$

$$|\mathcal{R}_B| \le p_{L,2}\left(1 - \frac{1}{d}\right)\left[\sigma_{G^L}\left((\Delta_L + \sigma_{G^L}\delta_L)^{L-1} - \Delta_L^{L-1}\right) + (L-1)\gamma(\mathcal{G})\Delta_L^{L-1}\right]. \tag{47c}$$

*Proof.* By Lemma E.2 and the convolution theorem, each quantity $\widetilde{\mathbb{P}}(\cdot)$ can be written as a sum of traces of products of Fourier operators. We isolate the trivial-representation contribution and bound the remaining terms using the decomposition $\widehat{\mu}_k(\lambda) = \Delta_L\lambda(g_k) + \delta_L W_k(\lambda)$. We spell out the details for $\widetilde{\mathbb{P}}(E)$; the bounds for $\widetilde{\mathbb{P}}(A_\ell \cap E)$ and $\widetilde{\mathbb{P}}(B_\ell \cap E)$ follow analogously.

**Estimation of $\widetilde{\mathbb{P}}(E)$.** Let us start with $\widetilde{\mathbb{P}}(E)$. By Lemma E.2, we have

$$\widetilde{\mathbb{P}}(E) = (\mu_L * \cdots * \mu_1)(G_*)$$

$$= \frac{1}{d}\sum_{\lambda \in \Lambda} d_\lambda \mathrm{Tr}\left(\widehat{\mu_L * \cdots * \mu_1}(\lambda)\lambda(G_*)^{-1}\right)$$

$$= \frac{1}{d}\sum_{\lambda \in \Lambda} d_\lambda \mathrm{Tr}\left(\underbrace{\left[\prod_{k=L}^{1}\widehat{\mu}_k(\lambda)\right]}_{=:\Pi(\lambda)}\lambda(G_*)^{-1}\right).$$

- For $\lambda = \mathbf{1}$: $\widehat{\mu}_k(\mathbf{1}) = 1$. Hence $\Pi(\lambda) = 1$, and we can obtain

$$\frac{1}{d}d_\lambda \mathrm{Tr}\left(\Pi(\lambda)\lambda(G_*)^{-1}\right) = \frac{1}{d}d_\mathbf{1} \cdot d_\mathbf{1} = \frac{1}{d}.$$

- For $\lambda \ne \mathbf{1}$: by the decomposition $\widehat{\mu}_k(\lambda) = \Delta_L\lambda(g_k) + \delta_L W_k(\lambda)$ from Lemma E.3, we have

$$\Pi(\lambda) = \prod_{k=L}^{1}\left(\Delta_L\lambda(g_k) + \delta_L W_k(\lambda)\right) = \Delta_L^L\prod_{k=L}^{1}\lambda(g_k) + T_{\mathrm{res}}(\lambda).$$

Then the trace contribution by the first term is:

$$\mathrm{Tr}\left(\Delta_L^L\left(\prod_{k=L}^{1}\lambda(g_k)\right)\lambda(G_*)^{-1}\right) = (\Delta_L)^L\mathrm{Tr}(\lambda(\prod_{k=L}^{1}g_k)\lambda(G_*)^{-1}) = d_\lambda(\Delta_L)^L.$$

Summing this over all $\lambda \neq \mathbf{1}$, we obtain

$$\frac{1}{d}\sum_{\lambda \neq \mathbf{1}} d_\lambda \mathrm{Tr}\left(\Delta_L^L \left(\prod_{k=L}^{1} \lambda(g_k)\right)\lambda(G_*)^{-1}\right) = \frac{1}{d}\sum_{\lambda \neq \mathbf{1}} d_\lambda \cdot d_\lambda (\Delta_L)^L$$
$$= \left(1 - \frac{1}{d}\right)(\Delta_L)^L,$$

where the last equality holds since $\sum_{\lambda \neq \mathbf{1}} d_\lambda^2 = d - 1$ by Lemma E.2. Therefore, it suffices to control the operator norm of the residual term $T_{\mathrm{res}}(\lambda)$. Notice that $T_{\mathrm{res}}(\lambda)$ can be expanded into $2^L - 1$ terms, each of the form $M_L \cdots M_1$, where for each $k$,

$$M_k \in \{\Delta_L \lambda(g_k), \delta_L W_k(\lambda)\},$$

and at least one factor $M_k$ equals $\delta_L W_k(\lambda)$. We further decompose $T_{\mathrm{res}}(\lambda)$ into two parts, $T_{\mathrm{res}}(\lambda) = T_{\mathrm{res},1}(\lambda) + T_{\mathrm{res},2}(\lambda)$:

– $T_{\mathrm{res},1}(\lambda)$ consists of the terms for which there exists a unique $k^* \in [L]$ such that $M_{k^*} = \delta_L W_{k^*}(\lambda)$. In this case,

$$\left|\mathrm{Tr}\left(\delta_L \Delta_L^{L-1}\left(\prod_{k=L}^{k^*+1} \lambda(g_k)\right)W_{k^*}(\lambda)\left(\prod_{k=k^*-1}^{1} \lambda(g_k)\right)\lambda(G_*)^{-1}\right)\right|$$
$$= \left|\delta_L \Delta_L^{L-1}\mathrm{Tr}(W_{k^*}(\lambda)\lambda(g_{k^*})^{-1})\right|$$
$$= \left|\delta_L \Delta_L^{L-1}\sum_{g \in G^L \setminus \{g_{k^*}\}} \mathrm{Tr}(\lambda(gg_{k^*}^{-1}))\right|$$
$$= \delta_L \Delta_L^{L-1}\sum_{g \in G^L \setminus \{g_{k^*}\}} \left|\chi_\lambda(gg_{k^*}^{-1})\right|$$
$$\leq \delta_L \Delta_L^{L-1}(L-1) \cdot d_\lambda \gamma(\mathcal{G}), \tag{48}$$

where the last inequality uses $gg_{k^*}^{-1} \neq e$ (here $e$ denotes the identity element of $\mathcal{G}$) and the definition of $\gamma(\mathcal{G})$. Since there are $L$ such terms in $T_{\mathrm{res},1}(\lambda)$, we obtain

$$\left|\frac{1}{d}\sum_{\lambda \neq \mathbf{1}} d_\lambda \mathrm{Tr}\left(T_{\mathrm{res},1}(\lambda)\lambda(G_*)^{-1}\right)\right| \leq \left(1 - \frac{1}{d}\right)\delta_L \Delta_L^{L-1}(L-1)L\gamma(\mathcal{G}).$$

– $T_{\mathrm{res},2}(\lambda)$ collects the remaining $2^L - 1 - L$ terms, i.e., those for which at least two factors $M_k$ equal $\delta_L W_k(\lambda)$. Then

$$\|T_{\mathrm{res},2}(\lambda)\|_{\mathrm{op}} = \left\|\sum_{M_k \in \{\Delta_L \lambda(g_k), \delta_L W_k(\lambda)\}, \sum_{k=1}^{L} \mathbb{1}_{M_k = \delta_L W_k(\lambda)} \geq 2} M_L \cdot, \cdots, \cdot M_1\right\|_{\mathrm{op}}$$
$$\leq \sum_{M_k \in \{\Delta_L \lambda(g_k), \delta_L W_k(\lambda)\}, \sum_{k=1}^{L} \mathbb{1}_{M_k = \delta_L W_k(\lambda)} \geq 2} \prod_{k=L}^{1} \|M_k\|_{\mathrm{op}}$$
$$\overset{(a)}{\leq} \sum_{i=2}^{L}\binom{L}{i}\Delta_L^{L-i}(\sigma_{G^L}\delta_L)^i = (\Delta_L + \sigma_{G^L}\delta_L)^L - \Delta_L^L - L\sigma_{G^L}\delta_L \Delta_L^{L-1},$$

Here $(a)$ holds since $\|\Delta_L \lambda(g_k)\|_{\mathrm{op}} \leq \Delta_L$, $\|\delta_L W_k(\lambda)\|_{\mathrm{op}} \leq \sigma_{G^L}\delta_L$, and there are exactly $\binom{L}{i}$ choices of indices for which $i$ different $M_k$'s equal $\delta_L W_k(\lambda)$. Consequently,

$$\left|\frac{1}{d}\sum_{\lambda \neq \mathbf{1}} d_\lambda \mathrm{Tr}\left(T_{\mathrm{res},2}(\lambda)\lambda(G_*)^{-1}\right)\right| \leq \frac{1}{d}\sum_{\lambda \neq \mathbf{1}} d_\lambda \cdot d_\lambda \|T_{\mathrm{res},2}(\lambda)\|_{\mathrm{op}}\|\lambda(G_*)^{-1}\|_{\mathrm{op}}$$
$$= \left(1 - \frac{1}{d}\right)\left((\Delta_L + \sigma_{G^L}\delta_L)^L - \Delta_L^L - L\sigma_{G^L}\delta_L \Delta_L^{L-1}\right).$$

Putting everything together, we obtain

$$\widetilde{\mathbb{P}}(E) = \frac{1}{d} + \left(1 - \frac{1}{d}\right)\Delta_L^L + \mathcal{R}_E,$$

where the remainder term $\mathcal{R}_E$ satisfies

$$|\mathcal{R}_E| \le \left(1 - \frac{1}{d}\right)\left((\Delta_L + \sigma_{G^L}\delta_L)^L - \Delta_L^L - L\sigma_{G^L}\delta_L\Delta_L^{L-1} + \delta_L\Delta_L^{L-1}(L-1)L\gamma(\mathcal{G})\right).$$

**Estimation of $\widetilde{\mathbb{P}}(A_\ell \cap E)$.** The analysis is similar to that of $\widetilde{\mathbb{P}}(E)$. The key difference is that we replace the measure at step $\ell$ by the Dirac measure $p_{L,1}\delta_{g_\ell}$, since the $\ell$-th step takes the action $g_\ell$. Correspondingly, its Fourier transform becomes $p_{L,1}\lambda(g_\ell)$. Hence,

$$\widetilde{\mathbb{P}}(A_\ell \cap E) = (\mu_L * \cdots p_{L,1}\delta_{g_\ell} \cdots * \mu_1)(G_*)$$
$$= \frac{1}{d}\sum_{\lambda \in \Lambda} d_\lambda \mathrm{Tr}\left(\underbrace{[\widehat{\mu}_L(\lambda)\cdots p_{L,1}\lambda(g_\ell)\cdots\widehat{\mu}_1(\lambda)]}_{=:\Pi_{A_\ell}(\lambda)}\lambda(G_*)^{-1}\right).$$

- For $\lambda = \mathbf{1}$: since $\widehat{\mu}_k(\mathbf{1}) = 1$, we have $\Pi_{A_\ell}(\mathbf{1}) = p_{L,1}$ and thus

$$\frac{1}{d}d_\lambda \mathrm{Tr}\left(\Pi_{A_\ell}(\lambda)\lambda(G_*)^{-1}\right) = \frac{p_{L,1}}{d}.$$

- For $\lambda \ne \mathbf{1}$: we can write

$$\Pi_{A_\ell}(\lambda) = \left(\prod_{k=L}^{\ell+1}\left(\Delta_L\lambda(g_k) + \delta_L W_k(\lambda)\right)\right)\left(p_{L,1}\lambda(g_\ell)\right)\left(\prod_{k=\ell-1}^{1}\left(\Delta_L\lambda(g_k) + \delta_L W_k(\lambda)\right)\right)$$
$$= p_{L,1}\Delta_L^{L-1}\left(\prod_{k=L}^{1}\lambda(g_k)\right)\lambda(g_\ell)\left(\prod_{k=L}^{1}\lambda(g_k)\right) + T_{\mathrm{res},A_\ell}(\lambda).$$

The trace contribution of the leading term is

$$\mathrm{Tr}\left(p_{L,1}\Delta_L^{L-1}\left(\prod_{k=L}^{1}\lambda(g_k)\right)\lambda(G_*)^{-1}\right) = p_{L,1}(\Delta_L)^{L-1}\mathrm{Tr}(\lambda(\prod_{k=L}^{1}g_k)\lambda(G_*)^{-1})$$
$$= d_\lambda p_{L,1}(\Delta_L)^{L-1}.$$

Summing over all $\lambda \ne \mathbf{1}$ yields

$$\frac{1}{d}\sum_{\lambda \ne \mathbf{1}} d_\lambda \mathrm{Tr}\left(p_{L,1}(\Delta_L)^{L-1}\left(\prod_{k=L}^{1}\lambda(g_k)\right)\lambda(G_*)^{-1}\right) = \frac{1}{d}\sum_{\lambda \ne \mathbf{1}} d_\lambda \cdot d_\lambda p_{L,1}(\Delta_L)^{L-1}$$
$$= \left(1 - \frac{1}{d}\right)p_{L,1}(\Delta_L)^{L-1}.$$

The residual term $T_{\mathrm{res},A_\ell}(\lambda)$ can be controlled exactly as in the analysis of $T_{\mathrm{res}}(\lambda)$, which gives

$$\left|\frac{1}{d}\sum_{\lambda \ne \mathbf{1}} d_\lambda \mathrm{Tr}\left(T_{\mathrm{res},A_\ell}(\lambda)\lambda(G_*)^{-1}\right)\right|$$
$$\le p_{L,1}\left(1 - \frac{1}{d}\right)\left((\Delta_L + \sigma_{G^L}\delta_L)^{L-1} - \Delta_L^{L-1} - (L-1)\sigma_{G^L}\delta_L\Delta_L^{L-2} + \delta_L\Delta_L^{L-2}(L-1)^2\gamma(\mathcal{G})\right).$$

Putting the above bounds together, we conclude that

$$\widetilde{\mathbb{P}}(A_\ell \cap E) = \frac{p_{L,1}}{d} + \left(1 - \frac{1}{d}\right)p_{L,1}\Delta_L^{L-1} + \mathcal{R}_A,$$

where

$$|\mathcal{R}_A| \le p_{L,1}\left(1 - \frac{1}{d}\right)\left((\Delta_L + \sigma_{G^L}\delta_L)^{L-1} - \Delta_L^{L-1} - (L-1)\sigma_{G^L}\delta_L\Delta_L^{L-2} + \delta_L\Delta_L^{L-2}(L-1)^2\gamma(\mathcal{G})\right).$$

**Estimation of $\widetilde{\mathbb{P}}(B_\ell \cap E)$.** For $B_\ell \cap E$, at step $\ell$ we use the measure $p_{L,2}\delta_{G^L \setminus \{g_\ell\}}$, whose Fourier operator is

$$\widehat{\mu}_{B_\ell}(\lambda) = p_{L,2} \sum_{g \in G^L \setminus \{g_\ell\}} \lambda(g) = p_{L,2}W_\ell(\lambda).$$

Hence,

$$\widetilde{\mathbb{P}}(B_\ell \cap E) = (\mu_L * \cdots p_{L,2}\delta_{G^L \setminus \{g_\ell\}} \cdots * \mu_1)(G_*)$$
$$= \frac{1}{d} \sum_{\lambda \in \Lambda} d_\lambda \mathrm{Tr}\Big( \underbrace{[\widehat{\mu}_L(\lambda) \cdots \widehat{\mu}_{B_\ell}(\lambda) \cdots \widehat{\mu}_1(\lambda)]}_{=:\Pi_{B_\ell}(\lambda)} \lambda(G_*)^{-1} \Big).$$

- For $\lambda = \mathbf{1}$: since $\widehat{\mu}_k(\mathbf{1}) = 1$, we have $\Pi_{B_\ell}(\mathbf{1}) = p_{L,2}(L-1)$, and thus

$$\frac{1}{d}d_\lambda \mathrm{Tr}\left(\Pi_{B_\ell}(\lambda)\lambda(G_*)^{-1}\right) = \frac{(L-1) \cdot p_{L,2}}{d}.$$

- For $\lambda \neq \mathbf{1}$: analogous to the decomposition of $T_{\mathrm{res}}(\lambda)$, the operator $\Pi_{B_\ell}(\lambda)$ can be expanded into $2^{L-1}$ terms of the form $M_L \cdots (p_{L,2}W_\ell(\lambda)) \cdots M_1$, where, for each $k \neq \ell$,

$$M_k \in \{\Delta_L \lambda(g_k), \delta_L W_k(\lambda)\}.$$

We further split $\Pi_{B_\ell}(\lambda)$ into two parts,

$$\Pi_{B_\ell}(\lambda) = T_{B_\ell,1}(\lambda) + T_{B_\ell,2}(\lambda).$$

- $T_{B_\ell,1}$ consists of the unique term for which $M_k = \Delta_L \lambda(g_k)$ for all $k \neq \ell$. In this case,

$$\left| \mathrm{Tr}\left( \Delta_L^{L-1} \Big( \prod_{k=L}^{k^*+1} \lambda(g_k) \Big) p_{L,2}W_\ell(\lambda) \Big( \prod_{k=k^*-1}^{1} \lambda(g_k) \Big) \lambda(G_*)^{-1} \right) \right|$$
$$\leq p_{L,2}\Delta_L^{L-1}(L-1) \cdot d_\lambda \gamma(\mathcal{G}),$$

where the inequality follows by an argument analogous to Equation (48). Therefore, we have:

$$\left| \frac{1}{d} \sum_{\lambda \neq \mathbf{1}} d_\lambda \mathrm{Tr}\left( T_{B_\ell,1}(\lambda)\lambda(G_*)^{-1} \right) \right| \leq \Big(1 - \frac{1}{d}\Big) p_{L,2}\Delta_L^{L-1}(L-1)\gamma(\mathcal{G}).$$

- $T_{B_\ell,2}$ collects the remaining terms, i.e., those for which at least one index $k \neq \ell$ satisfies $M_k = \delta_L W_k(\lambda)$. Then

$$\|T_{\mathrm{res},2}(\lambda)\|_{\mathrm{op}} = \left\| \sum_{M_k \in \{\Delta_L \lambda(g_k), \delta_L W_k(\lambda)\} \text{ for } k \neq \ell, \sum_{k \neq \ell} \mathbb{1}_{M_k = \delta_L W_k(\lambda)} \geq 1} M_L \cdots p_{L,2}W_\ell(\lambda) \cdots M_1 \right\|_{\mathrm{op}}$$
$$\leq p_{L,2}\sigma_{G^L} \sum_{i=1}^{L} \binom{L}{i} \Delta_L^{L-i}(\sigma_{G^L}\delta_L)^i$$
$$= p_{L,2}\sigma_{G^L} \Big( (\Delta_L + \sigma_{G^L}\delta_L)^{L-1} - \Delta_L^{L-1} \Big),$$

which can be shown by the same argument as in the bound for $T_{\mathrm{res},2}$. Consequently,

$$\left| \frac{1}{d} \sum_{\lambda \neq \mathbf{1}} d_\lambda \mathrm{Tr}\left( T_{B_\ell,2}(\lambda)\lambda(G_*)^{-1} \right) \right| \leq \frac{1}{d} \sum_{\lambda \neq \mathbf{1}} d_\lambda \cdot d_\lambda \|T_{B_\ell,2}\|_{\mathrm{op}} \|\lambda(G_*)^{-1}\|_{\mathrm{op}}$$
$$= \Big(1 - \frac{1}{d}\Big) p_{L,2}\sigma_{G^L} \Big( (\Delta_L + \sigma_{G^L}\delta_L)^{L-1} - \Delta_L^{L-1} \Big).$$

Putting everything together, we obtain

$$\widetilde{\mathbb{P}}(B_\ell \cap E) = \frac{(L-1)p_{L,2}}{d} + \mathcal{R}_B,$$

where the remainder term $\mathcal{R}_B$ satisfies

$$|\mathcal{R}_B| \le \left(1 - \frac{1}{d}\right)p_{L,2}\left(\sigma_{G^L}\left((\Delta_L + \sigma_{G^L}\delta_L)^{L-1} - \Delta_L^{L-1}\right) + \Delta_L^{L-1}(L-1)\gamma(\mathcal{G})\right).$$

$\square$

The expansions in Lemma E.4 immediately imply the following deviations of the posterior probabilities $\rho_{\ell,1}$ and $\rho_{\ell,2}$ from their corresponding priors.

**Proposition E.1** (Posterior deviation and dominant term). *The posterior deviations admit the exact identities*

$$\rho_{\ell,1} - p_{L,1} = \frac{p_{L,1}\Delta_L^{L-1}(1 - \Delta_L)\left(1 - \frac{1}{d}\right) + \mathcal{R}_A - p_{L,1}\mathcal{R}_E}{\widetilde{\mathbb{P}}(E)}, \tag{49a}$$

$$p_{L,2} - \frac{\rho_{\ell,2}}{L-1} = \frac{p_{L,2}\Delta_L^L\left(1 - \frac{1}{d}\right) + p_{L,2}\mathcal{R}_E - \frac{\mathcal{R}_B}{L-1}}{\widetilde{\mathbb{P}}(E)}. \tag{49b}$$

*Moreover, if*

$$\frac{\sigma_{G^L}\delta_L}{\Delta_L} \ll \frac{1}{L}, \tag{50}$$

*then the remainder terms satisfy*

$$|\mathcal{R}_A| \le p_{L,1}\Delta_L^{L-1}\left(1 - \frac{1}{d}\right)\left(O\left(\frac{\sigma_{G^L}^2\delta_L^2}{\Delta_L^2}\right) + \frac{(L-1)^2\gamma(\mathcal{G})\delta_L}{\Delta_L}\right), \tag{51a}$$

$$|\mathcal{R}_E| \le \Delta_L^L\left(1 - \frac{1}{d}\right)\left(O\left(\frac{\sigma_{G^L}^2\delta_L^2}{\Delta_L^2}\right) + \frac{(L-1)L\gamma(\mathcal{G})\delta_L}{\Delta_L}\right), \tag{51b}$$

$$\frac{|\mathcal{R}_B|}{L-1} \le p_{L,2}\Delta_L^{L-1}\left(1 - \frac{1}{d}\right)\left(O\left(\frac{\sigma_{G^L}^2\delta_L}{(L-1)\Delta_L}\right) + \gamma(\mathcal{G})\right). \tag{51c}$$

*Proof.* For $\rho_{\ell,1} - p_{L,1}$, by Lemma E.4 we have

$$\rho_{\ell,1} - p_{L,1} = \frac{\widetilde{\mathbb{P}}(A_\ell \cap E) - p_{L,1}\widetilde{\mathbb{P}}(E)}{\widetilde{\mathbb{P}}(E)}$$

$$= \frac{\frac{p_{L,1}}{d} + p_{L,1}\Delta_L^{L-1}\left(1 - \frac{1}{d}\right) + \mathcal{R}_A - p_{L,1}\left(\frac{1}{d} + \Delta_L^L\left(1 - \frac{1}{d}\right) + \mathcal{R}_E\right)}{\widetilde{\mathbb{P}}(E)}$$

$$= \frac{p_{L,1}\Delta_L^{L-1}(1 - \Delta_L)\left(1 - \frac{1}{d}\right) + \mathcal{R}_A - p_{L,1}\mathcal{R}_E}{\widetilde{\mathbb{P}}(E)},$$

which gives Equation (49a). For $p_{L,2} - \frac{\rho_{\ell,2}}{L-1}$, we similarly obtain

$$p_{L,2} - \frac{\rho_{\ell,2}}{L-1} = \frac{p_{L,2}(L-1)\widetilde{\mathbb{P}}(E) - \widetilde{\mathbb{P}}(B_\ell \cap E)}{(L-1)\widetilde{\mathbb{P}}(E)}$$

$$= \frac{p_{L,2}(L-1)\left(\frac{1}{d} + \Delta_L^L\left(1 - \frac{1}{d}\right) + \mathcal{R}_E\right) - \frac{(L-1)p_{L,2}}{d} - \mathcal{R}_B}{(L-1)\widetilde{\mathbb{P}}(E)}$$

$$= \frac{p_{L,2}\Delta_L^L\left(1 - \frac{1}{d}\right) + p_{L,2}\mathcal{R}_E - \mathcal{R}_B/(L-1)}{\widetilde{\mathbb{P}}(E)},$$

which gives Equation (49b).

It remains to bound $\mathcal{R}_E$, $\mathcal{R}_A$, and $\mathcal{R}_B$ under Equation (50). For notational simplicity, let $x \triangleq \sigma_{G^L}\delta_L/\Delta_L$. Then

$$(\Delta_L + \sigma_{G^L}\delta_L)^k = \Delta_L^k(1 + x)^k.$$

**Bounds for $\mathcal{R}_E$ and $\mathcal{R}_A$.** The expressions in the brackets for $\mathcal{R}_E$ and $\mathcal{R}_A$ contain

$$(1 + x)^k - 1 - kx, \qquad k \in \{L, L-1\}.$$

Under $x \ll 1/L$, the second-order Taylor remainder gives $(1 + x)^k - 1 - kx = O(k^2x^2)$, which implies

$$(\Delta_L + \sigma_{G^L}\delta_L)^k - \Delta_L^k - k\sigma_{G^L}\delta_L\Delta_L^{k-1} = \Delta_L^k \cdot O(k^2x^2) = O\big(k^2\sigma_{G^L}^2\delta_L^2\Delta_L^{k-2}\big).$$

Substituting this estimate into the displayed bounds for $\mathcal{R}_E$ and $\mathcal{R}_A$ yields the claimed controls for $|\mathcal{R}_E|$ and $|\mathcal{R}_A|$.

**Bound for $\mathcal{R}_B$.** Here the bracket contains $(\Delta_L + \sigma_{G^L}\delta_L)^{L-1} - \Delta_L^{L-1} = \Delta_L^{L-1}\big((1 + x)^{L-1} - 1\big)$. Under $x \ll 1/L$, the first-order estimate gives $(1 + x)^{L-1} - 1 = O((L - 1)x)$, hence

$$\sigma_{G^L}\big((\Delta_L + \sigma_{G^L}\delta_L)^{L-1} - \Delta_L^{L-1}\big) = \sigma_{G^L}\Delta_L^{L-1} \cdot O((L-1)\sigma_{G^L}\delta_L/\Delta_L) = O\big((L-1)\sigma_{G^L}^2\delta_L\Delta_L^{L-2}\big).$$

Plugging this into the displayed bound for $\mathcal{R}_B$ yields the stated control on $|\mathcal{R}_B|$.

$\square$

### E.5. Gradient Characterization: Proof of Lemma 5.2

Combining the posterior deviations in Proposition E.1 with Equation (45), we obtain the following characterization of the gradient, which is a formal version of Lemma 5.2.

**Proposition E.2** (Gradient characterization). *Given problem length L, suppose that*

$$\frac{L^2\delta_L}{\Delta_L} = o(1) \cdot (1 - \Delta_L) \qquad and \qquad \frac{p_{L,2}}{p_{L,1}} = o(1) \cdot (1 - \Delta_L).$$

*Then*

$$\nabla_q\widetilde{\mathcal{J}}_L = \Theta(\log d/d_{\mathsf{p}}) \cdot p_{L,1}\Delta_L^{L-1}\Big(1 - \frac{1}{d}\Big)(1 - \Delta_L),$$

$$|\nabla_r\widetilde{\mathcal{J}}_L| = O(1/d_{\mathsf{p}}) \cdot \nabla_q\widetilde{\mathcal{J}}_L.$$

*Proof.* Recall that

$$\nabla_q\widetilde{\mathcal{J}}_L = \frac{1}{2Ld_{\mathsf{p}}}\mathbb{E}_{Z^L}\Big[\mathbf{Attn}_L \cdot B \cdot \widetilde{\mathbb{P}}(E)\underbrace{\sum_{\ell=1}^{L}\Big((\rho_{\ell,1} - p_{L,1})(2 - \mathbf{Attn}_L) + \Big(p_{L,2} - \frac{\rho_{\ell,2}}{L-1}\Big)(1 - \mathbf{Attn}_L)\Big)}_{=:J_{\text{gap}}}\Big]. \tag{52}$$

Note that $2 - \mathbf{Attn}_L \geq 1$ and $1 - \mathbf{Attn}_L \in (0, 1)$. Under the stated assumptions, since $\sigma_{G^L} \leq L - 1$, the condition $\sigma_{G^L}\delta_L/\Delta_L \ll 1/L$ holds. Hence Proposition E.1 applies. We now verify that for $J_{\text{gap}}$, all remainder contributions are negligible compared to the leading term $p_{L,1}\Delta_L^{L-1}\big(1 - \frac{1}{d}\big)(1 - \Delta_L)$.

• **Bounding $\mathcal{R}_A$.** By Equation (51a) in Proposition E.1,

$$|\mathcal{R}_A| \leq p_{L,1}\Delta_L^{L-1}\Big(1 - \frac{1}{d}\Big)\left(O\Big(\frac{\sigma_{G^L}^2\delta_L^2}{\Delta_L^2}\Big) + \frac{(L-1)^2\gamma(\mathcal{G})\delta_L}{\Delta_L}\right).$$

Using $\sigma_{G^L} \leq L - 1$ and $\frac{L^2\delta_L}{\Delta_L} = o(1) \cdot (1 - \Delta_L)$, we have

$$\frac{\sigma_{G^L}^2\delta_L^2}{\Delta_L^2} \leq \frac{L^2\delta_L}{\Delta_L} \cdot \frac{\delta_L}{\Delta_L} = o(1) \cdot (1 - \Delta_L) \cdot \frac{\delta_L}{\Delta_L} = o(1) \cdot (1 - \Delta_L),$$

and similarly

$$\frac{(L-1)^2\gamma(\mathcal{G})\delta_L}{\Delta_L} \leq \gamma(\mathcal{G}) \cdot o(1) \cdot (1-\Delta_L).$$

Therefore,

$$|\mathcal{R}_A| \leq p_{L,1}\Delta_L^{L-1}\Big(1-\frac{1}{d}\Big)(1-\Delta_L)\big(o(1)+o(1)\gamma(\mathcal{G})\big) \ll p_{L,1}\Delta_L^{L-1}\Big(1-\frac{1}{d}\Big)(1-\Delta_L).$$

- **Bounding $\mathcal{R}_E$.** By Equation (51b) in Proposition E.1,

$$|\mathcal{R}_E| \leq \Delta_L^L\Big(1-\frac{1}{d}\Big)\left(O\left(\frac{\sigma_{G^L}^2\delta_L^2}{\Delta_L^2}\right) + \frac{(L-1)L\gamma(\mathcal{G})\delta_L}{\Delta_L}\right).$$

Using the same estimates as above and $\Delta_L \leq 1$, we obtain

$$|\mathcal{R}_E| \ll \Delta_L^L\Big(1-\frac{1}{d}\Big)(1-\Delta_L).$$

Consequently, $p_{L,1}|\mathcal{R}_E|$ is dominated by $p_{L,1}\Delta_L^L\Big(1-\frac{1}{d}\Big)(1-\Delta_L)$.

- **Bounding $\mathcal{R}_B$.** By Equation (51c) in Proposition E.1,

$$\frac{|\mathcal{R}_B|}{L-1} \leq p_{L,2}\Delta_L^{L-1}\Big(1-\frac{1}{d}\Big)\left(O\left(\frac{\sigma_{G^L}^2\delta_L}{(L-1)\Delta_L}\right) + \gamma(\mathcal{G})\right).$$

Using $\sigma_{G^L} \leq L-1$ and $\frac{L^2\delta_L}{\Delta_L} = o(1)\cdot(1-\Delta_L)$, we have

$$\frac{\sigma_{G^L}^2\delta_L}{(L-1)\Delta_L} \leq \frac{L^2\delta_L}{\Delta_L} = o(1)\cdot(1-\Delta_L).$$

Moreover, $\frac{p_{L,2}}{p_{L,1}} = o(1)\cdot(1-\Delta_L)$ implies

$$p_{L,2}\Delta_L^{L-1} \leq p_{L,1}\Delta_L^{L-1}\cdot o(1)\cdot(1-\Delta_L).$$

Thus,

$$\frac{|\mathcal{R}_B|}{L-1} \leq p_{L,1}\Delta_L^{L-1}\Big(1-\frac{1}{d}\Big)(1-\Delta_L)\big(o(1)+\gamma(\mathcal{G})\cdot o(1)\big) \ll p_{L,1}\Delta_L^{L-1}\Big(1-\frac{1}{d}\Big)(1-\Delta_L).$$

Finally, the same assumption $\frac{p_{L,2}}{p_{L,1}} = o(1)\cdot(1-\Delta_L)$ also yields

$$p_{L,2}\Delta_L^L\Big(1-\frac{1}{d}\Big) \ll p_{L,1}\Delta_L^{L-1}\Big(1-\frac{1}{d}\Big)(1-\Delta_L),$$

so the contribution of the second posterior deviation term is dominated by the first term.

Plugging the above bounds into the expression for $\nabla_q\mathcal{J}_L$, and using that $\mathbf{Attn}_L = \Theta(1)$, which is implied by $\frac{p_{L,2}}{p_{L,1}} = o(1)\cdot(1-\Delta_L)$, we conclude that

$$\nabla_q\widetilde{\mathcal{J}}_L = \Theta(\log d/d_{\mathsf{p}})\cdot p_{L,1}\Delta_L^{L-1}\Big(1-\frac{1}{d}\Big)(1-\Delta_L).$$

The analysis for $|\nabla_r\widetilde{\mathcal{J}}_L|$ is similar. Alternatively, we may invoke the direct comparison bound in Lemma C.3 to obtain $|\nabla_r\widetilde{\mathcal{J}}_L| = O(1/d_{\mathsf{p}})\cdot\nabla_q\widetilde{\mathcal{J}}_L$. $\qquad\square$

## E.6. Exponentially Flat Region for Long-Horizon Tasks: Proof of Proposition 3.1

Following the same decomposition underlying Lemma E.4, we show that when the step-invariant probability tuple $(p_{L,1}, p_{L,2}, p_{L,3})$ has small effective margins $\Delta_L := p_{L,1} - p_{L,3}$ and $\delta_L := p_{L,2} - p_{L,3}$, the resulting policy gradient is upper bounded by a quantity that decays exponentially in the horizon length $L$. We then specialize this general exponential barrier to our concrete setting, which immediately yields Proposition 3.1.

**Proposition E.3.** *Under Assumptions 2.1–2.3, for any $2 \leq L \leq L_{\max}$, suppose the step-invariant probability tuple $(p_{L,1}, p_{L,2}, p_{L,3})$ satisfies, with $\Delta_L := p_{L,1} - p_{L,3}$ and $\delta_L := p_{L,2} - p_{L,3}$,*

$$\Delta_L + L\delta_L \leq \widetilde{O}\left(d^{-\Omega(1)}\right), \qquad p_{L,i} \leq d^{-\Omega(1)} \text{ for } i \in [3]. \tag{53}$$

*Then,*

$$\left|\nabla_q \widetilde{\mathcal{J}}_L\right| \leq \widetilde{O}\left(\frac{1}{d_{\mathsf{p}}}\right) \cdot d^{-\Omega(L)}, \qquad \left|\nabla_r \widetilde{\mathcal{J}}_L\right| \leq \widetilde{O}\left(\frac{1}{d_{\mathsf{p}}^2}\right) \cdot d^{-\Omega(L)}. \tag{54}$$

*Proof.* A key takeaway from Lemma E.4 is that, when bounding the remainder contributions (e.g., $\mathcal{R}_E$), we decompose the remainder term $T_{\text{res}}(\lambda)$ into several parts. Independent of this finer decomposition, its operator norm admits the crude bound

$$\|T_{\text{res}}(\lambda)\|_{\text{op}} \leq (\Delta_L + \sigma_{G_L}\delta_L)^L - \Delta_L^L.$$

Using this bound directly gives

$$\mathcal{R}_E \leq \left(1 - \frac{1}{d}\right)(\Delta_L + \sigma_{G_L}\delta_L)^L.$$

The same argument applies to $\mathcal{R}_A$ and $\mathcal{R}_B$, yielding

$$\mathcal{R}_A \leq \left(1 - \frac{1}{d}\right)p_{L,1}(\Delta_L + \sigma_{G_L}\delta_L)^{L-1}, \qquad \mathcal{R}_B \leq \left(1 - \frac{1}{d}\right)p_{L,2}\sigma_{G_L}(\Delta_L + \sigma_{G_L}\delta_L)^{L-1}.$$

Invoking Equation (49) from Proposition E.1 and substituting the above bounds into Equation (52), we obtain

$$\left|\nabla_q \widetilde{\mathcal{J}}_L\right| \leq \widetilde{O}\left(\frac{B}{d_{\mathsf{p}}}\right)\left(p_{L,1}\Delta_L^{L-1} + p_{L,1}(\Delta_L + \sigma_{G_L}\delta_L)^{L-1} + p_{L,2}\frac{\sigma_{G_L}}{L-1}(\Delta_L + \sigma_{G_L}\delta_L)^{L-1}\right)$$
$$\leq \widetilde{O}\left(\frac{1}{d_{\mathsf{p}}}\right) \cdot d^{-\Omega(L)},$$

where in the last step we use $\sigma_{G_L} \leq L - 1$ together with the assumptions $\Delta_L + L\delta_L \leq \widetilde{O}\left(d^{-\Omega(1)}\right)$ and $p_{L,i} \leq d^{-\Omega(1)}$ for $i \in [3]$. The bound for $\left|\nabla_r \widetilde{\mathcal{J}}_L\right|$ follows by the same reasoning and is omitted. $\square$

**Proposition E.4** (Proposition 3.1 restated)**.** *Under Assumptions 2.1–2.3, suppose $\mathsf{TF}_{\theta^{(0)}}$ is initialized according to Assumption 2.4. Then for any horizon $L > 2C_B$, whenever the feature magnitudes satisfy $\max\{|r^{(t)}|, |q^{(t)}|\} \leq 0.01$, we have $\mathcal{J}_L^{(t)} = \frac{1}{d}(1 \pm o(1))$, and*

$$|\nabla_q \widetilde{\mathcal{J}}_L^{(t)}| \leq \widetilde{O}\left(\frac{1}{d_{\mathsf{p}}}\right) \cdot d^{-\Omega(L)}, \qquad |\nabla_r \widetilde{\mathcal{J}}_L^{(t)}| \leq \widetilde{O}\left(\frac{1}{d_{\mathsf{p}}^2}\right) \cdot d^{-\Omega(L)}.$$

*Proof.* Since $\max\{|r^{(t)}|, |q^{(t)}|\} \leq 0.01$, the attention weights satisfy

$$\mathbf{Attn}_L^{(t)}C_B \leq \frac{C_B e^{0.02}}{e^{0.02} + L - 1} < 1, \qquad \frac{1 - \mathbf{Attn}_L^{(t)}}{L - 1}C_B \leq \frac{C_B}{e^{-0.02} + L - 1} < 1.$$

In particular, this implies $p_{L,i}^{(t)} \leq d^{-\Omega(1)}$ for all $i \in [3]$. It remains to bound $\Delta_L^{(t)} + L\delta_L^{(t)}$, which we do by considering two regimes.

**Case 1: $L < d^{0.01}$.** By Lemma C.7,

$$p_{L,1}^{(t)} \leq O\left(\frac{1}{L + d^{1 - \frac{C_B e^{0.02}}{e^{0.02} + L - 1}}}\right) = d^{-\Omega(1)}, \qquad p_{L,2}^{(t)} \leq O\left(\frac{1}{L + d^{1 - \frac{C_B}{e^{-0.02} + L - 1}}}\right) \leq d^{-0.5}.$$

Therefore,

$$\Delta_L^{(t)} + L\delta_L^{(t)} \leq p_{L,1}^{(t)} + L p_{L,2}^{(t)} \leq d^{-\Omega(1)}.$$

**Case 2: $L \geq d^{0.01}$.** In this regime, we bound $\Delta_L^{(t)}$ and $\delta_L^{(t)}$ directly. In particular,

$$\Delta_L^{(t)} \leq O\left(\frac{e^{\frac{C_B e^{0.02}}{e^{0.02} + L - 1} \log d} - 1}{d}\right) \leq \widetilde{O}\left(\frac{1}{Ld}\right),$$

and

$$L\delta_L^{(t)} \leq O\left(L \cdot \frac{e^{\frac{C_B}{e^{-0.02} + L - 1} \log d} - 1}{d}\right) \leq \widetilde{O}\left(\frac{1}{d}\right).$$

Thus,

$$\Delta_L^{(t)} + L\delta_L^{(t)} \leq \widetilde{O}\left(\frac{1}{d}\right).$$

In both regimes, the conditions of Proposition E.3 are satisfied. Therefore, applying Proposition E.3 yields the desired gradient bound. Moreover, $\mathcal{J}_L^{(t)} = \frac{1}{d}(1 \pm o(1))$ follows directly from Lemma E.6.

$\square$

### E.7. Reward Characterization

Note that $\widetilde{\mathbb{P}}(E)$ is exactly the expected reward for a fixed instance $(y_0, G_L)$. Consequently,

$$\mathcal{J}_L = \mathbb{E}_{Z^L}\left[\widetilde{\mathbb{P}}(E)\right].$$

Therefore, the gradient characterization in Proposition E.2 immediately yields a corresponding characterization of the reward.

**Lemma E.5.** *Given a problem of length L, suppose that*

$$\frac{L^2 \delta_L}{\Delta_L} = o(1) \cdot (1 - \Delta_L) \qquad \text{and} \qquad \frac{p_{L,2}}{p_{L,1}} = o(1) \cdot (1 - \Delta_L).$$

*Then,*

$$\mathcal{J}_L = \frac{1}{d} + \left(1 - \frac{1}{d}\right)(1 \pm o(1)) \cdot \Delta_L^L.$$

Moreover, by adapting the argument in Proposition E.3 to control the residual term $\mathcal{R}_E$, we obtain the following coarse upper bound.

**Lemma E.6.** *Given a problem of length L, we have*

$$\left|\mathcal{J}_L - \frac{1}{d}\right| \leq \left(1 - \frac{1}{d}\right)\left(\Delta_L + \sigma_{G_L} \delta_L\right)^L.$$

## F. Learning Dynamics of Mixed-difficulty RL

In this section, we study the mixed-difficulty setting, where tasks of different lengths are interleaved. By combining the constant-length analysis in Section D with the gradient characterizations from Section E, we analyze two regimes of the difficulty ratio $R$: (i) the large difficulty ratio regime $R = \omega(1)$, which gives rise to grokking-style dynamics, and (ii) the moderate difficulty ratio regime $R = O(1)$, which leads to smoother relay dynamics.

We begin by reviewing the mixed-difficulty setup and introducing some timestamps that will be useful for characterizing the overall learning dynamics.

**Mixed-difficulty setup.** Let $R > 1$ denote the *difficulty ratio*, and set the starting (effectively short) horizon to be $L_1 := C_B$. Define the horizon set $\mathcal{L}_R = \{L_1, L_2, \ldots, L_K\}$ recursively by

$$L_k = \min\{\lceil RL_{k-1} \rceil, L_{\max}\}, \qquad 2 \le k \le K,$$

where $K = \lceil \log_R(L_{\max}/L_1) \rceil$, so that $L_K = L_{\max}$. For simplicity, we focus on the case $R \ge 2$ throughout.

**Mastery and visible return states.** For any $L_i \in \mathcal{L}_R$, we say the horizon $L_i$ has *visible return* at time $t$ if

$$\mathcal{J}_{L_i}^{(t)} \ge 0.01. \tag{55}$$

Denote the first iteration such that $L_i$ has visible return as $T_{\mathsf{vis},i}$. We say the horizon $L_i$ is *mastered* at time $t$ if

$$\mathcal{J}_{L_i}^{(t)} \ge 0.99. \tag{56}$$

Denote the first iteration such that $L_i$ is mastered as $T_{\mathsf{mas},i}$.

**Plateau between consecutive horizons.** For $k \in \{1, \ldots, K-1\}$, define

$$\mathcal{T}_k \triangleq T_{\mathsf{vis},k+1} - T_{\mathsf{mas},k} = \left| \left\{ t \,\middle|\, \mathcal{J}_{L_k}^{(t)} \ge 0.99, \ \mathcal{J}_{L_{k+1}}^{(t)} < 0.01 \right\} \right|. \tag{57}$$

In words, $\mathcal{T}_k$ counts the number of iterations during which $L_k$ is already mastered while $L_{k+1}$ has not yet achieved a visible return.

## F.1. Analysis of Large Difficulty Gap Regime

In this subsection, we analyze the large difficulty ratio regime, where $R = \omega(1)$. Following the similar proof strategy as in Section D, we start with the induction hypothesis that is expected to hold through the training process.

**Induction F.1.** *Given* $\Omega\left(\frac{1}{\mathsf{poly}\log d}\right) < \epsilon < \frac{1}{4}$, *and let* $T^\star$ *be the first iteration such that* $\mathbf{Attn}_{L_{\max}}^{(t)} \ge 1 - \epsilon$. *Then, for all iterations* $t < T^\star$, *we have the following holds:*

*(a)* $0 \le q^{(t)} \le O\left(\log \frac{L_{\max}}{\epsilon}\right)$, *and* $q^{(t)}$ *monotonically increases.*
*(b)* $|r^{(t)}| \le O(1/d_{\mathsf{p}})q^{(t)}$.

### F.1.1. PROPERTIES OF THE ATTENTION SCORES AND CRITICAL THRESHOLDS

We record some properties of the attention scores and critical thresholds.

**Lemma F.1.** *If Induction F.1 holds for all iterations* $< t$, *then we have*

*(a)* $\mathbf{Attn}_L^{(t)} = \frac{e^{q^{(t)}-r^{(t)}}}{e^{q^{(t)}-r^{(t)}}+(L-1)} \ge \frac{1}{L}$;
*(b)* $\mathbf{Attn}_{a,\ell-1\to p,k}^{(t)} = \frac{1}{(L-1)+e^{q^{(t)}-r^{(t)}}} = \frac{1}{L-1}\left(1 - \mathbf{Attn}_L^{(t)}\right)$ *for* $k \ne \ell$.

**Lemma F.2** (Critical threshold of $q$). *If Induction F.1 holds, then given* $L \in \mathcal{L}_R$, *the critical threshold of $q$ required to satisfy* $\mathcal{J}_L \ge 1 - \xi$ *for some constant* $0 < \xi \le 1$ *is given by:*

$$q \ge \log \frac{L-1}{C_B - 1} + f\left(\frac{\log L - \log\log \frac{1}{1-\xi}}{\log d}\right)$$

$$\ge \log \frac{L-1}{C_B - 1} + \frac{C_B}{C_B - 1} \cdot \frac{\log L - \log\log \frac{1}{1-\xi}}{\log d} + \mathcal{O}\left(\frac{\log^2 L}{\log^2 d}\right).$$

*where* $f(x) = \log\left(\frac{1+x}{1-x/(C_B-1)}\right)$. *Similarly, the critical threshold of $q$ required to satisfy* $\mathbf{Attn}_L \ge 1 - \xi$ *for any* $0 < \xi \le 1$ *is given by:*

$$q \ge \log \frac{(1-\xi)(L-1)}{\xi}.$$

*Proof.* Given $\mathcal{J}_L \geq 1 - \xi$, by Lemma E.5, we have

$$\Delta_L \geq (1 - \xi)^{1/L} = 1 - \frac{-\log(1 - \xi)}{L}.$$

Then, by Lemma C.7, we can derive that

$$\mathbf{Attn}_L \geq \frac{1}{C_B} + \frac{\log L - \log(-\log(1 - \xi))}{C_B \log d}.$$

Hence, applying Lemma F.1, we have

$$q \geq \log(L - 1) + \log\left(\frac{\mathbf{Attn}_L}{1 - \mathbf{Attn}_L}\right) \geq \log \frac{L - 1}{C_B - 1} + f\left(\frac{\log L - \log\log \frac{1}{1-\xi}}{\log d}\right).$$

Here, we then use the first-order Taylor expansion for $f(x)$ to get the second inequality. $\qquad\square$

Notice that in the large difficulty ratio regime, the changes in $\log L$ between two consecutive horizons are $\Omega(\log R) \gg 1$, which is much larger than the $\frac{\log L}{\log d} \leq O(1)$ term. Therefore, the above lemma implies that the change in $q$ between two consecutive horizons is dominated by $\Omega(\log R)$.

### F.1.2. WARM-UP STAGE FOR $L_1$

We define the warm-up stage as the period during which the starting horizon $L_1$ reaches the mastery state, namely $0 \leq t < T_{\mathsf{mas},1}$. At initialization, the attention scores are essentially uniform across horizons. We will show that, during this stage, the only non-negligible gradient contribution comes from the effectively short horizon $L_1$.

We first record several basic properties of $q$, $r$, and the attention scores throughout the warm-up stage.

**Lemma F.3.** *If Induction F.1 holds, then for all iterations $0 \leq t < T_{\mathsf{mas},1}$:*

*(a) $0 \leq q^{(t)} \leq O\left(\frac{L_1}{\log d}\right)$, and $q^{(t)}$ is monotonically increasing in $t$.*
*(b) $|r^{(t)}| \leq O\left(\frac{1}{d_{\mathsf{p}}}\right) q^{(t)}$.*

*Proof.* The range of $q^{(t)}$ is a direct consequence of Lemma F.2. The monotonicity of $q^{(t)}$ and the bound on $r^{(t)}$ follow directly from Induction F.1. $\qquad\square$

**Lemma F.4.** *If Induction F.1 holds, then for all iterations $0 \leq t < T_{\mathsf{mas},1}$ and for any horizon $L_i$ with $i \geq 2$, we have*

$$\mathbf{Attn}^{(t)}_{\mathsf{ans},\ell-1\to k} \leq O\left(\frac{1}{L_i}\right) = o(1), \qquad \forall \ell \in [L_i], k \in [\ell].$$

*Proof.* This follows directly from Lemma F.1. Moreover, since $R = \omega(1)$ in this regime, we have $L_i \geq \omega(1)$ for all $i \geq 2$, so the bound is indeed $o(1)$ as $d$ grows. $\qquad\square$

Combining the above with the same reasoning as in Proposition E.4, we can verify that the condition Equation (53) in Proposition E.3 holds for all longer horizons during warm-up, which yields the following.

**Lemma F.5.** *If Induction F.1 holds, then for all iterations $0 \leq t < T_{\mathsf{mas},1}$ and for any horizon $L_i$ with $i \geq 2$, we have*

$$\left|\nabla_q \widetilde{\mathcal{J}}^{(t)}_{L_i}\right| \leq \widetilde{O}\left(\frac{1}{d_{\mathsf{p}}}\right) \cdot d^{-\Omega(L_i)}, \qquad \left|\nabla_r \widetilde{\mathcal{J}}^{(t)}_{L_i}\right| \leq \widetilde{O}\left(\frac{1}{d_{\mathsf{p}}^2}\right) \cdot d^{-\Omega(L_i)}.$$

Compared with Lemmas D.3 and D.4 in Section D, Lemma F.5 shows that during warm-up, the gradients contributed by longer horizons $L_i$ (for $i \geq 2$) are negligible relative to the shortest horizon $L_1 = C_B$. Therefore, we can apply the constant-length analysis from Section D to the warm-up stage for $L_1$, which yields the following characterization at the end of warm-up.

**Lemma F.6.** *Induction F.1 holds through* $0 \leq t < T_{\mathsf{mas},1}$ *with*

$$T_{\mathsf{mas},1} = O\left(\frac{K L_{\max} L_1}{\eta \log^2 d}\right),$$

*and at time* $T_{\mathsf{mas},1}$ *we have* $q^{(T_{\mathsf{mas},1})} \geq \Omega\left(\frac{\log L_1}{\log d}\right)$.

### F.1.3. TRANSITION BETWEEN MASTERY STATES

Since we have established that the initial horizon can reach the mastery state, we next analyze how mastery propagates across *consecutive* horizons. Specifically, we study the transition from horizon $i$ to horizon $i + 1$ over the time interval $[T_{\mathsf{mas},i}, T_{\mathsf{mas},i+1})$.

Recall the definition $K = \lceil \log_R (L_{\max}/L_1) \rceil$. By construction, the horizons grow by a factor $R$ up to index $K - 1$, while the last step may be truncated so that $L_K = L_{\max}$; consequently, $L_K/L_{K-1}$ is not necessarily equal to $R$. For notational convenience, we therefore restrict attention to $i \in \{1, \ldots, K - 2\}$, and fix an arbitrary $i^\star \in \{1, \ldots, K - 2\}$ for the remainder of the analysis. Moreover, we absorb the gradient term $\nabla_q \mathcal{J}_{L_K}$ into $\nabla_q \mathcal{J}_{L_{K-1}}$, since for all times prior to $T_{\mathsf{mas},K-1}$, $\nabla_q \mathcal{J}_{L_K}$ can be upper bounded by $\nabla_q \mathcal{J}_{L_{K-1}}$.

By the critical threshold of $q$ in Lemma F.2, we have the following characterization of the attention scores:

**Lemma F.7.** *If Induction F.1 holds, then for all iterations* $T_{\mathsf{mas},i^\star} \leq t < T_{\mathsf{mas},i^\star+1}$:

*(a) if* $i^\star > 1$, *then for any* $i < i^\star$, *we have*

$$\mathbf{Attn}_{L_i}^{(t)} \geq 1 - O\left(\frac{1}{R^{i^\star - i}}\right) \cdot (1 - \mathbf{Attn}_{L_{i^\star}}^{(t)}) = 1 - o(1).$$

*(b) for* $i = i^\star$, *we have*

$$\frac{1}{C_B} + \Omega\left(\frac{\log L_i}{\log d}\right) < \mathbf{Attn}_{L_i}^{(t)} \leq 1 - \Omega\left(\frac{1}{R}\right).$$

*(c) if* $i^\star < K - 2$, *then for any* $i > i^\star + 1$, *we have*

$$\mathbf{Attn}_{L_i}^{(t)} \leq O\left(\frac{1}{L_i}\right).$$

This immediately implies the following characterization of the logits:

**Lemma F.8.** *If Induction F.1 holds, then for all iterations* $T_{\mathsf{mas},i^\star} \leq t < T_{\mathsf{mas},i^\star+1}$:

*(a) if* $i^\star > 1$, *then for any* $i < i^\star$, *we have* $(p_{L_i,1}^{(t)})^{L_i} \geq \Omega(\mathcal{J}_{L_{i^\star}}^{(t)}) = \Omega(1)$, *and also*

$$\Omega\left(\frac{1}{d^{C_B-1}}\right) \leq 1 - p_{L_i,1}^{(t)} \leq O\left(\frac{1}{d^{(1-e^{-q^{(t)}}R^{-(i^\star-i)}L_{i^\star})C_B-1}}\right).$$

*(b) for* $i = i^\star$, *we have*

$$1 - p_{L_i,1}^{(t)} \geq \Omega\left(\frac{1}{d^{(1-\Theta(e^{-q^{(t)}}L_i))C_B-1}}\right).$$

*(c) if* $i^\star < K - 2$, *then for any* $i > i^\star + 1$, *we have*

$$p_{L_i,1}^{(t)} \leq O\left(\frac{1}{d}\right).$$

The logit conditions imply that for any $i < i^\star$, we can invoke the gradient characterization in Proposition E.2, and for any $i > i^\star + 1$, we can invoke the gradient characterization in Proposition E.3. Therefore, we have the following characterization of the gradient:

**Lemma F.9.** *If Induction F.1 holds, then for all iterations* $T_{\mathsf{mas},i^\star} \leq t < T_{\mathsf{mas},i^\star+1}$,

(a) if $i^\star > 1$, then for any $i < i^\star$, we have

$$\Omega\Big(\frac{1}{d^{C_B-1}}\Big)\cdot\frac{\log d}{d_{\mathsf{p}}} \leq \nabla_q\widetilde{\mathcal{J}}_{L_i}^{(t)} \leq O\Big(\frac{1}{d^{(1-\frac{L_{i^\star}}{e^{q^{(t)}}R^{i^\star-i}})C_B-1}}\Big)\cdot\frac{\log d}{d_{\mathsf{p}}}$$

(b) for $i = i^\star$, we have

$$\nabla_q\widetilde{\mathcal{J}}_{L_i}^{(t)} = \Omega\Big(\frac{1}{d^{(1-\Theta(e^{-q^{(t)}}L_i))C_B-1}}\Big)\cdot\frac{\log d}{d_{\mathsf{p}}}$$

(c) if $i^\star < K - 2$, then for any $i > i^\star + 1$, we have

$$|\nabla_q\widetilde{\mathcal{J}}_{L_i}^{(t)}| \leq \widetilde{O}\Big(\frac{1}{d_{\mathsf{p}}}\Big)\cdot d^{-\Omega(L_i)}.$$

Lemma F.9 immediately implies a gradient lower bound for $\mathcal{J}_{\mathrm{mix},R}$ during $[T_{\mathsf{mas},i^\star}, T_{\mathsf{mas},i^\star+1})$:

**Lemma F.10.** *If Induction F.1 holds, then for all iterations $T_{\mathsf{mas},i^\star} \leq t < T_{\mathsf{mas},i^\star+1}$, we have*

$$\nabla_q\widetilde{\mathcal{J}}_{\mathrm{mix},R}^{(t)} \geq \frac{\log d}{Kd_{\mathsf{p}}}\Omega\Big(\frac{i^\star}{d^{C_B-1}}\Big).$$

*Proof.* By Lemma F.9, when $R \leq o(\log d)$, we have

$$\nabla_q\widetilde{\mathcal{J}}_{L_{i^\star}}^{(t)}/\nabla_q\widetilde{\mathcal{J}}_{L_i}^{(t)} \leq O\big(d^{1/R}\big) = O(e^{\log d/R}) \gg 1.$$

Thus $\nabla_q\widetilde{\mathcal{J}}_{L_{i^\star}}^{(t)}$ dominates the gradient of short horizons, which leads to the following lower bound:

$$\nabla_q\widetilde{\mathcal{J}}_{\mathrm{mix},R}^{(t)} \geq \frac{\log d}{Kd_{\mathsf{p}}}\Omega\Big(\frac{1}{d^{(1-\frac{1}{R})C_B-1}}\Big).$$

On the other hand, we have

$$\nabla_q\widetilde{\mathcal{J}}_{\mathrm{mix},R}^{(t)} \geq \frac{1}{K}\sum_{i=1}^{i^\star}\nabla_q\widetilde{\mathcal{J}}_{L_i}^{(t)} \geq \Omega\Big(\frac{i^\star\log d}{Kd_{\mathsf{p}}d^{C_B-1}}\Big).$$

Further noting that $i^\star \leq K - 2 \leq O(\log d)$, thus when $R \leq o(\log d)$, we have $d^{\frac{C_B}{R}} \geq i^\star$, which implies that in both cases, we have

$$\nabla_q\widetilde{\mathcal{J}}_{\mathrm{mix},R}^{(t)} \geq \frac{\log d}{Kd_{\mathsf{p}}}\Omega\Big(\frac{i^\star}{d^{C_B-1}}\Big).$$

$\square$

So far, we have already controlled the gradient for the horizons before or after the current consecutive mastery state. In the following, we are going to exam $\nabla_q\widetilde{\mathcal{J}}_{L_{i^\star}}^{(t)} + \nabla_r\widetilde{\mathcal{J}}_{L_{i^\star+1}}^{(t)}$.

**Lemma F.11.** *If Induction F.1 holds, then for all iterations $T_{\mathsf{vis},i^\star+1} \leq t < T_{\mathsf{mas},i^\star+1}$, we have*

$$\nabla_q\widetilde{\mathcal{J}}_{L_{i^\star}}^{(t)} + \nabla_q\widetilde{\mathcal{J}}_{L_{i^\star+1}}^{(t)} \geq \Omega\Big(\frac{\log d}{L_{i^\star+1}d_{\mathsf{p}}}\Big)$$

*Proof.* By the critical threshold of $q$ in Lemma F.2, when $t \geq T_{\mathsf{vis},i^\star+1}$, we have $p_{L_{i^\star}+1,1}^{(t)} \geq 1 - O(\frac{1}{L_{i^\star+1}})$. Hence, the conditions of Proposition E.2 are satisfied, and invoking it, we then obtain

$$\nabla_q\widetilde{\mathcal{J}}_{L_{i^\star}+1}^{(t)} \geq \Omega\Big(\frac{\log d}{d_{\mathsf{p}}}\Big)(1 - p_{L_{i^\star}+1,1}^{(t)}).$$

On the other hand, since $t \leq T_{\mathsf{mas},i^\star+1}$, again by Lemma F.2, we have $p_{L_{i^\star},1}^{(t)} \leq 1 - \Omega(\frac{1}{L_{i^\star}})$. Thus, we have

$$\nabla_q\widetilde{\mathcal{J}}_{L_{i^\star}+1}^{(t)} \geq \Omega\Big(\frac{\log d}{L_{i^\star+1}d_{\mathsf{p}}}\Big),$$

which completes the proof. $\square$

In the following, we are going to show that during the time period $[T_{\mathsf{mas},i^\star}, T_{\mathsf{vis},i^\star+1})$, there exists a major period during which the gradient is dominated by the current mastery state $L_{i^\star}$.

**Lemma F.12.** *If Induction F.1 holds, then during* $[T_{\mathsf{mas},i^\star}, T_{\mathsf{vis},i^\star+1})$, *when*

$$q^{(t)} \in [\Omega(\log R^{0.01} L_{i^\star}), O(\log R^{0.99} L_{i^\star})] \tag{58}$$

*we have*

$$\nabla_q \widetilde{\mathcal{J}}_{L_{i^\star}}^{(t)} + \nabla_q \widetilde{\mathcal{J}}_{L_{i^\star}+1}^{(t)} = (1 + o(1)) \nabla_q \widetilde{\mathcal{J}}_{L_{i^\star}}^{(t)}.$$

*Moreover,*

$$\nabla_q \widetilde{\mathcal{J}}_{L_{i^\star}}^{(t)} = \Theta\Big(\frac{1}{d^{(1-e^{-q^{(t)}} L_{i^\star})C_B - 1}}\Big) \cdot \frac{\log d}{d_{\mathsf{p}}}.$$

*Proof.* By the critical threshold of $q$ in Lemma F.2,

$$\Omega(\log L_{i^\star}) \le q^{(t)} \le O(\log R L_{i^\star}) = O(\log L_{i^\star+1}).$$

So the condition Equation (58) is well-defined. Furthermore, by Lemma F.2, when Equation (58) holds, we have $p_{L_{i^\star}+1,1}^{(t)} \le O(\frac{1}{d})$. Hence applying Proposition E.3, and we have

$$\big|\nabla_q \widetilde{\mathcal{J}}_{L_{i^\star}+1}^{(t)}\big| \le \widetilde{O}\Big(\frac{1}{d_{\mathsf{p}}}\Big) \cdot d^{-\Omega(L_{i^\star+1})}.$$

Furthermore, Equation (58) combined with Lemma F.2 implies that

$$1 - \mathbf{Attn}_{L_{i^\star}}^{(t)} = \Theta(e^{-q^{(t)}} L_{i^\star}).$$

Hence,

$$1 - p_{L_{i^\star},1}^{(t)} = \Theta\Big(\frac{1}{d^{(1-e^{-q^{(t)}} L_{i^\star})C_B - 1}}\Big).$$

Therefore, invoking Proposition E.2, we complete the proof. $\qquad\square$

Putting everything together, we can then characterize the grokking-style behaviour happening during the transition period $[T_{\mathsf{mas},i^\star}, T_{\mathsf{mas},i^\star+1})$.

**Lemma F.13.** *Induction F.1 holds through* $[T_{\mathsf{mas},i^\star}, T_{\mathsf{mas},i^\star+1})$, *where* $T_{\mathsf{mas},i^\star+1} = T_{\mathsf{mas},i^\star} + O\Big(\frac{d^{C_B-1}K d_{\mathsf{p}} \log R}{\eta i^\star \log d}\Big)$

*(a) the reward of* $J_{L_{i^\star}+1}$ *saturates below* $0.01$ *for a time period of*

$$\mathcal{T}_k \ge \Omega\Big(\frac{d^{C_B-1}K d_{\mathsf{p}}}{i^\star \eta \log d}\Big) \cdot \frac{\log R}{1 + C_B R^{-0.01} \log d}.$$

*(b)* $T_{\mathsf{mas},i^\star+1} - T_{\mathsf{vis},i^\star+1} \le O(\frac{L_{i^\star+1} d_{\mathsf{p}} K}{\eta \log d})$.

*Proof.* The existence of $T_{\mathsf{mas},i^\star+1} = T_{\mathsf{mas},i^\star} + O\Big(\frac{d^{C_B-1}K d_{\mathsf{p}} \log R}{\eta i^\star \log d}\Big)$ is guaranteed by the gradient lower bound in Lemma F.10. Moreover, the second item is guaranteed by the gradient lower bound in Lemma F.11. Then we focus on the first statement. We approximate the total number of iterations $\mathcal{T}_{i^\star}$ by the integral

$$\mathcal{T}_{i^\star} \gtrsim \int_{\Omega(\log R^{0.01} L_{i^\star})}^{O(\log R^{0.99} L_{i^\star})} \frac{dq}{\eta \nabla_q \widetilde{\mathcal{J}}_{\mathrm{mix},R}}.$$

By Lemma F.12, we can have a naive upper bound on the gradient:

$$\nabla_q \widetilde{\mathcal{J}}_{\mathrm{mix},R} \le \frac{i^\star}{K} \cdot \nabla_q \widetilde{\mathcal{J}}_{L_{i^\star}}^{(t)} \le O\Big(\frac{i^\star \log d}{K d_{\mathsf{p}}}\Big) \cdot \frac{1}{d^{(1-e^{-q^{(t)}L_{i^\star}})C_B - 1}}.$$

Plugging this into the integral, we have

$$\mathcal{T}_{i^\star} \ge \Omega\Big(\frac{d^{C_B-1}Kd_{\mathsf{p}}}{i^\star \eta \log d}\Big) \int_{\log R^{0.01}}^{\log R^{0.99}} d^{-C_B e^{-q}} dq = \Omega\Big(\frac{d^{C_B-1}Kd_{\mathsf{p}}}{i^\star \eta \log d}\Big) \int_{R^{-0.99}}^{R^{-0.01}} \frac{e^{-(C_B \log d)u}}{u} du$$

$$\ge \Omega\Big(\frac{d^{C_B-1}Kd_{\mathsf{p}}}{i^\star \eta \log d}\Big) \cdot e^{-(C_B R^{-0.01} \log d)} \int_{R^{-0.99}}^{R^{-0.01}} \frac{1}{u} du$$

$$\ge \Omega\Big(\frac{d^{C_B-1}Kd_{\mathsf{p}}}{i^\star \eta \log d}\Big) \cdot e^{-(C_B R^{-0.01} \log d)} \cdot \log R$$

$$\ge \Omega\Big(\frac{d^{C_B-1}Kd_{\mathsf{p}}}{i^\star \eta \log d}\Big) \cdot \frac{\log R}{1 + C_B R^{-0.01} \log d},$$

where we use the Taylor expansion for the last inequality. $\qquad\square$

### F.1.4. PROOF OF THEOREM 4.1 AND COROLLARY 4.1

*Proof.* For any $k \in \{1, \ldots, K-2\}$, Lemma F.13 implies

$$\mathcal{T}_k \ge \Omega\Big(\frac{d^{C_B-1}K d_{\mathsf{p}}}{k \eta \log d}\Big) \cdot \frac{\log R}{1 + C_B R^{-0.01} \log d}$$

$$\ge \widetilde{\Omega}\Big(\frac{d^{C_B-1} d_{\mathsf{p}}}{\eta \log d}\Big),$$

where the last inequality uses $\frac{\log R}{1 + C_B R^{-0.01} \log d} = \Omega(1/\log d)$ and $K/k = \widetilde{\Omega}(1)$.

Moreover,

$$T_{\mathsf{mas},k+1} - T_{\mathsf{vis},k+1} \le O\Big(\frac{L_{k+1} d_{\mathsf{p}} K}{\eta \log d}\Big) \le O\Big(\frac{d_{\mathsf{p}}^2 K}{\eta \log d}\Big) \ll \mathcal{T}_k,$$

since $d_{\mathsf{p}} = d^{c_x}$ and $c_x < C_B - 1$. Therefore,

$$T_{\mathsf{mas},k+1} - T_{\mathsf{mas},k} \ge \widetilde{\Omega}\Big(\frac{d^{C_B-1} d_{\mathsf{p}}}{\eta \log d}\Big).$$

On the other hand, Lemma F.13 also gives the matching upper bound

$$T_{\mathsf{mas},k+1} - T_{\mathsf{mas},k} \le \widetilde{O}\Big(\frac{d^{C_B-1} d_{\mathsf{p}}}{\eta \log d}\Big).$$

Summing over $k \in \{1, \ldots, K-2\}$, we obtain

$$T_{\mathsf{mas},K-1} - T_{\mathsf{mas},1} = \widetilde{\Theta}\Big(\frac{d^{C_B-1} d_{\mathsf{p}}}{\eta \log d}\Big).$$

Finally, by Lemma F.6, the time spent in the warm-up stage is negligible compared to $T_{\mathsf{mas},K-1} - T_{\mathsf{mas},1}$. Moreover, the final step can be bounded as

$$T_{\mathsf{mas},K} - T_{\mathsf{mas},K-1} \le O\big(T_{\mathsf{mas},K-1} - T_{\mathsf{mas},K-2}\big).$$

This completes the proof. $\qquad\square$

## F.2. Analysis of Moderate Difficulty Ratio Regime

In this subsection, we analyze the moderate difficulty ratio regime, where $2 \leq R = O(1)$. Our overall proof strategy follows that of the large difficulty ratio regime, with several adjustments to account for the smaller gap between two consecutive scales.

We begin by stating the induction hypothesis, which we expect to remain valid throughout training.

**Induction F.2.** *Given* $\Omega\left(\frac{1}{\text{poly} \log d}\right) < \epsilon < \frac{1}{4}$, *and let* $T^{\star}$ *be the first iteration such that* $\mathbf{Attn}_{L_{\max}}^{(t)} \geq 1 - \epsilon$. *Then, for all iterations* $t < T^{\star}$, *we have the following holds:*

*(a)* $0 \leq q^{(t)} \leq O\left(\log \frac{L_{\max}}{\epsilon}\right)$, *and* $q^{(t)}$ *monotonically increases.*
*(b)* $|r^{(t)}| \leq O(1/d_{\mathsf{p}})q^{(t)}$.

### F.2.1. PROPERTIES OF THE ATTENTION SCORES AND CRITICAL THRESHOLDS

We record several basic properties of the attention scores and the critical thresholds.

**Lemma F.14.** *If Induction F.2 holds for all iterations* $< t$, *then:*

*(a)*

$$\mathbf{Attn}_{L}^{(t)} = \frac{e^{q^{(t)} - r^{(t)}}}{e^{q^{(t)} - r^{(t)}} + (L - 1)} \geq \frac{1}{L};$$

*(b) for any* $k \neq \ell$,

$$\mathbf{Attn}_{a,\ell-1 \to p,k}^{(t)} = \frac{1}{(L-1) + e^{q^{(t)} - r^{(t)}}} = \frac{1}{L-1}\left(1 - \mathbf{Attn}_{L}^{(t)}\right).$$

**Lemma F.15** (Critical threshold of $q$). *If Induction F.1 holds, then for any* $L \in \mathcal{L}_{R}$, *a sufficient threshold on* $q$ *for* $\mathcal{J}_{L} \geq 1 - \xi$, *where* $0 < \xi \leq 1$ *is a constant, is*

$$q \geq \log \frac{L-1}{C_{B} - 1} + f\left(\frac{\log L - \log \log \frac{1}{1-\xi}}{\log d}\right),$$

*where* $f(x) = \log\left(\frac{1+x}{1 - x/(C_{B}-1)}\right)$. *Similarly, a sufficient threshold on* $q$ *for* $\mathbf{Attn}_{L} \geq 1 - \xi$ *is*

$$q \geq \log \frac{(1-\xi)(L-1)}{\xi}.$$

The above lemmas mirror their counterparts in the large-difficulty regime. However, to track the variation of the critical threshold when $L$ increases only by a constant factor $R$, we need a more careful comparison than the coarse Taylor-expansion argument used for widely separated scales.

**Lemma F.16.** *If Induction F.2 holds and* $0 < \xi \leq 1$ *is a constant, let* $q_{\xi}(L)$ *denote the critical threshold of* $q$ *required to ensure* $\mathcal{J}_{L,1} \geq 1 - \xi$. *Then for any* $L_{k}, L_{k+1} \in \mathcal{L}_{R}$,

$$q_{\xi}(L_{k+1}) - q_{\xi}(L_{k}) = \log R \cdot \left(1 + O(1/\log d)\right).$$

*Proof.* By Lemma F.15, we have

$$\frac{dq_{\xi}(L)}{dL} = \frac{1}{L}\left[1 + \frac{1}{\log d} \cdot \frac{C_{B}}{(1 + X(L))(C_{B} - 1 - X(L))}\right],$$

where $X(L) = \frac{\log L - \log \log \frac{1}{1-\xi}}{\log d}$. Since $0 \leq X(L) \leq 1 + O(1/\log d)$, it follows that $\frac{dq_{\xi}(L)}{dL} = \frac{1}{L}\left(1 + O(1/\log d)\right)$. Therefore,

$$q_{\xi}(L_{k+1}) - q_{\xi}(L_{k}) = \int_{L_{k}}^{L_{k+1}} \frac{1}{L}\left(1 + O(1/\log d)\right) dL$$
$$= \log R \cdot \left(1 + O(1/\log d)\right).$$

$\square$

### F.2.2. WARM-UP STAGE FOR $L_1$

We define the warm-up stage as the period during which the starting horizon $L_1$ reaches the mastery state, namely $0 \le t < T_{\mathsf{mas},1}$. The analysis is similar to the large difficulty ratio regime, but with some modifications since $L_i$ with $i \ge 2$ could be relatively small and still at the constant-length regime.

**Lemma F.17.** *If Induction F.2 holds, then for all iterations $0 \le t < T_{\mathsf{mas},1}$:*

*(a) $0 \le q^{(t)} \le O\left(\frac{L_1}{\log d}\right)$, and $q^{(t)}$ is monotonically increasing in t.*

*(b) $|r^{(t)}| \le O\left(\frac{1}{d_{\mathsf{p}}}\right) q^{(t)}$.*

**Lemma F.18.** *If Induction F.2 holds, then for all iterations $0 \le t < T_{\mathsf{mas},1}$, and for any $L_i \in \mathcal{L}_R$ with $i \ge 2$, we have*

*(a) if $L_i = O(1)$, then $1 - C_B \mathbf{Attn}_{L_i}^{(t)} \ge 1 - \frac{1}{0.99 R^{i-1}+1}$;*

*(b) else, $\mathbf{Attn}_{L_i}^{(t)} \le O\left(\frac{1}{L_i}\right) = o(1)$.*

*Proof.* The second item is similar to the large difficulty ratio regime. For the first item, by Lemma F.17, we have

$$\mathbf{Attn}_{L_i}^{(t)} \le \frac{1}{(L_i - 1) \cdot e^{-O(L_1/\log d)} + 1} = \frac{1}{R^{i-1}C_B \cdot e^{-O(L_1/\log d)} + 1}.$$

Hence,

$$1 - C_B \mathbf{Attn}_{L_i}^{(t)} \ge 1 - \frac{1}{0.99 R^{i-1} + 1}.$$

$\square$

Similarly as Lemma F.5, the condition Equation (53) in Proposition E.3 holds for $L_i = \omega(1)$, which yields the following.

**Lemma F.19.** *If Induction F.2 holds, then for all iterations $0 \le t < T_{\mathsf{mas},1}$ and for any horizon $L_i = \omega(1)$, we have*

$$\left|\nabla_q \widetilde{\mathcal{J}}_{L_i}^{(t)}\right| \le \widetilde{O}\left(\frac{1}{d_{\mathsf{p}}}\right) \cdot d^{-\Omega(L_i)}, \qquad \left|\nabla_r \widetilde{\mathcal{J}}_{L_i}^{(t)}\right| \le \widetilde{O}\left(\frac{1}{d_{\mathsf{p}}^2}\right) \cdot d^{-\Omega(L_i)}.$$

Lemma F.18 shows that even some longer horizons $L_i$ are still at the constant-length regime, its target attention scores are still below $\frac{1}{C_B}$, which means $p_{L_i,1}^{(t)} \le d^{-\Omega(1)}$ is still close to 0. However, directly applying Proposition E.3 to $L_i = O(1)$ only gives a bound of $d^{-\Omega(L_i)}$, which may be too loose in the constant-length regime. Thus we use a variant of Proposition E.3 and precise characterization of $1 - C_B \mathbf{Attn}_{L_i}^{(t)}$ to get a more precise bound.

**Lemma F.20.** *If Induction F.2 holds, then for all iterations $0 \le t < T_{\mathsf{mas},1}$ and for any horizon $L_i = O(1)$ with $i \ge 2$, we have*

$$\left|\nabla_q \widetilde{\mathcal{J}}_{L_i}^{(t)}\right| \le \widetilde{O}\left(\frac{1}{d_{\mathsf{p}}}\right) \cdot d^{-L_i\left(1 - \frac{1}{0.99 R^{i-1}+1}\right)}, \qquad \left|\nabla_r \widetilde{\mathcal{J}}_{L_i}^{(t)}\right| \le \widetilde{O}\left(\frac{1}{d_{\mathsf{p}}^2}\right) \cdot d^{-L_i\left(1 - \frac{1}{0.99 R^{i-1}+1}\right)}.$$

Since $L_i \ge R^{i-1}C_B$, we have $L_i(1 - \frac{1}{0.99 R^{i-1}+1}) \ge C_B + \Omega(1)$. Compared with Lemma D.3 in Section D, Lemma F.19 and Lemma F.20 show that during warm-up the gradients contributed by longer horizons $L_i$ (for $i \ge 2$) are negligible relative to the shortest horizon $L_1 = C_B$. Therefore, we can apply the constant-length analysis from Section D to the warm-up stage for $L_1$, which yields the following characterization at the end of warm-up.

**Lemma F.21.** *Induction F.2 holds through $0 \le t < T_{\mathsf{mas},1}$ with*

$$T_{\mathsf{mas},1} = O\left(\frac{K L_{\max} L_1}{\eta \log^2 d}\right),$$

*and at time $T_{\mathsf{mas},1}$ we have $q^{(T_{\mathsf{mas},1})} \ge \Omega\left(\frac{\log L_1}{\log d}\right)$.*

### F.2.3. TRANSITION BETWEEN MASTERY STATES

In this part, we analyze the transition of the mastery state across consecutive horizons. Concretely, we focus on the time interval $[T_{\mathsf{mas},i}, T_{\mathsf{mas},i+1})$ for $i \in \{1, \dots, K-2\}$. As before, we fix an arbitrary $i^\star \in \{1, \dots, K-2\}$ for the remainder of the analysis (The restriction $i \le K-2$ excludes the final truncated step where $L_K/L_{K-1}$ is not necessarily $R$.)

By the critical threshold of $q$ in Lemma F.15, we have the following characterization of the attention scores:

**Lemma F.22.** *If Induction F.2 holds, then for all iterations $T_{\mathsf{mas},i^\star} \le t < T_{\mathsf{mas},i^\star+1}$:*

*(a) if $i^\star > 1$, then for any $i < i^\star$, we have*

$$\mathbf{Attn}_{L_i}^{(t)} - \mathbf{Attn}_{L_{i^\star}}^{(t)} \ge \Omega(1).$$

*(b) for $i = i^\star$, we have*

$$\frac{1}{C_B} + \Omega\Big(\frac{\log L_i}{\log d}\Big) < \mathbf{Attn}_{L_i}^{(t)} \le 1 - \Omega(1).$$

*(c) if $i = i^\star + 1$, we have*

$$\frac{1}{RC_B} \le \mathbf{Attn}_{L_i}^{(t)} \le \frac{1}{C_B} + O\Big(\frac{\log L_i}{\log d}\Big).$$

*(d) if $i^\star < K-2$, then for any $i > i^\star + 1$, we have*
- *if $L_i = O(1)$, then $1 - C_B\mathbf{Attn}_{L_i}^{(t)} \ge 1 - \frac{1}{0.99R^{i-i^\star-1}+1}$;*
- *else, $\mathbf{Attn}_{L_i}^{(t)} \le O\left(\frac{1}{L_i}\right) = o(1)$.*

This immediately implies the following characterization of the logits:

**Lemma F.23.** *If Induction F.2 holds, then for all iterations $T_{\mathsf{mas},i^\star} \le t < T_{\mathsf{mas},i^\star+1}$:*

*(a) if $i^\star > 1$, then for any $i < i^\star$, we have $(p_{L_i,1}^{(t)})^{L_i} \ge \Omega(\mathcal{J}_{L_{i^\star}}^{(t)}) = \Omega(1)$, and also*

$$1 - p_{L_i,1}^{(t)} \le d^{-\Omega(1)}\Big(1 - p_{L_{i^\star},1}^{(t)}\Big).$$

*(b) for $i = i^\star$, we have*

$$1 - p_{L_i,1}^{(t)} = \Theta\Big(d^{-\left(1 - \frac{1}{e^{q^{(t)}}/(L_i-1)+1}\right)C_B+1}\Big) \ge \Omega(d^{-(1-\Omega(1))C_B+1}).$$

*(c) if $i = i^\star + 1$, we have*

$$p_{L_i,1}^{(t)}/p_{L_i,2}^{(t)} \ge d^{\Omega(1)}.$$

*(d) if $i^\star < K-2$, then for any $i > i^\star + 1$, we have*
- *if $L_i = O(1)$, then $p_{L_i,1}^{(t)} \le O\Big(d^{-\left(1 - \frac{1}{0.99R^{i-i^\star-1}+1}\right)}\Big)$;*
- *else, $p_{L_i,1}^{(t)} \le O\Big(1/d\Big)$.*

The logit conditions also guarantee that for any $i \le i^\star + 1$, we can invoke the gradient characterization in Proposition E.2, and for any $i > i^\star + 1$, we can invoke the gradient characterization in Proposition E.3 and the variant in Lemma F.20. Therefore, we have the following characterization of the gradient:

**Lemma F.24.** *If Induction F.1 holds, then for all iterations $T_{\mathsf{mas},i^\star} \le t < T_{\mathsf{mas},i^\star+1}$,*

*(a) if $i^\star > 1$, then for any $i < i^\star$, we have*

$$\frac{\log d}{d_{\mathsf{p}}} \cdot \frac{1}{d^{C_B-1}} \le \nabla_q \widetilde{\mathcal{J}}_{L_i}^{(t)} \le d^{-\Omega(1)} \nabla_q \widetilde{\mathcal{J}}_{L_{i^\star}}^{(t)}.$$

*(b) for $i = i^\star$, we have*

$$\nabla_q \widetilde{\mathcal{J}}_{L_{i^\star}}^{(t)} = \Theta(1 - p_{L_i,1}^{(t)}) \cdot \frac{\log d}{d_{\mathsf{p}}}.$$

*(c) if $i = i^\star + 1$, we have*

$$\nabla_q \widetilde{\mathcal{J}}_{L_{i^\star+1}}^{(t)} = \Theta\left( \left(1 - p_{L_{i^\star+1},1}^{(t)}\right) \left(p_{L_{i^\star+1},1}^{(t)}\right)^{L_{i^\star+1}} \right) \cdot \frac{\log d}{d_{\mathsf{p}}}.$$

*(d) if $i^\star < K - 2$, then for any $i > i^\star + 1$, we have*

$$|\nabla_q \widetilde{\mathcal{J}}_{L_i}^{(t)}| \leq d^{-\Omega(1)} \nabla_q \widetilde{\mathcal{J}}_{L_{i^\star}}^{(t)}.$$

Thus, to control the gradient of $\mathcal{J}_{\mathrm{mix},R}$ during $[T_{\mathsf{mas},i^\star}, T_{\mathsf{mas},i^\star+1})$, we only need to focus on the gradient of $L_{i^\star}$ and $L_{i^\star+1}$. Similar to Lemma F.11, we have the following lower bound for the period that the reward of $L_{i^\star+1}$ becomes visible.

**Lemma F.25.** *If Induction F.2 holds, then for all iterations $T_{\mathsf{vis},i^\star+1} \leq t < T_{\mathsf{mas},i^\star+1}$, we have*

$$\nabla_q \widetilde{\mathcal{J}}_{L_{i^\star}}^{(t)} + \nabla_q \widetilde{\mathcal{J}}_{L_{i^\star+1}}^{(t)} \geq \Omega\left( \frac{\log d}{L_{i^\star+1} d_{\mathsf{p}}} \right).$$

Now we turn to the period $[T_{\mathsf{mas},i^\star}, T_{\mathsf{vis},i^\star+1})$. The main difference from the large difficulty ratio regime is that in this stage, $L_{i^\star}$ and $L_{i^\star+1}$ will jointly decide a gradient lower bound for $\mathcal{J}_{\mathrm{mix},R}$, which is significantly larger than the one in the long-plateau stage in the large difficulty ratio regime.

**Lemma F.26.** *If Induction F.2 holds, then during $[T_{\mathsf{mas},i^\star}, T_{\mathsf{vis},i^\star+1})$, we have*

$$\nabla_q \widetilde{\mathcal{J}}_{L_{i^\star}}^{(t)} + \nabla_q \widetilde{\mathcal{J}}_{L_{i^\star+1}}^{(t)} \geq \Omega\left( d^{-\frac{RC_B}{R+C_B}+1} \right) \cdot \frac{\log d}{d_{\mathsf{p}}}.$$

*Proof.* Notice that during $[T_{\mathsf{mas},i^\star}, T_{\mathsf{vis},i^\star+1})$, by Lemma F.24, $\nabla_q \widetilde{\mathcal{J}}_{L_{i^\star}}^{(t)}$ is dominated by the term $1 - p_{L_{i^\star},1}^{(t)}$. On the other hand, for $L_{i^\star+1}$, by Lemma F.22 and Lemma F.24, we have $\nabla_q \widetilde{\mathcal{J}}_{L_{i^\star+1}}^{(t)}$ firstly dominated by the term $(p_{L_{i^\star+1},1}^{(t)})^{L_{i^\star+1}}$, which will increase as $q$ increases, and then by the term $(1 - p_{L_{i^\star+1},1}^{(t)})$ when $(p_{L_{i^\star+1},1}^{(t)})^{L_{i^\star+1}}$ reaches the constant level, and $(1 - p_{L_{i^\star+1},1}^{(t)})$ is lower bounded by $\Omega(1/L_{i^\star+1})$. Therefore, to lower bound the gradient summation, we only need to find the time $t$ when $(p_{L_{i^\star+1},1}^{(t)})^{L_{i^\star+1}}$ reaches the same level as $1 - p_{L_{i^\star},1}^{(t)}$. Thus, consider

$$C_B \cdot \mathbf{Attn}_{L_{i^\star}} - 1 = L_{i^\star+1}(1 - C_B \cdot \mathbf{Attn}_{L_{i^\star+1}}),$$

which can be rewritten as

$$C_B \frac{e^q}{e^q + L_{i^\star} - 1} - 1 = L_{i^\star+1}\left(1 - C_B \frac{e^q}{e^q + L_{i^\star+1} - 1}\right) \tag{59}$$

which is a quadratic equation in $e^q$. Denoting $W_L(x) = C_B \frac{x}{x+L-1}$, then solving Equation (59) is equivalent to finding the solution $x^\star$ of $W_{L_{i^\star}}(x^\star) - 1 = L_{i^\star+1}(1 - W_{L_{i^\star+1}}(x^\star))$. Consider the point $x_0 = \frac{L_{i^\star+1}-1}{C_B-1}$. Note that $1 - W_{L_{i^\star+1}}(x_0) = 0$ and $W_L(x)$ is monotonically increasing. Thus $x^\star < x_0$. Hence,

$$W_{L_{i^\star}}(x^\star) - 1 \leq W_{L_{i^\star}}(x_0) - 1 = \frac{C_B \frac{L_{i^\star+1}-1}{L_{i^\star}-1}}{\frac{L_{i^\star+1}-1}{L_{i^\star}-1} + (C_B - 1)} - 1.$$

Hence,

$$\nabla_q \widetilde{\mathcal{J}}_{L_{i^\star}}^{(t)} + \nabla_q \widetilde{\mathcal{J}}_{L_{i^\star+1}}^{(t)} \geq \Omega\left( d^{W_{L_{i^\star}}(x^\star)-1} \right) \cdot \frac{\log d}{d_{\mathsf{p}}} \geq \left( d^{-C_B(1 - \frac{C_B}{R+C_B})-1} \right) \cdot \frac{\log d}{d_{\mathsf{p}}}.$$

$\square$

Putting everything together, we can then characterize the relay behaviour happening during the transition period $[T_{\mathsf{mas},i^\star}, T_{\mathsf{mas},i^\star+1})$.

**Lemma F.27.** *Induction F.1 holds through* $[T_{\mathsf{mas},i^\star}, T_{\mathsf{mas},i^\star+1})$, *where* $T_{\mathsf{mas},i^\star+1} = T_{\mathsf{mas},i^\star} + O\left(\frac{d^{C_B-1}Kd_{\mathsf{p}}\log R}{\eta i^\star \log d}\right)$

*(a) the reward of* $J_{L_{i^\star+1}}$ *only saturates below* $0.01$ *for a time period of at most*

$$\mathcal{T}_k \leq O\left(\frac{d^{\frac{RC_B}{R+C_B}-1}Kd_{\mathsf{p}}}{\eta \log d}\right) \cdot \log R.$$

*(b)* $T_{\mathsf{mas},i^\star+1} - T_{\mathsf{vis},i^\star+1} \leq O(\frac{L_{i^\star+1}d_{\mathsf{p}}K}{\eta \log d})$.

*Proof.* The proof is straightforward by Lemma F.25 and Lemma F.26 and the fact that $q^{(t)}$ changes $\Theta(\log R)$ during $[T_{\mathsf{mas},i^\star}, T_{\mathsf{vis},i^\star+1})$ and $O(1)$ during $[T_{\mathsf{vis},i^\star+1}, T_{\mathsf{mas},i^\star+1})$ due to Lemma F.2. $\qquad\square$

### F.2.4. PROOF OF THEOREM 4.2 AND COROLLARY 4.2

*Proof.* Theorem 4.2 follows immediately from Lemmas F.25 and F.27 together with the bound $K = O(\log d)$.

For Corollary 4.2, we apply Lemma F.27 iteratively for $K-2$ transitions (from horizon 1 up to horizon $K-1$):

$$T_{\mathsf{mas},K-1} - T_{\mathsf{mas},1} \leq O\left(\frac{d^{\frac{RC_B}{R+C_B}-1}Kd_{\mathsf{p}}}{\eta \log d}\right) \cdot (K-2)\log R \;+\; O\left(\frac{d_{\mathsf{p}}K}{\eta \log d}\right) \cdot L_1\left(\frac{R^{K-1}-1}{R-1}\right)$$

$$\leq \widetilde{O}\left(\frac{d_{\mathsf{p}}}{\eta}\right) \cdot d^{\frac{RC_B}{R+C_B}-1} \;+\; \widetilde{O}\left(\frac{d_{\mathsf{p}}}{\eta}\right) \cdot L_{\max},$$

where the last inequality uses $K = O(\log d)$ and $R^{K-1} = O(L_{\max})$. Combining this with the condition $L_{\max} = O(d^{c_x})$ and $c_x < \frac{2C_B}{2+C_B} \leq \frac{RC_B}{R+C_B}$, we obtain

$$T_{\mathsf{mas},K-1} - T_{\mathsf{mas},1} \leq \widetilde{O}\left(\frac{d_{\mathsf{p}}}{\eta}\right) \cdot d^{\frac{RC_B}{R+C_B}-1}.$$

Finally, by Lemma F.21, the time spent in the warm-up stage is negligible compared to $T_{\mathsf{mas},K-1} - T_{\mathsf{mas},1}$. Moreover, we can bound the final step by $T_{\mathsf{mas},K} - T_{\mathsf{mas},K-1} \leq O\left(T_{\mathsf{mas},K-1} - T_{\mathsf{mas},K-2}\right)$. This completes the proof. $\qquad\square$

