# OpenReview forum: "On the Emergence of Implicit Curriculum in RLVR Learning Dynamics"
_ICML.cc/2026/Conference — ICML 2026 regular_

### Official Review · Reviewer_d4gn · 2026-03-11

**Soundness:** 3
**Presentation:** 3
**Significance:** 2
**Originality:** 2
**Overall Recommendation:** 2
**Confidence:** 5

**Summary:**

This paper studies the learning dynamics of outcome-based RLVR on multi-step compositional reasoning tasks. The authors formalize reasoning chains as sequential group actions of length $L$, and show a sharp horizon-dependent phenomenon: when training on a fixed difficulty, short chains are learnable in polynomial time, while long chains suffer from exponentially small policy gradients, yielding severe plateaus. To overcome this barrier, they analyze mixed-difficulty training over lengths $\{L_1,\dots,L_K\}$ and prove that it can keep optimization at the "edge of competence," enabling progressive skill acquisition. Their key theoretical result characterizes the normalized gradient for length $L$ as scaling like $(\Delta_L)^L(1-\Delta_L)$, where $\Delta_L$ is an effective per-step margin, which explains both the exponential difficulty of long-horizon learning and the sudden transition once internal features become sufficiently sharpened. Depending on the spacing ratio $R$ between adjacent difficulty levels, the theory predicts two distinct regimes: large gaps induce grokking-like long plateaus followed by abrupt jumps, while smooth curricula produce a relay effect in which adjacent difficulty levels are learned progressively. Synthetic experiments on group-composition tasks validate these predictions and show that mixed-length RL training substantially accelerates long-horizon reasoning.

**Compliance With Llm Reviewing Policy:**

Affirmed.

**Final Justification:**

The authors responses strengthen my negative views on their contributions. They decline to admit their proof artifacts and presentation issues.

Sor far, the current result implies that $d$ must grow with the target test error (or population reward, whatever) $\epsilon$ at a rate on the order of poly$(\epsilon^{-1})$, which is highly undesirable. For a fixed target test error, one should not need the dimension $d$ to grow with it. This is not intrinsic even to the specific task, but the limitation of the proof. Fixing this crucial issues require substantial revisions. Also, I maintain my point that for readers' convenience, comparing SFT and RL should share the same test error/accuracy for presentation, where population SFT loss or reward objective can also be shown.

As I already emphasized, the notion of the difficulty ratio is not explicitly introduced and must be inferred from context. The text states, "Let us choose difficulty ratio $R > 1$ ... Define a set of horizons ...", but at that point the reader has not yet been given a direct definition of **what "difficulty ratio" means**. The readers only infer it after parsing the recursive definition of $L_k$ themselves. Indeed the view that this informal way of definition hurts clarity is shared by another reviewer, but the author still declined to admit that this is a failure. The authors could simply acknowledge this presentation issue instead of asking me to reread the section and restating my point. They fail to do it twice.

Hiding an $o(·)$ term inside $\Omega(·)$ is not a professional way. In Theorem F.1, the term $1-o(1)$ is paraphrased as being "extremely close to 1" in the author's response, which is not appropriate. In particular, the closeness to $1$ appears to depend on the difficulty ratio $R$. If the target $1-o(1)$  is $1-a$, then $R$ must scale at least on the order of $C_B a^{-1}$, up to constants determined by the exact exponent. So the result imposes a nontrivial lower bound on $R$, not merely that $R$ be $\omega(1)$. This is a false claim and another proof artifact, along with presentation issues.

Again, the cited works study pretraining rather than the post-training setting considered here. The authors appear to assume that pretrained attention carries no task-relevant prior knowledge for the task class being fine-tuned. This means that rather than theoretical investigations over empirically-grounded finetuning to show underlying principles, the authors consider tractability first, impairing the practical relevance.

The task in [1] is related but structurally different, but the authors falsely claim they consider a harder regime beyond [1] in their response. [1] studies a permutation composition and explicitly characterizes its statistical difficulty via an SQ. By contrast, the toy problem constructed by the author introduces paired anchors $x$ for attention, which simplify the task and welcome Fourier techniques, much differing from [1]. The resulting theory is limited in scope and claiming their natural Fourier analyses on this toy as a good contribution is not appropriate.

**Key Questions For Authors:**

1. Since the authors aim to consider finetuning, Assumption 3.4 might need further justification. What did this mean for a pretrained attention?
2. Tian 2025 also consider utilising Fourier-transform techniques for their grokking study. Albeit they consider pretraining, your task, with zero initialisation for attention, is essentially similar to a pre-training scheme, despite considering a different loss. Please cast a more comprehensive techinical comparison with this work, especially the root causes of grokking. Importantly, would the "leak of gradient", which is more close to real dynamics validated by their experiments, still play a role?

**Limitations:**

The author failed to cast a comprehensive discussion of the limitation of their work.

**Strengths And Weaknesses:**

**Strength**

The paper is clearly written and well organized. The theoretical explanations of the grokking and relay phenomena are relatively easy to follow. The use of Fourier decomposition for analyzing group-based structures is natural.

**Weakness**

1. It is difficult to assess the statistical difficulty of the concrete reasoning task considered in this work. The task appears to be specifically designed to facilitate Fourier-basis analysis, which the authors highlight as a technical contribution. However, this analytical technique is unlikely to generalize to problems with broader or different structural properties, hugely limitating its contribution to the community.

2. The lower bound on the target attention precision $\epsilon$ in Theorem 4.1 is unusual for learning theory. In particular, $\log(d)^{-1}$ is only of logarithmic order and does not appear sufficiently small. In addition, $\epsilon$ is typically used to denote arbitarily small test error rather than attention precision, which is less of interest. Intuitively, gradients often vanish when attention becomes highly concentrated, yet the authors appear to avoid this nonlinear behavior in the dynamics by imposing a target attention precision. This simplification trick impairs the theory's impact.

3. The iteration lower bounds in Theorem 4.2 (for SFT) and Theorem 5.1 (for mixed-difficulty training) still contain an exponential dependency $d^{C_B}$. Intuitively, the time complexity for SFT and mixed-difficulty training should ideally avoid such dependence on $C_B$.

4. Using the notation $L_{max}$ to represent the dimensionality of the positional encoding space is potentially confusing, since it naturally suggests a sequence length rather than a cardinality. It might be clearer to directly use the order $d^{c_x}$ in Assumption 3.2, or alternatively denote this quantity as $d_{pos}$.

5. The definition of the difficulty ratio is not explicitly introduced and must be inferred from context. It would improve clarity to first provide a formal definition of the "difficulty ratio".

6. Theorem 4.2 is formulated in terms of a loss function, while Theorem 5.1 is expressed using the objective $J$. For clarity and comparability, it would be preferable to use either the loss function or the reward objective consistently.

7. The appearance of $1 - o(1)$ inside $\Omega(\cdot)$ in Theorem 5.2 is somewhat unusual and may benefit from clarification.

8. The simulation results could be moved into a section of the main paper rather than being placed entirely in the appendix.

9. The theoretical analysis relies on population gradients rather than gradients estimated from finite samples.

---

> ### Author Rebuttal · Authors · 2026-03-31
>
> We thank the reviewer for the detailed feedback. We respectfully disagree with several of the assessments and address each point below.
>
> **Q1. Fourier analysis limits generalizability.**
>
> **A1**. The group-compositional task is not chosen merely for convenience: even its supervised version requires sophisticated analysis [1]. We study the harder RLVR setting, and our Fourier-based technique provides a new approach to analyzing RL-driven compositional reasoning, a problem that has resisted theoretical treatment. We view the framework as a contribution, not a limitation.
>
> **Q2. The target attention precision**
>
> **A2**. We respectfully disagree that this choice limits impact.
> - At the stated precision, the policy gradient is already $1/\text{poly}(d)$ and $J$ reaches $1-1/\text{poly}(d)$, which is operationally meaningful. The analysis also extends to smaller attention errors without modification, but yields no new insight. Presenting results in terms of attention precision is standard in transformer analysis [2–3].
> - We also highlight that we do not avoid nonlinear behavior, on the contrary, analyzing the nonlinear training dynamics of the policy gradient is central to our contribution. The reviewer's concern appears to stem from a misreading of the role of $\epsilon$ in our framework.
>
> **Q3. Exponential dependency $d^{C_B}$ in lower bounds.**
>
> **A3.** We highlight that the exponential dependence is not a drawback of our result, but reflects the intrinsic difficulty of the task and cannot be avoided by choice without altering the problem structure. In our setup, $C_B$ is a constant determined by the MLP's feature structure, and the $d^{C_B}$ factor arises directly from the gradient magnitude at convergence: as the attention sharpens toward the correct token, the gradient signal scales as $d^{-C_B}$, making $d^{C_B}$ iterations necessary for gradient-based methods to accumulate sufficient updates. Thus, this dependence is not an artifact of our analysis, but a direct consequence of the underlying optimization dynamics.
>
> **Q4. Notation $L_{max}$**
>
> **A4**. We retain $L_{\max}$ as it reflects a meaningful relation: maximum task length determines positional encoding dimension (see Assumption 3.2). We will clarify this further in revision.
>
>
> **Q5. definition of the difficulty ratio.**
>
> **A5**.  We would suggest the reviewer check the beginning of Section 5.1 where $R$ is formally defined. If the reviewer finds that a dedicated definition environment would improve clarity, we are happy to add one in the revision.
>
> **Q6. Inconsistent use of loss vs. reward objective.**
>
> **A6**. The objectives differ because SFT minimizes loss while RLVR maximizes reward. This reflects the paradigms themselves, not a notational inconsistency. Unifying them under a single objective would be conceptually misleading.
>
> **Q7. $1-o(1)$ inside $\Omega(\cdot)$.**
>
> **A7**. The $1 - o(1)$ term is used for presentation clarity to indicate that the exponent is extremely close to 1. The exact form appears in the proof of Theorem F.1. We will add a clarifying remark in the revision.
>
> **Q8. Simulation results in appendix.**
>
> **A8**. Thanks for the suggestions. We will move the key simulation results into the main paper in the revision.
>
>
> **Q9. Population gradients.**
>
> **A9**. The use of population gradients is standard in transformer theory [3-4]. Extending to finite samples requires substantial effort to control stochastic terms without additional insight, so we follow this convention and leave it as future work.
>
> **Q10. Justification of Assumption 3.4 for pretrained attention.**
>
> A10. Uniform initialization is natural in our setting: it reflects the absence of prior compositional knowledge, with all components contributing equally. This is also standard in transformer learning theory [3-5]. We will add further justification in the revision.
>
> **Q11. Comparison with Tian (2025).**
>
> **A11**. The two works share only superficial similarities: use of Fourier techniques and grokking-like phenomena. Tian (2025) analyzes gradient leakage in a pretraining setting and provides descriptive insights into grokking dynamics, whereas our work provides end-to-end optimization guarantees for RLVR compositional learning, which is a substantially different theoretical objective. The gradient leakage they identify is specific to their pretraining dynamics, and does not directly transfer to our outcome-reward setting. We therefore do not view a detailed technical comparison as necessary.
>
> ---
>
> **References**：
>
> [1] Wang et al.  Learning Compositional Functions with Transformers from Easy-to-Hard Data
>
> [2] Li et al.  A Theoretical Understanding of shallow Vision Transformers: Learning, Generalization, and Sample Complexity
>
> [3] Nichani et al. How Transformers Learn Causal Structure with Gradient Descent
>
> [4] Zhang et al. Trained transformers learn linear models in-context
>
> [5] Kim et al. Transformers provably solve parity efficiently with a chain of thought

---

> > ### Author Rebuttal · Reviewer_d4gn · 2026-04-01
> >
> > The rebuttal does not address my concerns and falsely claim my point as misreading. Resolving my concerns require significant updates of the paper.
> >
> > > A1
> >
> > The task in [1] is related but structurally different. [1] studies a permutation composition and explicitly characterizes its statistical difficulty via an SQ. By contrast, your construction introduces paired anchors $x$ for attention, which simplify the task and welcome Fourier techniques, much differing from [1]. The resulting theory is limited in scope.
> >
> > > A2
> >
> > The rebuttal does not address my concern and falsely claim my point as misreading.
> >
> > First, in learning theory, $\epsilon$ usually denotes a target test error or accuracy tolerance. Using $\epsilon$ to denote an attention-precision target is not good. Second, having $\epsilon \ge \log(d)^{-1}$ is unsatisfactory. Since $\log(d)$ is only logarithmic.
> >
> > Third, I did not claim that the paper avoids all nonlinearity. My point was narrower: requiring precision $\ge\log(d)^{-1}$ appears to sidestep the gradient-vanishing effect when attention becomes concentrated. This lower bound seems to come from a coarse proof used to lower-bound the gradient, rather than an intrinsic bottleneck of the actual process. In realistic training, attention precision can converge well below the $\log(d)^{-1}$ scale - this lower bound is best viewed as a proof artifact.
> >
> > Finally, although I did not explicitly mention $J$, the current result also implies that $d$ must grow with the target test error $\epsilon$ at a rate on the order of poly$(\epsilon^{-1})$, which is highly undesirable. For a fixed target test error, one should not need the dimension to grow with it. This is not intrinsic even to the specific task, but rather another limitation of the paper.
> >
> > > A3
> >
> > The exponential dependence on $d^{C_B}$ is too strong and is unlikely to be intrinsic. It potentially appears to result from a coarse treatment of the MLP logits. More specifically, the gradient magnitude is bounded using worst-case estimates of the form $d^{C_B}$, which makes the dependence on $C_B$ exponential. This looks more like a proof artifact than a fundamental barrier or a faithful measure of task difficulty.
> >
> > > A4
> >
> > Having the word vocabulary scale with the maximum sequence length is a particular feature of this specially structured task. This makes the notation more task-specific and obscures how complexity would depend on vocabulary cardinality in more realistic settings.
> >
> > > A5
> >
> > My comment was clearly based on the beginning of Section 5.1. As I already emphasized, the notion of the difficulty ratio is not explicitly introduced and must be inferred from context. The text states, "Let us choose difficulty ratio $R > 1$ ... Define a set of horizons ...", but at that point the reader has not yet been given a direct definition of what "difficulty ratio" means. The reader only infers it after parsing the recursive definition of $L_k$. The authors could simply acknowledge this presentation issue instead of asking me to reread the section and restating my point.
> >
> > > A6
> >
> > I disagree. SFT and RLVR may use different training losses, but evaluation should still be based on a common metric. In learning theory, test error or test accuracy is the standard objective, while training losses are just surrogates. The same applies to experiments: people care about comparing test error or accuracy in the same figure. Plotting SFT with decreasing loss and RLVR with increasing reward in one figure is not a meaningful comparison.
> >
> > > A7
> >
> > First, hiding an $o(·)$ term inside $\Omega(·)$ is not a professional way. Second, in Theorem F.1, the term $1-o(1)$ is paraphrased as being "extremely close to 1" in your response, which is not appropriate. In particular, the closeness to $1$ appears to depend on the difficulty ratio $R$. If the target $1-o(1)$  is $1-a$, then $R$ must scale at least on the order of $C_B a^{-1}$, up to constants determined by the exact exponent. So the result imposes a nontrivial lower bound on $R$, not merely that $R$ be $\omega(1)$.
> >
> > > A10
> >
> > This response does not address the concern. The cited works study pretraining rather than the post-training setting considered here. The authors appear to assume that pretrained attention carries no task-relevant prior knowledge for the task class being fine-tuned?
> >
> > > A11
> >
> > As I already noted, and as your own response implicitly reflects, your setting with zero initialization of attention is much closer to a pretraining-style setup, even if the loss function is different. I did not claim that this comparison is mandatory; that is precisely why I raised it as a question rather than a weakness. More importantly, the leakage arises under weight decay, regardless of the loss, and may also be relevant to the grokking in real-world finetuning. This is why a more careful comparison would be valuable, especially to clarify whether that mechanism is more realistic than the one in your analysis. Also, please cite it in ICLR format.

---

> > > ### Author Response · Authors · 2026-04-03
> > >
> > > The concerns raised by the reviewer do not affect the correctness, clarity, or significance of our results. Broadly, they relate to notation and presentation, and to the interpretation of results and technical assumptions. We address these in turn.
> > >
> > > 1. We first address comments concerning the results and technical assumptions, **which do not fully align with the current mainstream understanding in the transformer theory literature.**
> > >
> > >    - **$1/log(d)$ attention precision bound.** The reviewer suggests that this bound may be a proof artifact and implies an undesirable dependence between $d$ and $\epsilon$. This concern appears to arise from how attention precision is related to downstream performance. In our setting, loss/reward scales approximately as $e^{-{attn score} \cdot {mlp feature}}$, so a $1/log(d)$ precision in attention is already sufficient to guarantee $1/{poly}(d)$ convergence in loss or reward. The bound is therefore sufficient for strong downstream guarantees. We present convergence in attention primarily to illustrate the mechanism by which the transformer performs the task.
> > >     - **Exponential dependence on $C_B$.** The reviewer notes that the $d^{C_B}$ dependence may be overly strong and  attributes it to worst-case analysis. As clarified previously, $C_B$ is treated as a constant with respect to $d$, so the dependence is polynomial in $d$ for any fixed task. In addition, this factor arises from a gradient lower bound of order $d^{-C_B}$. There are settings where gradients scale at this order, in which case slower convergence is expected. The current bound therefore reflects this behavior.
> > >      - **The attention initialization and comparison to pretraining.** The reviewer suggests that our zero-initialized attention setting is closer to a pretraining-style setup and that comparison with prior work would be valuable. We agree that studying richer initialization assumptions is an interesting direction for future work. At the same time, we adopt uniform (zero) attention initialization because it provides a clean and analytically tractable setting that is uninformative for compositional learning. We hope this serves as a useful starting point for future extensions.
> > >
> > > 2. We now turn to the comments related to notation and presentation. **These comments largely reflect strong stylistic preferences and differ from conventions commonly adopted in the transformer theory literature.**
> > >
> > >     - **The use of $\epsilon$ for attention precision.** The reviewer states that "using $\epsilon$ to denote an attention-precision target is not good" since $\epsilon$ typically denotes test error in learning theory. We note that $\epsilon$ is widely used across theoretical disciplines, including optimization and theoretical computer science, to denote generic notions of error or tolerance. In our setting, it represents the gap between learned and optimal attention, which is an error quantity and is clear from context. Using an alternative symbol (e.g., $\zeta$ or $\iota$) would not affect any result or its interpretation.
> > >
> > >    - **The definition of the difficulty ratio.** The reviewer notes that the difficulty ratio "is not explicitly introduced and must be inferred from context." In our paper, the sentence “Let $R > 1$ be a difficulty ratio …” introduces $R$ as a scalar parameter that is then used to define the mixed-difficulty distribution. This follows standard conventions in ML theory, where quantities are introduced upon first use and then used to parameterize subsequent definitions. We are happy to add a brief clarifying sentence if it improves readability.
> > >
> > >    - **Using reward/loss vs. test error as the reported metric.** The reviewer argues that evaluation should be based on a common metric such as test error. In our setting, the reported loss and reward are population-level quantities, not empirical surrogates, and they directly imply test error optimality. Moreover, different areas of ML theory routinely adopt different primary metrics (e.g., reward in RL, training loss in optimization), depending on the natural objective of the problem. Reporting loss (for SFT) and reward (for RLVR) is therefore consistent with standard practice.
> > >
> > > In summary, we hope this resolves the reviewer’s concerns in notation preferences and clarifies the interpretation of our results.

---

### Official Review · Reviewer_e4v1 · 2026-03-12

**Soundness:** 3
**Presentation:** 2
**Significance:** 2
**Originality:** 3
**Overall Recommendation:** 4
**Confidence:** 1

**Summary:**

I don’t understand this work at all. So the whole review, including the summary, is based on my very limited understanding.

This work explains the training dynamics of Reinforcement Learning with Verifiable Rewards (RLVR) from a theoretical perspective. It proves that when the model is training with problems with varying difficulties, there will be prolonged plateaus before progress recurs. When the model is training with increasingly difficult problems, the growth will be smoother.

**Compliance With Llm Reviewing Policy:**

Affirmed.

**Final Justification:**

My initial understanding of this work was very shallow and limited, which prevented me from evaluating this paper properly. By reading the reviews, revisiting the paper, and engaging in the rebuttal, I established an entry-level understanding of this work, which changed my view on it.

The main strength of this work is proving theoretical explanations and proofs for RL training behaviors (such as grokking-like phase transitions and smooth relay effects), considering factors like attention concentration and skill composition. But I think the major limitation is the whole analysis was done based on the simplified LEGO model. While the simplified model isolates skill composition from other confounding factors and provides a clean analysis of the model’s learning behaviors, it remains unclear to what extent the findings can be generalized to real reasoning tasks.

I agree with other reviewers that this work has theoretical value, but with limited knowledge of the theoretical field, I am not sure how big the value is. So, my recommendation is based only on the values I am able to assess and may miss important points that were not deeply discussed in the review period. Overall, I am on the border line.

**Key Questions For Authors:**

1. What is high-level intuition behind the theorems? Can you explain the high-level idea in English, not symbolic language.

**Limitations:**

- This paper is very hard to understand.
- There is no solid proof for any of the major theorems.

**Strengths And Weaknesses:**

Strengths:
- I don’t see any solid strength because I don’t understand the work at all.

Weaknesses:
- There are too many symbols, making it hard to follow the logic.
- I don’t see any solid proof for the major theorems, such as theorems 4.1, 5.1, 5.2, and 5.3.
- A very important assumption of this work, “Prior analysis (Huang et al., 2025b) shows that an MLP can learn such a feature-separated structure under supervised training with suitable initialization.” is currently in arxiv state. I am not sure whether this assumption is solid. If not, this is not a solid assumption for the capability of MLP. That says the statement of “Under the above assumption, the pre-trained MLP contains all the atomic skills for one-step transitions” may not hold.

---

> ### Author Rebuttal · Authors · 2026-03-31
>
> We note with concern that the reviewer explicitly states, "I don't understand this work at all" and assigns a confidence score of 1, yet provides a rejection recommendation with scores of 1 across all dimensions. We briefly address the specific points for completeness.
>
>
> **Q1. I don’t see any solid proof for the major theorems**
>
> **A1**. The proofs for all major theorems are provided in full in the appendix. We are unclear what the reviewer means by "no solid proof."
>
> **Q2. Assumption regarding Huang et al.**
>
> **A2**.  This is an explicit modeling assumption stated in Sec. 3.3, not an unverified claim. Prior work (Huang et al., NeurIPS 2025) shows that supervised training can induce such feature-separated structures in MLPs.  Importantly, our results do not depend on re-proving this property, but on characterizing what follows if such a structure exists. This is standard in theoretical ML: we isolate a minimal and interpretable setting in which compositional learning can be rigorously analyzed.
>
> **Q3. What is high-level intuition behind the theorems? Can you explain the high-level idea in English, not symbolic language.**
>
> **A3**. Please read the abstract and introduction, which are written in plain English. It should go without saying, though we will emphasize here, that presenting technical results in mathematics (what the reviewer referred to as “symbolic language”) is essential in putting these “high-level idea” on rigorous footing.

---

> > ### Author Rebuttal · Reviewer_e4v1 · 2026-04-03
> >
> > I apologize for being unable to evaluate the merit of this work due to my very limit knowledge. By reading other reviewers’ comments and the rebuttals, I established an initial but still shallow understanding of this work. With such a shallow understanding I read the paper again as well as serval key references to evaluate its value. The following are mainly questions that I think would help outliers like me better understand this work. My recommendation is mainly based on my limited understanding of the interactions between the authors and other reviewers.
> >
> > Questions:
> > 1. LEGO is a lot simpler than real reasoning tasks such as AMC and AIME. Do the theorems established in this work apply to real reasoning tasks?
> > 2. In 3.3. Pretrained Atomic Skills, why the skills can be represented with equation 5a-c. Does the feature response implicitly represent atomic skills? The Yuan et al. work showed that LLMs can composite skills in general reasoning tasks, but did not provide a formalization of the skills.
> > 3. Theorem 4.1 proves that RL is optimal for learning multi-step compositional reasoning with short-horizon. But I don’t understand what mechanism of transformer guarantees this optimality. Is it that the attention properly captures the connection between different atomic skills? Does the appendix explain the reason? If so, which part?
> > 4. Theorem 4.2 seems to prove that SFT works well for long reasoning chains. I think the main reason is it has immediate rewards as more accurate supervision signals. But I don’t understand how SFT combines atomic skills, also through attention?
> > 5. In 5.1. Easy-to-Hard Mixture. How is the difficulty ratio defined. Is it just the average accuracy of responses? Additionally, how easy-to-hard mixtures help smooth the learning curve, by providing more stable learning signals?
> >
> > By reading the paper carefully again, I believe I established a slightly deeper understanding of problem being solved and proved. I think the main limitation of this work is the whole analysis is done on the LEGO problem, which is a lot simpler than real reasoning problems. But its contribution to reinforcement learning is still clear. It explains and proves RL learning behaviors from a mathematical perspective, providing solid ground for future research.
> >
> > Given my limited knowledge in this area, I am not able to fully evaluate the detailed proof provided in the appendix. So I adjust my scores to provide a conservative evaluation with very low confidence, which may not reflect the true merit of this work. Scores will be adjusted after more discussions, which I believe will further deepen my understanding.

---

> > > ### Author Response · Authors · 2026-04-03
> > >
> > > We appreciate the effort you made to revisit the paper and engage with both the rebuttal and the broader discussion. Below, we address your questions and clarify the key points.
> > >
> > > **Q1. Do the theorems established in this work apply to real reasoning tasks**
> > >
> > > **A1.** We acknowledge that our problem setting, being an abstracted and simplified model of compositional reasoning, cannot capture all aspects of real-world reasoning.  In practice, real-world reasoning often involves complex compositions of heterogeneous skills, including planning, search, and knowledge manipulation, which go beyond the atomic and homogeneous skills considered in our framework. Our goal is to provide a principled foundation by isolating and formalizing the compositional structure of reasoning. By abstracting skills as atomic and homogeneous, we enable theoretical analysis of learning dynamics that would otherwise be intractable. While our results do not fully explain RLVR’s behavior in real-world reasoning tasks, we believe they capture fundamental aspects of compositionality and can serve as a useful basis for future work toward more realistic settings.
> > >
> > > **Q2. Does the feature response implicitly represent atomic skills?**
> > >
> > > **A2.** Yes, in our formulation, the feature responses can be interpreted as implicitly representing atomic skills. Each transformation (or action) corresponding to a group element $g$ applied to a state $y$ can be viewed as a single step of reasoning, i.e., the application of an atomic skill. In real-world settings, this could correspond to incrementally updating a chessboard state given a sequence of moves, or tracing the execution of a program based on function definitions and inputs. While this abstraction does not capture the full heterogeneity of real-world reasoning, we believe it isolates a fundamental aspect of compositionality and provides a principled foundation for analyzing more complex settings.
> > >
> > > **Q3. Is it that the attention properly captures the connection between different atomic skills?**
> > >
> > > **A3.**  The key mechanism is attention concentration. Concretely, attention concentration ensures that, at each reasoning step, the model focuses on the relevant parts of the context, allowing it to correctly associate and execute atomic skills in sequence. Without this property, the model would be overwhelmed by ambiguous intermediate information and fail to carry out multi-step reasoning reliably. This mechanism is explained in the paragraph “How does the transformer reason sequentially?” at the end of Section 3.3, and is also the main technical objective of the proof. In our setting, since atomic skills are independent and homogeneous, attention concentration is sufficient to retrieve and organize them into a valid reasoning chain, which enables effective credit assignment in the short-horizon regime and leads to the optimality result.
> > >
> > > **Q4. I don’t understand how SFT combines atomic skills, also through attention?**
> > >
> > > **A4.** In our setting, both RL and SFT learn the same underlying feature representations (i.e., atomic skills), as we avoid introducing factors that would cause them to diverge. Moreover, SFT combines atomic skills through the same attention-based mechanism as RL. Its advantage lies not in a different composition mechanism, but in providing dense, step-wise supervision, which alleviates long-horizon credit assignment. Understanding how different training paradigms may lead to different learned features is an important direction, but is beyond the scope of our current theoretical framework.
> > >
> > > **Q5. In 5.1. Easy-to-Hard Mixture. How is the difficulty ratio defined. Is it just the average accuracy of responses? Additionally, how easy-to-hard mixtures help smooth the learning curve, by providing more stable learning signals?**
> > >
> > > **A5.** The difficulty ratio is basically $L_{n+1}/L_n$, that is, the ratio between the horizons of two consecutive difficulty levels. It characterizes the gap between neighboring levels: a larger ratio indicates a wider gap. Easy-to-hard mixtures smooth the learning process by enabling positive transfer from easier to harder problems. The effectiveness of this transfer depends on how gradual the difficulty progression is. Specifically, smaller difficulty ratios lead to smoother transitions and more stable learning signals, as also illustrated in Figure 1.

---

### Official Review · Reviewer_B2kt · 2026-03-13

**Soundness:** 4
**Presentation:** 4
**Significance:** 2
**Originality:** 2
**Overall Recommendation:** 5
**Confidence:** 4

**Summary:**

The paper studies training dynamics of RLVR for a compositional reasoning task, learning to compose group actions. It is shown that policy gradient can learn short-horizon compositions but have exponential difficulty learning long-horizon compositions. In comparison, poly time convergence is proved on easy-to-hard mixture data. Moreover, a grokking-type phase transition is characterized depending on the mixture difficulty gap.

**Compliance With Llm Reviewing Policy:**

Affirmed.

**Final Justification:**

The authors have satisfied my remaining concerns throughly in the rebuttal, therefore I recommend the paper for acceptance.

**Key Questions For Authors:**

See weaknesses.

**Limitations:**

See weaknesses.

**Strengths And Weaknesses:**

**Strengths**

* The paper studies an interesting and timely problem, to understand compositional learning in post-training.
* The studied group action task, while stylized is a nice toy model which captures various aspects of RL training and goes beyond simpler tasks like parity which have been studied in similar settings.
* The theoretical exposition is rigorous and neatly explains the interplay of horizon and difficulty in compositional learning.
* The paper is overall well-written and the theory seems to be correct and interesting.

**Weaknesses**

* The overall story seems very similar to the paper [1]. While not framed as a study of RLVR, this paper also studies a compositional task for composing elements of the symmetric group, and analyzes the learning dynamics of the attention layers of a transformer model on both easy-to-hard curriculum and mixture data. Moreover, their $k$-hop task seems to be fundamentally more difficult since it is conjectured to require $\log k$ layers, and they also prove a rigorous query lower bound. The main difference here seems to be while [1] uses a doubling schedule for the hop horizon for their positive result, the current paper uses an arbitrary ratio $R$ and analyzes the transition times on more depth. Nonetheless, the main intuition is very similar, I would expect at least a detailed comparison to this work.

* Only synthetic experiments are provided. It would be nice to verify the observations regarding grokking and relay on real-world reasoning tasks at least qualitatively. In particular, the observations regarding the relay effect are new and interesting but currently it is unclear if it is only a proof artifact.

* Theorem 4.1 requires roughly $d^{O(C_B)}$ iterations to reach optimal reward when $L<C_B$, while Proposition 4.1 gives a bound $d^{-\Omega(L)}$ for the gradient signal for $L>2C_B$. From this, it seems that the learning difficulty is just $d^{\Theta(L)}$ in general, similar to results in subset parity or multi-index models. Why is this a "critical horizon"? Why is there a gap for $C_B<L<2C_B$, and is the analysis fundamentally different in the two regimes?

* Similarly, why does the phase transition analysis need to differentiate two regimes for $R$ (Theorem 5.1 and 5.2)? They seem to tell mostly the same story, i.e., $O(\log R)$ time to proceed to the next plateau. Minor: there seems to be a typo for the formula of $T_{grok}$ in Theorem 5.2(a).

[1] Wang, Z., et al., 2025. Learning Compositional Functions with Transformers from Easy-to-Hard Data.

---

> ### Author Rebuttal · Authors · 2026-03-31
>
> We thank the reviewer for the constructive feedback. We address each concern in turn below.
>
> **Q1. The overall story seems very similar to the paper [1].**
>
> **A1.**   We thank the reviewer for pointing out this related work. While both papers study compositional learning in transformers, the settings are fundamentally different.
> [1] study a specially constructed O(log⁡k)-layer transformer for the kkk-hop task without chain-of-thought reasoning, and analyze its learnability under supervised, layer-wise training. While they also provide gradient-based learning results, their algorithm rely on specially designed guided supervision, and their purpose was to show how the prescribed structure can be learned progressively across layers. In contrast, we study **end-to-end RLVR with outcome-only reward after long CoT reasoning, with no intermediate supervision or prescribed decomposition**. In our case, compositional structure must be learned from sparse final-answer feedback alone. Our theory characterizes precisely this emergence mechanism, which is orthogonal to and not addressed by Wang et al.
> Thank you for bringing this question. We will include a more detailed comparison in the revision.
>
> **Q2. Only synthetic experiments are provided.**
>
> **A2.** We thank the reviewer for the suggestion. We conducted an additional experiment on multi-digit multiplication using Qwen2.5-1.5B-Instruct, where digit count serves as the difficulty level.
>
> - **Setup.** We fine-tune with GRPO on N-digit $\times$ N-digit multiplication (exact match reward). Pre-training shows the competence edge at 3-digit (3-digit: 8%, 4-digit+: $\approx$0%), used as the starting level.
>
> - **Results.**
>   - *Mixed-difficulty:*
> | Setting | Step | 3-digit | 4-digit | 5-digit | 6-digit |
> |---|---|---|---|---|---|
> | M1: 3+4+5 ($R\approx1.33$) | 0 | 8% | 1% | 0% | 0% |
> |  | 500 | 91% | 76% | 40% | 11% |
> |  | 1875 | 94% | 82% | 61% | 15% |
> | M2: 3+6+12 ($R=2$) | 0 | 8% | 1% | 0% | 0% |
> |  | 500 | 91% | 47% | 2% | 0% |
> |  | 1875 | 93% | 25% | 0% | 0% |
>    - *Single-difficulty (eval on 4-digit):*
> | Setting | Step 0 | Step 100 | Step 200 | Step 1875 |
> |---|---|---|---|---|
> | 4-digit only | 1% | 2% | 75% | 88% |
> | 5/6-digit only | 0% | 0% | 0% | 0%  |
>
> We highlight three observations: (1) **Relay effect:**   in M1, sequential propagation ($3\text{d}\rightarrow4\text{d}\rightarrow5\text{d}$) with OOD generalization to 6-digit (15%). (2) **Large $R$ breaks relay:** in M2, transfer fails (4-digit 47%→25%, 5-digit+ 0%). (3) **Grokking:** in single-difficulty, 4-digit shows a plateau-then-jump (2%→75%).  These observations qualitatively match our theoretical predictions and suggest that the relay effect and grokking behavior arise in real language models, rather than being artifacts of the proof.  We plan to include additional experiments in future revisions.
>
> **Q3.**  Why call $C_B$ a “critical horizon” ?
>
> **A3.** The key point is that $C_B$ marks a change in gradient dominance, rather than simply scaling. For $L<C_B$, the fully correct trajectory provides a dominant, structured gradient signal, enabling precise characterization and efficient learning (Theorem 4.1). In contrast, for $L>2C_B$, this dominance breaks down: contributions from correct trajectories become exponentially small in $L$, and learning is driven by noise rather than informative signal (Proposition 4.1).  Thus, the distinction is not only quantitative (e.g., $d^{\Theta(L)}$), but qualitative in terms of which components dominate the gradient, leading to different analyses. This is why $C_B$ serves as a “critical horizon.” The intermediate regime $C_B<L<2C_B$ lies at the boundary where neither regime fully applies, and current analysis does not yield tight guarantees. We thank the reviewer and will clarify this in the revision.
>
>
> **Q4. Why does the phase transition analysis need to differentiate two regimes for $R$?**
>
> **A4.** The two regimes reflect a real difference in plateau scaling, rather than a purely technical distinction. This is also different in nature from the phase transition at  C_B. For $R = O(1)$, the longest plateau can be bounded by $d^{C_B-\Omega(1)}$, i.e., it remains separated from the critical timescale $d^{C_B}$ by a fixed polynomial gap. For larger $R$, this gap disappears: the bound becomes $d^{\,C_B-o(1)}$, so the longest plateau can approach the critical scale arbitrarily closely. This difference in time scales is driven by the underlying gradient dynamics: as R increases, the gradient signal from the current stage diminishes before the next stage becomes dominant, leading to near-critical slowdown. This separation is further amplified under finite numerical precision: for sufficiently large R, gradient magnitudes can fall below machine precision, causing optimization to stall entirely. We will add a clarifying remark in the revision. We also thank the reviewer for catching the typo, which we will correct in the revision.

---

> > ### Author Rebuttal · Reviewer_B2kt · 2026-03-31
> >
> > I believe the contribution is solid and the authors have clarified my questions.

---

### Official Review · Reviewer_rr3d · 2026-03-16

**Soundness:** 3
**Presentation:** 3
**Significance:** 3
**Originality:** 3
**Overall Recommendation:** 5
**Confidence:** 3

**Summary:**

This paper develops a theoretical framework to understand the learning dynamics of RLVR. The authors study a minimalist transformer (one attention layer + fixed pretrained MLP) on a group-theory-based multi-step compositional reasoning task, where the MLP already implements atomic single-step transitions and the attention layer must learn to compose them over L steps. The paper establishes three main theoretical results: (1) For short horizons (L <= C_B), REINFORCE-style policy gradients provably learn L-step composition with polynomial time convergence, while beyond the critical horizon, an exponentially flat gradient landscape emerges; (2) SFT can overcome this critical horizon via intermediate supervision; (3) Under mixed-difficulty data distributions, the difficulty ratio R = L_{k+1}/L_k determines whether learning exhibits grokking-like phase transitions (large R) or smooth relay effects (small R) where progress on easier problems bootstraps harder ones. The theoretical results are developed using novel Fourier analysis techniques on finite groups, and are validated through synthetic experiments on cyclic group Z_96.

**Compliance With Llm Reviewing Policy:**

Affirmed.

**Key Questions For Authors:**

see weaknesses

**Limitations:**

yes

**Strengths And Weaknesses:**

# Strengths

1. The paper asks a fundamental question: why does RLVR primarily improve performance near the model's edge of competence? and provides a rigorous theoretical answer. This is one of the most important open questions in the RLVR community, and the paper contributes meaningfully to our understanding.

2. Theorem 4.1 provides the first provable guarantee that outcome-based RL can learn multi-step compositional reasoning, while Proposition 4.1 quantifies the exponentially flat gradient region beyond the critical horizon. The contrast between RL and SFT (Theorem 4.2) in overcoming this barrier is clean and practically relevant. The characterization of grokking vs. relay dynamics (Theorems 5.2 and 5.3) through the difficulty ratio R is particularly elegant and directly connects to empirical observations.

3.  The key insight of decomposing trajectory level success events into convolutions on the group structure (Section 6.3), enabling tractable gradient estimation through spectral properties, is creative and could find applications beyond this specific setting.

4.The reduction to two scalar quantities (q, r) via Lemma 6.2 simplifies the dynamics elegantly. The paper does a good job of providing intuition alongside formal statements.

5. The synthetic experiments (Figure 2, 3) clearly validate the theoretical predictions: short-horizon learning vs. long-horizon plateau (Message 1), grokking with large R (Message 2), and smooth relay with small R (Message 3). The attention alignment measurements (Figure 3) confirm the predicted attention concentration pattern from Theorem 4.1(b).

# Weaknesses

1. The theoretical framework makes several strong simplifying assumptions that are far from modern LLM practice: Modern LLMs have 32-128 layers. The single-layer analysis cannot capture inter-layer information flow that is central to deep transformer reasoning. Assuming the MLP is pretrained and frozen eliminates the co-adaptation dynamics between attention and MLP that occur in real RLVR training. Only the Q matrix is trained, which is a dramatic simplification of real parameter updates. This is a mathematically convenient but highly specific structure. Real reasoning tasks have varying degrees of compositionality and do not admit clean algebraic characterizations. While the paper acknowledges these are simplifications, the gap between the theory and practice is large enough to question whether the identified mechanisms (critical horizon, relay effect) are the dominant factors in real RLVR or merely artifacts of the simplified setting.

2. The paper validates theory only on Z_96 cyclic group composition. While this is appropriate for verifying theoretical predictions, there are no experiments on more realistic reasoning tasks (even toy-scale NLP tasks) to suggest that the phenomena generalize. Even simple experiments on, e.g., multi-digit arithmetic or small-scale math word problems with actual language models would significantly strengthen the paper's practical relevance.

3. The critical horizon C_B = O(1) is determined by the pretrained MLP's feature structure (Equations 5a-5c). In real LLMs, what determines this "edge of competence"? The paper does not bridge the theoretical construct to observable or measurable properties of real models. Without this bridge, the theory's practical prescriptive value is limited -- practitioners cannot use it to design better curricula or data mixtures.

4. The relay effect (Theorem 5.3) holds when R is a moderate constant. But real RL datasets have complex, non-geometric difficulty distributions. The paper does not discuss how the theory's predictions change under more realistic difficulty spectra (e.g., continuous distributions, multi-modal difficulty, or problems with heterogeneous structure).

---

> ### Author Rebuttal · Authors · 2026-03-31
>
> We thank the reviewer for the positive feedback and address the concerns below.
>
> **Q1.The theoretical framework makes several strong simplifying assumptions.**
>
> **A1.** We respectfully argue that the concern is due to a mismatch between the stated goals of our theoretical framework and the standard expectations for engineering-oriented models.
>  - The modeling choices (e.g., single-laye) follow *standard transformer theory* [1–4], where simplifications enable tractable analysis. More importantly, our analysis yields a **non-trivial result and fills the gap in the literature**: a single-layer transformer trained solely on outcome rewards can learn compositional reasoning without intermediate supervision, which on its own is a novel contribution.
> - To provide better insight and leverage theoretical tools, we study compositional reasoning in a cleaner setting than practical LLMs. A more realistic setting would obscure the core mechanism, making it harder to disentangle components. For example, RLVR in LLMs is influenced by gradient variance, off-policyness, multi-layer dynamics, and complex data recipes. Our approach isolates the minimal structure needed to reveal the core compositional mechanism of RLVR and transformers. We refer to [5-6] for similarly foundational but simplified theories.
>
> **Q2. Experiments with actual language models.**
>
> **A2.** We thank the reviewer for the suggestion. We conducted an additional experiment on multi-digit multiplication using Qwen2.5-1.5B-Instruct, where digit count serves as the difficulty level.
>
> - **Setup.** We fine-tune with GRPO on N-digit $\times$ N-digit multiplication (exact match reward). Pre-training shows the competence edge at 3-digit (3-digit: 8%, 4-digit+: $\approx$0%), used as the starting level.
>
> - **Results.**
>   - *Mixed-difficulty:*
> | Setting | Step | 3-digit | 4-digit | 5-digit | 6-digit |
> |---|---|---|---|---|---|
> | M1: 3+4+5 ($R\approx1.33$) | 0 | 8% | 1% | 0% | 0% |
> |  | 500 | 91% | 76% | 40% | 11% |
> |  | 1875 | 94% | 82% | 61% | 15% |
> | M2: 3+6+12 ($R=2$) | 0 | 8% | 1% | 0% | 0% |
> |  | 500 | 91% | 47% | 2% | 0% |
> |  | 1875 | 93% | 25% | 0% | 0% |
>    - *Single-difficulty (eval on 4-digit):*
> | Setting | Step 0 | Step 100 | Step 200 | Step 1875 |
> |---|---|---|---|---|
> | 4-digit only | 1% | 2% | 75% | 88% |
> | 5/6-digit only | 0% | 0% | 0% | 0%  |
>
> We highlight three observations: (1) **Relay effect:**   in M1, sequential propagation ($3\text{d}\rightarrow4\text{d}\rightarrow5\text{d}$) with OOD generalization to 6-digit (15%). (2) **Large $R$ breaks relay:** in M2, transfer fails (4-digit 47%→25%, 5-digit+ 0%). (3) **Grokking:** in single-difficulty, 4-digit shows a plateau-then-jump (2%→75%). We agree broader empirical validation would further strengthen the paper and plan to include additional experiments in the revision.
>
> **Q3. Critical horizon $C_B$**
>
> **A3.** We agree that $C_B$ is not directly measurable outside controlled settings. However, it admits empirical proxies observable in practice. Metrics such as pass@k or non-trivial reward rates under outcome-only feedback capture the probability of sampling correct rollouts and indicate the edge of competence [7]. This is supported by recent empirical findings. For example, POPE [8] identifies a similar boundary, showing RLVR fails when correct trajectories are rarely explored. This aligns with our characterization that learning breaks down once difficulty exceeds a threshold. Our framework provides a principled interpretation of practices such as difficulty-based filtering and curriculum design, which keep training in regimes where successful exploration remains likely.
>
>  **Q4. Real RL datasets have complex, non-geometric difficulty distributions.**
>
> **A4.** Using a moderate constant $R$ is a deliberate simplification to obtain a clean result on how structured difficulty supports compositional learning. Extending to general spectra would require additional assumptions on task structure and data distribution, making unified analysis difficult. We believe such effort would be better suited for investigating other aspects of RLVR, e.g., multi-task setting. We therefore refrain from over-extending and hope our work serves as a basis for studying more complex, task-dependent regimes.
>
> ---
> **References**:
>
> [1] Jelassi et al.  Vision transformers provably learn spatial structure
>
> [2] Kim et al. Transformers provably solve parity efficiently with a chain of thought
>
> [3] Zhang et al. Trained transformers learn linear models in-context
>
> [4] Nichani et al. How Transformers Learn Causal Structure with Gradient Descent
>
> [5] Allen-Zhu et al., Towards understanding ensemble, knowledge distillation and self-distillation in deep learning
>
> [6] Cao et al. Benign Overfitting in Two-layer Convolutional Neural Networks
>
> [7] Zhang et al.  On the Interplay of Pre-Training, Mid-Training, and RL on Reasoning Language Models
>
> [8] Qu et al. Pope: Learning to reason on hard problems via privileged on-policy exploration

---

> > ### Author Rebuttal · Reviewer_rr3d · 2026-04-05
> >
> > I thank the authors for the clear and thoughtful rebuttal. The responses effectively address my concerns and further strengthen the paper.
> >
> > In terms of soundness and originality, the theoretical framework is well-grounded and provides novel insights into the learning dynamics of RLVR. The rebuttal clarifies the role of simplifying assumptions and appropriately positions the work within the context of theoretical analysis.
> >
> > Importantly, the additional experiment on multi-digit multiplication significantly improves the practical relevance of the paper, demonstrating that key theoretical predictions (e.g., relay effects and difficulty-dependent learning dynamics) also manifest in realistic settings. The discussion on empirical proxies for the critical horizon further helps bridge theory and practice.
> >
> > Overall, the rebuttal reinforces my confidence in the paper’s contribution. I maintain my recommendation.

---

### Decision · Program_Chairs · 2026-04-30

**Decision:**

Accept (regular)

**Comment:**

The paper provides a theoretical analysis of RLVR training dynamics on compositional reasoning tasks, using Fourier-based techniques to characterize how gradient signals depend on task difficulty and horizon. The main predictions — grokking-like plateaus under large difficulty gaps, smooth relay effects under gradual curricula — are validated on synthetic tasks and extended to multi-digit multiplication. The analysis is novel and addresses an important question about when and why RLVR works. One reviewer strongly objects, citing proof artifacts and an overly simplified task structure. Some of these concerns have merit — the single-layer transformer setting is limiting — but the theoretical insights are still valuable, and the objections do not invalidate the main results. On balance, I side with the majority that this is a meaningful theoretical contribution.